# Wnt signalosome assembly is governed by conformational flexibility of Axin and by the AP2 clathrin adaptor

Melissa V. Gammons [1,2,3] ✉, Elsa Franco-Echevarría[1,3], Tie-Mei Li[1,3], Trevor J. Rutherford [1], Miha Renko[1,2], Christopher Batters [1] & Mariann Bienz [1] ✉

Wnt signal transduction relies on the direct inhibition of GSK3 by phosphorylated PPPSPxS motifs within the cytoplasmic tail of the LRP6 co-receptor. How GSK3 is recruited to LRP6 remains unclear. Here, we use nuclear magnetic resonance spectroscopy to identify the membrane-proximal PPPSPxS motif and its flanking sequences as the primary binding site for both Axin and GSK3, and an intrinsically disordered segment of Axin as its LRP6-interacting region (LIR). Co-immunoprecipitation and CRISPR-engineered mutations in endogenous Axin indicate that its docking at LRP6 is antagonized by a phospho-dependent foldback within LIR and by a PRTxR motif that allows Axin and GSK3 to form a multi-pronged interaction which favors their detachment from LRP6. Crucially, signaling by LRP6 also depends on its binding to the AP2 clathrin adaptor. We propose that the Wnt-driven clustering of LRP6 within clathrin-coated locales allows the Axin-GSK complex to dock at adjacent LRP6 molecules, while also exposing it to co-targeted kinases that change its activity in Wnt signal transduction.

The Wnt/β-catenin signaling pathway ('canonical' Wnt signaling) is an ancient cell communication system that controls cell fates during animal development and stem cell renewal in adult tissues[1]. Dysregulation of canonical Wnt signaling can cause cancer in many tissues, most notably the bowel[2]. A key step in the transduction of an extracellular Wnt signal to the nucleus is the inhibition of glycogen synthase kinase 3 (GSK3). This serine/threonine (S/T) kinase is constitutively active, with widespread functions beyond Wnt signaling[3], but its association with the Axin scaffold protein directs its enzymatic activity towards β-catenin[4], to route this Wnt-specific substrate for ubiquitylation and proteasomal degradation[5]. However, upon Wnt stimulation of cells, the Axin-associated GSK3 is inhibited, causing stabilization of β-catenin, which thus accumulates in the cytoplasm and nucleus where it co-activates the transcription of Wnt target genes[6].

Wnt signal transduction is initiated by the simultaneous binding of Wnt to a Frizzled (FZD) receptor and its co-receptor LRP5/6 (Low-density lipoprotein receptor-related protein 5 or 6)[7,8]. This triggers the assembly of a membrane-associated signalosome[9], facilitated by Dishevelled, whose DEP domain binds to the cytoplasmic face of FZD[10–13], likely facilitated by Wnt-induced dimerization of FZD[14,15]. Once bound, Dishevelled activates phosphatidylinositol-4-kinase (PI4K) and phosphatidylinositol-4-phosphate 5-kinase (PIP5K) to drive local production of PIP(4,5)$_2$ (phosphatidylinositol 4,5-bisphosphate, PIP2 below) in the surrounding plasma membrane[16,17]. This locally enriched PIP2 ('PIP2 patch') increases the affinity between Dishevelled DEP and FZD, thereby consolidating their mutual association[12]. Dishevelled also uses its DIX domain to undergo limited head-to-tail oligomerization[18–20], which increases its local concentration at the receptor complex. It thus attains a high binding avidity for

[1]MRC Laboratory of Molecular Biology, Francis Crick Avenue Cambridge, Cambridge, UK. [2]Present address: Department of Medical Genetics, Cambridge Institute for Medical Research, University of Cambridge, Cambridge, UK. [3]These authors contributed equally: Melissa V. Gammons, Elsa Franco-Echevarría, Tie-Mei Li. ✉e-mail: mg2128@cam.ac.uk; mb2@mrc-lmb.cam.ac.uk

Axin with which it co-polymerizes through Axin's DIX domain (designated DAX), thereby recruiting Axin together with GSK3 and other degradasome components to the cytoplasmic tail (ctail) of LRP6, as elaborated below. Note that the affinity between DIX and DAX is in the mid micromolar range[21,22], which explains why these domains do not interact at the physiological sub-micromolar concentrations of Dishevelled and Axin[23,24] in the absence of DIX-dependent polymerization which enhances their mutual binding avidity.

Once assembled at the Wnt receptor complex, the signalosome initiates inhibition of GSK3 through five closely related proline-rich motifs (motifs A-E; PPP[S/T]Px[S/T], or PPPSPxS for short; Fig. 1a) within the ctail of the LRP6 co-receptor[25]. These inhibit GSK3 by binding directly to its catalytic pocket as pseudo-substrates, contingent on phosphorylation of their S/T residue at position P + 4[26–28]. This prevents GSK3 from phosphorylating substrates such as β-catenin and Axin, and the latter is dephosphorylated within half an hour of Wnt stimulation[29,30], potentially by protein phosphatase 1, which promotes Wnt signaling[30,31]. The pivotal phosphorylation at P + 4 of these motifs may be mediated by GSK3 itself[26,32,33] or by a cyclin-dependent kinase (CDK) activated by cyclin Y[34], an atypical member of the cyclin family that is myristoylated and recruits its cognate CDKs to the plasma membrane[35]. Once phosphorylated, the P + 4 residue primes phosphorylation of the P + 7 residue by casein kinase 1γ (CK1γ), a membrane-associated member of the CK1 family[32,36,37]. Dually phosphorylated PPPSPxS motifs (designated PPPpSPxpS) act synergistically and interchangeably to inhibit GSK3 in Wnt-stimulated cells[33], however

only the phosphorylation at P + 4 is essential for Wnt signal transduction while that at P + 7 is merely contributory[25]. Indeed, the latter is solvent-exposed in the co-crystal structures of GSK3 bound to PPPpSPxpS[27] and is dispensable for GSK3 inhibition in vitro[26], indicating a subsidiary role of CK1 in the direct inhibition of GSK3, but also posing the question why membrane-associated CK1γ promotes canonical Wnt signaling.

Previous evidence indicated that recombinant GSK3 binds directly to the membrane-proximal motif A with very low affinity once it is phosphorylated ($K_i$ 1–13 μM, depending on the assay[26,27]), but this interaction is not measurable with unphosphorylated motif. Therefore, it seems unlikely that GSK3 could dock at LRP6 at its normal physiological concentrations (~70–120 nM in human cell lines[23]). However, its association with phosphorylated LRP6 depends on Axin[25,32,38] to which GSK3 binds with mid nanomolar affinity[39]. Of note, the local concentration of Axin at the receptor complex is likely to be higher than its average cytoplasmic concentration (~110–150 nM[23]), owing to its co-polymerization with Dishevelled[21,22] which binds to FZD in the receptor complex upon Wnt stimulation (see above). Therefore, if GSK3 and Axin could bind as a complex to the LRP6 ctail cooperatively, their joint affinity would be substantially increased, which might enable them to overcome the affinity hurdle preventing their docking at LRP6. However, direct binding between recombinant Axin and LRP6 tail has never been demonstrated, owing to considerable technical challenges in expressing these largely unstructured proteins.

Here, we combine nuclear magnetic resonance (NMR) spectroscopy with functional assays in CRISPR-engineered human embryonic kidney (HEK293T) cells, to delineate and validate the elements within LRP6, GSK3 and Axin through which these signalosome components bind to each other to transduce Wnt signals. We thus found that GSK3 and Axin bind to the same site within the membrane-proximal LRP6 ctail with low-affinity and we defined an intrinsically disordered segment within Axin (called LIR, LRP6-interacting region) that mediates its binding to LRP6. Our evidence suggests that a phospho-dependent intramolecular foldback loop within Axin LIR antagonizes its binding to LRP6 in the absence of Wnt stimulation. Furthermore, we uncover a multi-pronged interaction between GSK3 and Axin that opposes the binding of GSK3 to LRP6 which may presage termination of Wnt signaling. Notably, efficient Wnt signal transduction by LRP6 additionally relies on its binding to the AP2 clathrin adaptor through tandem AP2-binding sites in its ctail. This consolidates previous evidence that LRP6 is targeted by AP2 to large clathrin-coated structures upon Wnt stimulation[40]. We propose that the clustering of LRP6 in these structures might allow a single Axin-GSK3 complex to bind to two juxtaposed ctails, enabling it to overcome the affinity hurdle that opposes its binding to LRP6 in the absence of Wnt. Additionally, the AP2-dependent localization of LRP6 in these clathrin locales could expose Axin to co-targeted kinases that may impact on its ability to transduce Wnt signals by regulating its conformation in a phosphorylation-dependent fashion.

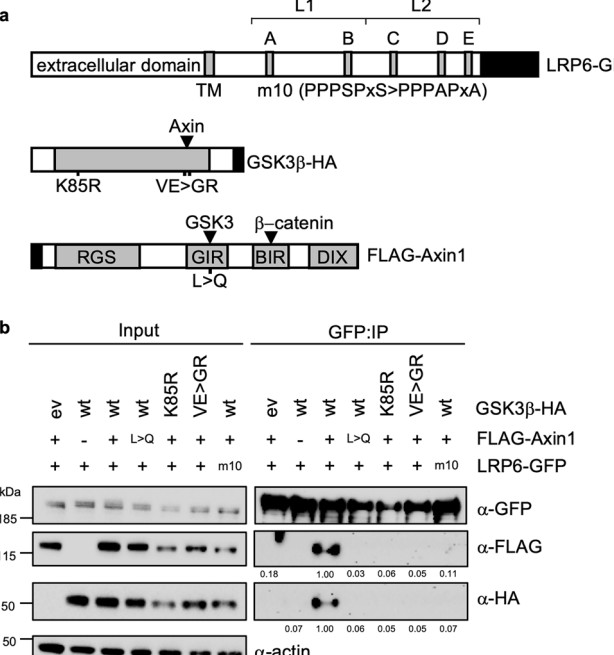

**Fig. 1 | Association of GSK3 and Axin with LRP6. a** *Top*, cartoon of LRP6-GFP, with transmembrane domain (TM) and wt (PPPSPxS) or m10 mutant (PPPAPxA) motifs A-E as gray bars; *brackets*, L1$_{LRP6}$ and L2$_{LRP6}$ fragments used for NMR; *below*, cartoons of GSK3β-HA and FLAG-Axin1, with folded domains (RGS, DIX) or α-helices (GIR, GSK3-interacting region; BIR, β-catenin-interacting region) as gray boxes; positions of mutations blocking ligand binding (VE > GR, L396Q) or catalytic activity (K85R) are indicated below cartoons; *black*, tags used for coIP assays. **b** CoIP assays in transiently transfected HEK293T cells; shown are Western blots probed with antibodies (α) as indicated on the right following immunoprecipitation (IP); numbers underneath blots indicate mean values relative to wt (= 1.00) as obtained by densitometry of IP blots normalized to corresponding input blots, each showing similar results across three independent experiments; *left*, positions of molecular weight markers (in kDa).

## Results

### Association of GSK3 and Axin with LRP6

We used co-immunoprecipitation (coIP) between LRP6-GFP, FLAG-Axin and GSK3-HA upon co-expression in HEK293T cells, to monitor their mutual association. Optimal binding conditions were determined by titrating the amounts of transfected DNA and their ratios (Supplementary Fig. 1). We found that robust coIP between the three proteins is only seen with their wild-type (wt) versions, but not with catalytically-dead GSK3 (K85R) nor with the non-phosphorylatable mutant LRP6m10 in which all five PPPSPxS motifs were mutated to PPPAPxA[32] (Fig. 1a, b). Furthermore, coIP between the three proteins is blocked by mutations in GSK3 or Axin that abolish their mutual binding (VE > GR, L > Q)[41] (Fig. 1b; Supplementary Fig. 1). Therefore, the

robust association of Axin and GSK3 with LRP6 requires their mutual binding and catalytic activity of GSK3.

## Binding sites of Axin and GSK3 in the LRP6 ctail

Since direct binding between Axin and LRP6 has never been demonstrated, we used NMR to test whether we could detect binding between recombinant proteins in solution. We expressed various

internal Axin fragments in bacteria, but this proved challenging, likely because they are intrinsically disordered[42]. However, we succeeded in purifying two Axin fragments upon tagging with a Lipoyl solubility tag (Lip-A3$_{Axin}$, A308-D426; Lip-A5$_{Axin}$, A308-V366; Fig. 2a). We incubated each of these fragments with an $^{15}$N-labeled fragment of LRP6 spanning its PPPSPxS motifs A and B (LRP6$_{1463-1538}$, or Lip-L1$_{LRP6}$; Fig. 1a), also intrinsically disordered (Supplementary Fig. 2a), which we used

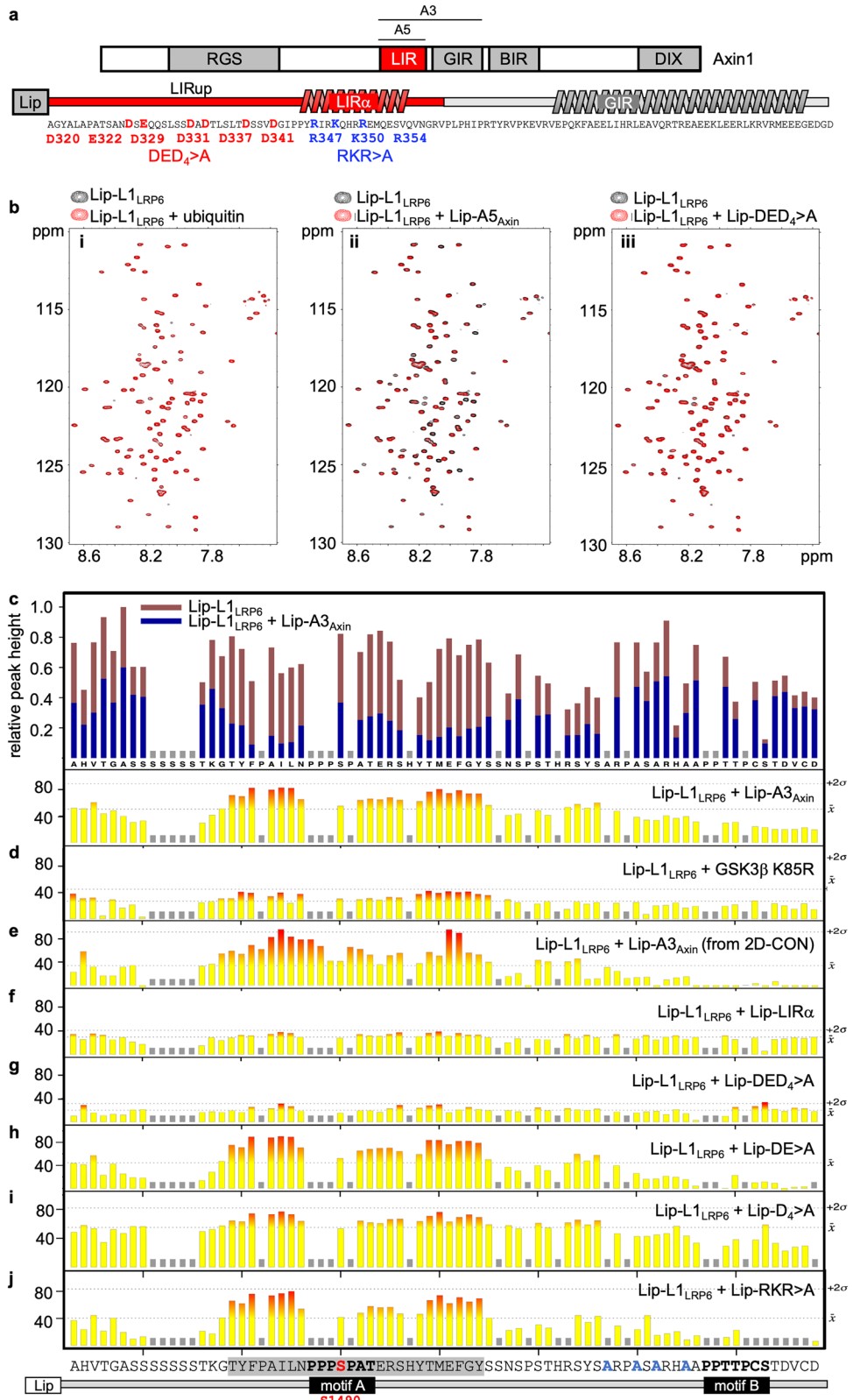

**Fig. 2 | Binding of Axin and GSK3 to the PPPSPAT element of LRP6. a** *Top*, cartoon of Axin1 (as in Fig. 1a), with extents of A3$_{Axin}$ (A3) and A5$_{Axin}$ (A5) fragments indicated; *below*, Lip-A3$_{Axin}$ comprising helical elements (LIRα, GIR) as predicted by AF2 and AF3, with sequence underneath; conserved residues mutated in DED$_4$ > A and RKR > A in bold; *red*, Axin LIR. **b** BEST-TROSY spectra of 80 µM $^{15}$N-labeled Lip-L1$_{LRP6}$ recorded by itself (*black*) or incubated with 80 µM ubiquitin, wt Lip-A5$_{Axin}$ or its DED$_4$ > A mutant (*red*), as indicated in color key above panels. **c** Peak heights in BEST-TROSY spectra of 80 µM $^{15}$N-labeled Lip-L1$_{LRP6}$ by itself (*brown*) or incubated with 80 µM Lip-A3$_{Axin}$ (*blue*) and corresponding bleach map underneath, depicting percentage attenuation of peak height by incubation with ligand ranging from strong (*red*) to weak (*yellow*); mean attenuations ($\bar{x}$) and means + 2 standard

deviations (+2σ) are given on the right in this and subsequent bleach maps. Gray bars indicate residues for which NMR peaks are not assigned or obscured by overlap. **d** Bleach map generated from BEST-TROSY spectrum of 80 µM $^{15}$N-labeled Lip-L1$_{LRP6}$ +/−80 µM of GSK3β-K85R. **e** Bleach map generated from 2D-CON spectrum of 150 µM $^{13}$C/$^{15}$N-labeled Lip-L1$_{LRP6}$ +/−150 µM Lip-A3$_{Axin}$ (see also Supplementary Fig. 2d, e). Bleach maps generated from BEST-TROSY spectra of 80 µM $^{15}$N-labeled Lip-L1$_{LRP6}$ +/−80 µM (**f**) Axin Lip-LIRα, (**g**) Lip-DED$_4$ > A, (**h**) Lip-DE > A, (**i**) Lip-D$_4$ > A or (**j**) Lip-RKR > A (all mutants in A5$_{Axin}$). *Bottom*, sequence of Lip-L1$_{LRP6}$ with PPPSPxS motifs A and B (*bold*), PPPSPAT element (*gray underlay*) and S1490 (*red*); *blue*, YYYF > A substitution, to render Lip-L1$_{LRP6}$ soluble in aqueous buffer (see Methods).

initially to test binding since an LRP6 truncation retaining these motifs suffices to confer efficient signaling activity in Xenopus embryos and mammalian cell-based assays[25,36].

We observed significant line broadening (a spectral perturbation termed 'bleaching' below) of multiple resonances in BEST-TROSY spectra recorded from $^{15}$N-labeled Lip-L1$_{LRP6}$ upon incubation with either Axin fragment but not with a ubiquitin control (Fig. 2b, c; Supplementary Fig. 2b). Resonance assignment of $^{15}$N-$^{13}$C double-labeled Lip-L1$_{LRP6}$ allowed us to generate a bleach map, since each bleaching event provides evidence that the corresponding LRP6 residue undergoes an interaction with Axin. The heat maps for A3$_{Axin}$ and A5$_{Axin}$ are essentially indistinguishable, with the bleached residues forming a contiguous patch in each case (Fig. 2c; Supplementary Fig. 2b). This patch is centered on motif A (PPPSPAT) previously implicated in binding to GSK3[26]. Indeed, incubation of $^{15}$N-labeled Lip-L1$_{LRP6}$ with wt or catalytically-dead GSK3 produced almost indistinguishable bleach maps (Fig. 2d; Supplementary Fig. 2c), each with similar limits as the Axin-binding site of LRP6. We conclude that Axin and GSK3 bind essentially to the same site within the membrane-proximal region of LRP6. We shall refer to this Axin- and GSK3-binding site as the PPPSPAT element (T1477-Y1504; Fig. 2, *bottom*). Of note, phosphorylation of this element is not essential for the binding by either protein in vitro.

Interaction with prolines cannot be observed by recording BEST-TROSY spectra since these show one peak for each amide H-N in the peptide backbone, which is present in all amino acids except proline. To observe bleaching of the proline-derived peaks within the PPPSPAT element, we recorded $^{13}$C-detected 2D-CON spectra of double-labeled Lip-L1$_{LRP6}$ probed with Lip-A3$_{Axin}$ (GSK3 binding to Lip-L1$_{LRP6}$ is too weak to produce meaningful results with this approach). This revealed significant bleaching of four of the ten proline peaks obtained with this fragment (corresponding to P1482 and P+1, P+2 and P+5 of PPPSPAT; Fig. 2e; Supplementary Fig. 2d, e), consolidating our evidence for the binding of Axin to motif A. Structural prediction by Alphafold 2 (AF2)[43] and AF3[44] revealed a short stretch with weak helical potential (P345-Q361), called Axin LIRα (Fig. 2a) within our shortest Axin fragment (A5$_{Axin}$). However, we do not observe any interactions of the minimal LIRα nor of the Axin sequences upstream of LIRα (A308-P344, called LIRup; Fig. 2a) with $^{15}$N-labeled Lip-L1$_{LRP6}$ (Fig. 2f). Therefore, the minimal fragment of Axin that binds to LRP6 (A308-V366) is bipartite as it spans both LIRα and LIRup (Fig. 2a, *red*). This bipartite Axin element is termed LRP6-interacting region (LIR) below.

Next, we used isothermal calorimetry (ITC) to determine the binding affinity between Lip-A5$_{Axin}$ and Lip-L1$_{LRP6}$. This allowed us to calculate an apparent $K_D$ of 1.37 µM (Fig. 3a), assuming a 1:1 stoichiometry of the complex. The molar ratios are <1 in these measurements, likely because of the inevitable partial degradation during the preparations of wt and mutant Lip-A5$_{Axin}$ proteins (Supplementary Fig. 3), suggesting that the corresponding calculated $K_D$ values are underestimates. By contrast, binding of GSK3 to Lip-L1$_{LRP6}$ is barely measurable (Fig. 3b), indicating that the affinity of GSK3 for the PPPSPAT element is at least two orders of magnitudes lower than the affinity of Axin for this element.

JackHMMER analysis[45] revealed that the sequences within Axin LIRup are unusually rich in negatively-charged residues, some of them highly conserved, including five aspartates (D) and a glutamate (E; Fig. 2a). To test the functional relevance of these six residues for LRP6 binding, we substituted each with alanine (Lip-DED$_4$ > A), which reduced binding to Lip-L1$_{LRP6}$ to background levels (Figs. 2g, 3c) while binding of partial mutants (DE > A, D$_4$ > A) was still detectable (Fig. 2h, i). In contrast, alanine substitutions of three basic residues (RKR > A) whose side-chains project from the same surface of Axin LIRα (R347, K350, R354; see below) did not block binding to Lip-L1$_{LRP6}$ (Fig. 2j). Although RKR > A appeared to reduce the $K_D$ value slightly (Fig. 3d), this value may not be directly comparable to that determined for its wt control, given that A5$_{Axin}$ is prone to degradation (Supplementary Fig. 3). We conclude that the acidic residues within Axin LIRup are essential for its binding to LRP6 while the RKR residues within LIRα are at best contributory.

We also asked whether Axin can bind to the distal LRP6 ctail (spanning PPPSPxS motifs C-E; Fig. 1a). We incubated an $^{15}$N-labeled distal fragment (LRP6$_{1539-1613}$, Lip-L2$_{LRP6}$) with Lip-A3$_{Axin}$, which revealed weak bleaching of multiple peaks in the BEST-TROSY spectra, mostly near motif D (Supplementary Fig. 4a). Similarly, some of these residues were perturbed weakly upon incubation of Lip-L2$_{LRP6}$ with GSK3 (Supplementary Fig. 4b). Of note, motif D (CPPSPYT) is the most divergent of all five PPPSPxS motifs, with barely any activity in conferring signaling activity in cells[33], and may therefore serve as a secondary LRP6 binding site for Axin and GSK3. This might explain the reduced signaling activity of LRP6 and LRP5 truncations from which the distal-most two motifs were deleted[36,46]. However, binding of GSK3 or Axin to the CPPSPYT element is not detectable by ITC, suggesting that their affinity for this secondary site is exceedingly low (estimated $K_D$ > 500 µM).

## A phospho-dependent foldback loop within Axin LIR

Axin LIRup contains three near-invariant serines (S317, S321, S325; called 'S1 triplet'; Fig. 4a, *green*) that can be phosphorylated by GSK3 in the absence of Wnt[39]. Indeed, a search of the Kinase Library (KL)[47] via PhosphoSitePlus[48] predicts with high confidence that S317 and S321 are GSK3 phospho-acceptor sites contingent on a CK1-catalyzed priming phosphorylation of S325 (Fig. 4a, *bottom*). Immediately adjacent is a second S/T triplet (S328, T332, T336; Fig. 4a, *orange*; 'S2 triplet' below) where phosphorylation of S325 primes phosphorylation of S328 by CK1 (Fig. 4a, *bottom*). These S1 and S2 triplets appear to be dephosphorylated upon Wnt stimulation[49,50], along with GSK3 phospho-acceptor sites downstream of the β-catenin-interacting region of Axin[29,30] (called 'S3 and S4 doublets'; Fig. 4a).

To test the function of these phospho-acceptors, we used coIP assays to monitor association of non-phosphorylatable alanine substitution mutants of Axin (S1 > A, S2 > A, S3 > A, S4 > A) with LRP6-GFP. To our surprise, the coIP signals obtained with S1 > A and S2 > A are markedly stronger than their wt Axin control (Fig. 4b, c). As expected (Fig. 1b), the enhanced signal depends on binding to, and catalytic activity of, GSK3 (Fig. 4b). An enhanced coIP signal is also seen with the phospho-mimic mutant S1 > D albeit not with S2 > D (Fig. 4c). By

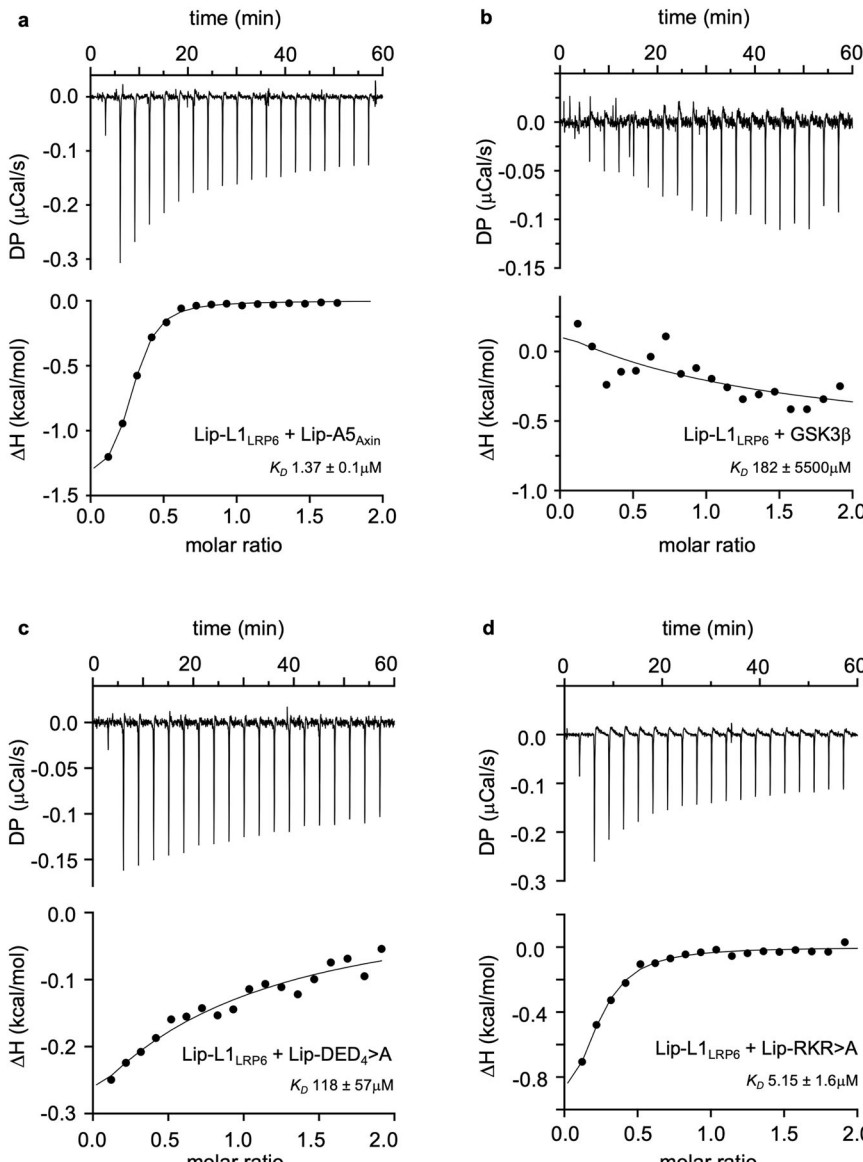

**Fig. 3 | Binding affinities between LRP6 and Axin or GSK.** ITC profiles of 50 μM (**a**) Lip-A5$_{Axin}$, (**b**) GSK3β, (**c**) Lip-DED$_4$ > A or (**d**) Lip.RKR > A titrated with 500 μM Lip-L1$_{LRP6}$ (see also Fig. 2); in each panel, the $K_D$ value was calculated from three independent experiments, and its standard deviation is given. The non-sigmoidal curve and large standard error of fit in the case of GSK3β (**b**) indicates that its binding to Lip-L1$_{LRP6}$ is barely detectable. Note also that the molar ratios are <1 in the case of Lip-A5$_{Axin}$ because of the partial degradation during the preparations of wt and mutant Lip-A5$_{Axin}$ proteins (Supplementary Fig. 3), suggesting that the calculated $K_D$ values are underestimates.

contrast, coIP is not affected by any of the mutations of the S3 and S4 doublets (Fig. 4c). We hypothesized that the phosphorylation of the S1 triplet may oppose the binding of Axin to LRP6 through its LIR, for example through a phosphorylation-dependent intramolecular interaction between Axin LIRα and LIRup. Of note, the charge distribution is strikingly asymmetric even within unmodified LIR, with negative charges concentrated in LIRup and positive charges in LIRα (Fig. 4d, e). This asymmetry would become even more pronounced upon phosphorylation of the serine residues in LIRup.

To test this hypothesis, we used ITC to measure binding between Axin LIRα and a synthetic 43mer peptide spanning Axin LIRup (A308-Q351) with or without phosphorylation of the S1 triplet (Fig. 4a, *green*). Robust binding strictly depends on phosphorylation of this triplet, with an estimated $K_D$ in the mid micromolar range (Fig. 4f), which represents an efficient intramolecular interaction. Importantly, the RKR > A triple-mutation reduces binding by ~10x to near-background levels (Fig. 4f, *right*). This indicates that the phosphorylated S1 triplet in

LIRup engages with the positively-charged RKR residues on the LIRup-facing surface of Axin LIRα in a direct intramolecular interaction, thus forming a foldback loop within Axin LIR (Fig. 4g) that may compromise its ability to bind to LRP6 (see Discussion).

## A multi-pronged interaction between Axin and GSK3

Both versions of AlphaFold predict confidently (in each of the top-5 models; Supplementary Fig. 5) that the GSK3-interacting α-helix of Axin (GIR) extends further downstream than the minimal α-helix observed in Axin-GSK3 co-crystals[27,51] (Fig. 5a, *red*). Indeed, probing ¹⁵N-labeled Lip-A3$_{Axin}$ with GSK3 by NMR revealed extensive interactions with residues upstream and downstream of Axin GIR. Strong GSK3-induced bleaching is shown by a single arginine (R347) and histidine (H352) in Axin LIRα (Fig. 5a, *dark gray underlay*), and by a threonine (T374) within a highly conserved PRTxR motif (Fig. 5a, *underlined*) located between Axin LIRα and GIR. Therefore, Axin binds to GSK3 through LIRα, PRTxR and the whole GIR-containg α-helix to its

predicted C-terminal end – a far more extensive interaction than that seen in co-crystal structures[27,51].

To gain further insight into this interaction, we docked Axin with GSK3 using AF2 and AF3. Both versions predict that Axin LIRα (Fig. 5b, c, *turquoise*) and GIR (Fig. 5b, c, *dark gray*) bind to GSK3 (Fig. 5b, c, *light gray*; shown in surface representation) in a multi-pronged interaction, which is mediated essentially by the same LIRα

surface in both models, i.e. the GSK3-facing surface defined by residues H352 and S359 (Fig. 5d, e), leaving the RKR residues on the opposite side of LIRα solvent-exposed (Fig. 5f, g). Strikingly, both models predict that the PRTxR motif between LIRα and GIR binds across the priming site of the catalytic pocket of GSK3, perpendicularly to known pseudo-substrates[27] (Fig. 5h–j). Although this pose of PRTxR is similar in the two models, its proline (P372) appears to be

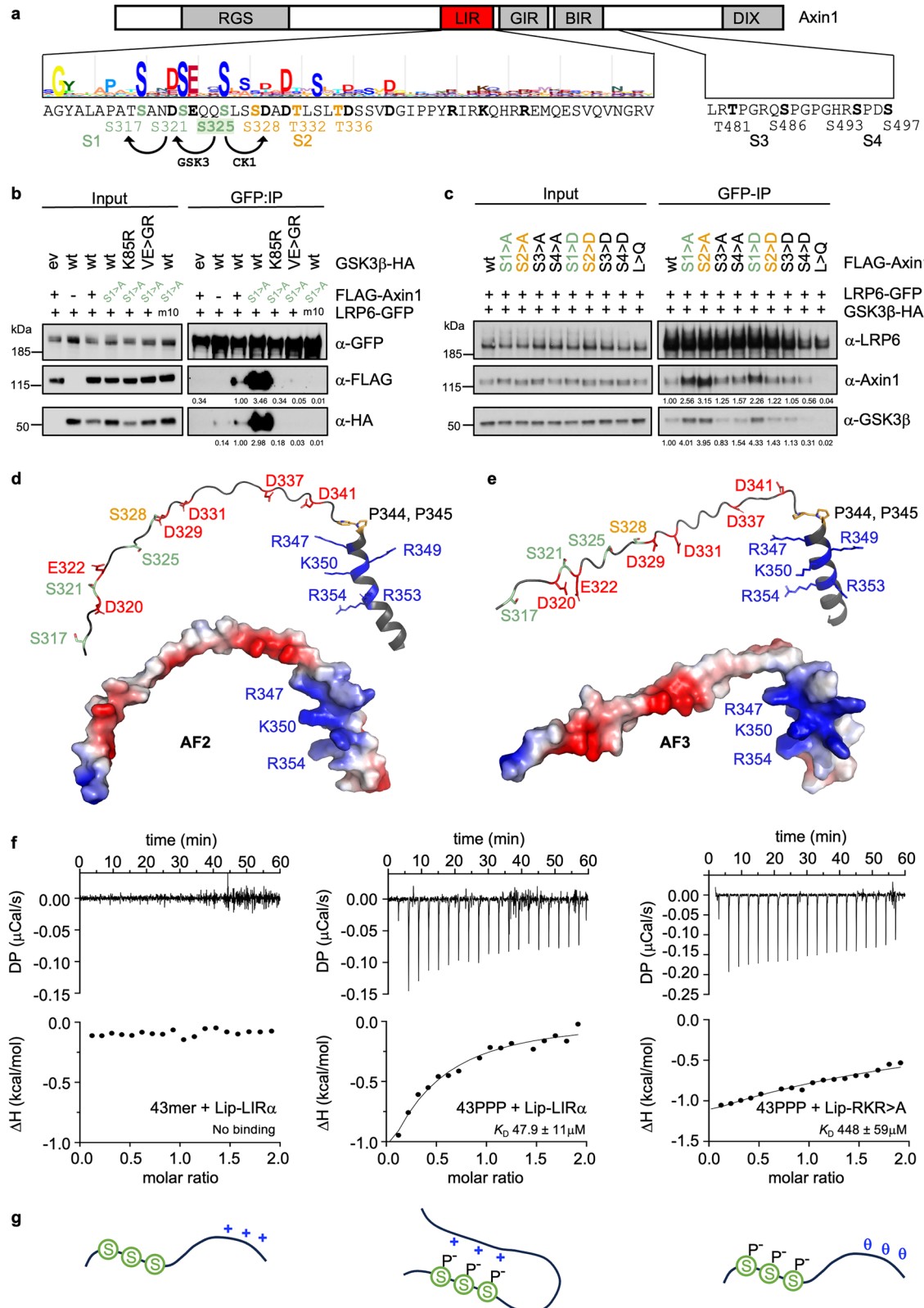

**Fig. 4 | A phospho-dependent foldback loop in Axin LIR. a** *Top*, cartoon of Axin1 (as in Fig. 1a), with sequence of LIR underneath whose evolutionary conservation is indicated by JackHMMER plot; *below*, predicted GSK3 and CK1 phospho-acceptor sites mutated in $S_4 > A$ (*green*); S325, priming residue for CK1; *green, orange*, mutated S1 and S2 triplet residues; *bold*, other mutated residues (including S3 and S4 doublets). **b, c** CoIP assays as in Fig. 1, following co-expression of wt or mutant FLAG-Axin1, LRP6-GFP and GSK3β-HA, as specified in panels; colors as in (**a**); numbers underneath blots indicate mean values relative to wt (= 1.00) as obtained by densitometry of IP blots normalized to corresponding input blots, each showing similar results across eight independent experiments; *left*, positions of molecular weight markers (in kDa). An enhanced coIP signal for the S1 > A, S2 > A and S1 > D mutants was consistently observed compared to their wt Axin control.

**d, e** Structural predictions of Axin LIR by AF2 or AF3 as indicated, with conserved residues colored as in (**a**) in stick representations (*above*), revealing striking clustering of negative (*red*) and positive (*blue*) charges in surface representation (*below*); note also kink induced by tandem prolines (P344, P345). **f** ITC profiles of 50 μM unphosphorylated (43mer) or phosphorylated (43PPP) Axin LIRup peptides titrated with 500 μM wt or RKR mutant Axin Lip-LIRα, revealing phospho-dependent interaction between phosphorylated LIRup and positively-charged LIRα; in each panel, the $K_D$ value was calculated from three independent experiments, and its standard deviation is given. Note that the LIRup peptides include 7 residues of LIRα (to render them soluble in aqueous buffer) which may allow residual intramolecular folding back of the carboxy terminus in the case of 43PPP, in which case the calculated $K_D$ value for 43PPP may represent an underestimate. **g** Cartoon depicting intramolecular interaction within Axin LIR ('foldback loop') which depends on phosphorylation of Axin LIRup.

structurally important only in the AF2 model where its side-chain faces GSK3, but not in the AF3 model in which its side-chain is solvent-exposed.

By contrast, the two AlphaFold versions predict different poses for LIRα: in the AF2 model, LIRα interacts loosely through H352 and S359 (and additional residues) with the 'front' surface of the N-lobe of GSK3 (Fig. 5d) near the catalytic site (Fig. 5b, *yellow asterisk*), albeit with low confidence (Supplementary Fig. 5). By contrast, the AF3 model predicts that LIRα bends round the apex of the N-lobe of GSK3 (Fig. 5c), forming close contacts with its 'back' surface through H352 and S359 (Fig. 5e). Additionally, AF3 but not AF2 predicts a loose interaction of LIRup with the catalytic pocket of GSK3 (Fig. 5c), with each of the top-5 models placing the C-terminus of LIRup across its priming site, albeit with low confidence (Supplementary Fig. 5). In summary, while the prediction of LIRα binding to GSK3 is borne out by our NMR recordings (Fig. 5a), the pose of LIRα on GSK3 remains uncertain. Indeed, LIRα may oscillate between different poses.

**Testing LIR function in CRISPR-engineered Axin1-mutant cells**

Next, we sought to test the signaling function of some of the key residues of Axin identified in our binding assays. We used CRISPR engineering to introduce point mutations into endogenous Axin1 in HEK293T cells lacking Axin2 (Supplementary Fig. 6; of note, the Wnt response appears normal in this Axin2 KO line as wt Axin1 compensates for the loss of Axin2). First, we introduced alanine substitutions into five of the six above-mentioned acidic residues ($DED_3 > A$; Fig. 6a) that are critical for Axin LIR binding to LRP6 (Fig. 2g), but also into the S1 triplet serines and S328 ($S_4 > A$; Fig. 6b) within LIRup (as these four serines are encoded within a single exon of Axin1; see Methods), because their mutations enhance Axin's coIP with LRP6 (Fig. 4b, c). We isolated multiple independent lines in each case and tested their Wnt responses with SuperTOP[52], a sensitive reporter assay for quantifying β-catenin-dependent transcription.

We observed much reduced Wnt inducibility in both cases, and also somewhat enhanced activity in the absence of Wnt (Fig. 6a, b), especially in the case of $S_4 > A$ (Fig. 6b). However, each of these mutant lines also show a marked reduction of the Axin levels (Fig. 6a, b, f), also seen to a lesser extent in mutant lines bearing the RKR > A triple-mutation (Fig. 6c, f) though neither in P372A nor in RR > D mutant lines (Fig. 6d–f; see also below). A similar reduction of the Axin levels was previously found in pulse-chase experiments, which revealed that the turnover of $S_3 > A$ triple-mutant Axin is accelerated compared to wt Axin[50] (see also ref. 53, for identification of destabilizing sequences overlapping Axin LIR). Most likely, the reason for the marked destabilization of these Axin mutants is that $S_4 > A$ creates alanine substitutions of the serines of two conserved P/AxxS motifs (PATS and ANDS) that are essential for Axin's binding to the USP7 de-ubiquitylase[54] (Fig. 5a, *light gray underlay*; see also Supplementary Fig. 7): USP7 guards Axin against its ubiquitin-dependent degradation[54] which can be promoted by the SIAH ubiquitin E3 ligase[55] (whose cog-

nate binding site in Axin is VRVEP, immediately upstream of its GIR[55]; Fig. 5a, *light gray underlay*). Binding of USP7 may also be impaired by $DED_3$, given that the serine in the ANDS motif is flanked by D320 and E322 that are substituted in this mutant (Fig. 5a). This destabilization of Axin by $S_4 > A$ and $DED_3$ complicates the interpretation of their mutant phenotypes. However, the reduced Wnt response consistently observed in both mutants supports the notion that the conserved acidic residues and interspersed serines within LIRup contribute to the binding between Axin and LRP6.

The proline in the PRTxR motif (P372) is near-invariant across the animal kingdom (Fig. 5a; Supplementary Fig. 7) and is predicted to be crucial for placing PRTxR across the catalytic pocket of GSK3 in the AF2 model (Fig. 5h). We therefore substituted this proline with alanine (P372A) and examined the Wnt response of two independent mutant lines. In contrast to $S_4 > A$ and $DED_3$, P372A does not affect the levels of Axin (or GSK3; Fig. 6d, f) nor the pattern of LRP6 phosphorylation as monitored by the phospho-specific α-pS1490 antibody (M. V. G., unpublished). Moreover, we find that each P372A line is hyperinducible by Wnt, while there is only background activity without Wnt. These results are mirrored by Wnt-dependent increases of active β-catenin in both wt and mutant lines as detected by Western blotting (Fig. 6d, f). We conclude that P372 functions to attenuate Wnt-induced β-catenin activation in HEK293T cells without affecting the basal β-catenin levels prior to Wnt stimulation, implicating the multi-pronged Axin-GSK3 complex in the termination of the Wnt response.

We also observe a mild Wnt hyperactivity in two independent lines bearing RKR > A (Fig. 6c). Recall that RKR > A blocks the formation of the intramolecular foldback within Axin LIR (Fig. 4d–f), thereby promoting the binding of Axin to LRP6, which could explain the elevated Wnt activity of the RKR > A mutant cells. However, it is unlikely that this phenotype reflects an impairment of the multi-pronged interaction between Axin and GSK3, given that the positively-charged side chains of the RKR residues in this multi-pronged complex are solvent-exposed in both AF models (Fig. 5f, g).

We also examined the function of the two arginines on the opposite surface of LIR (R349 and R353), which may interact with GSK3 (Fig. 5f, g), testing the effects of an RR > D double-mutation aimed at repelling this interaction. Indeed, both mutant lines show a tendency to be mildly hyperinduced by Wnt, although the values are not statistically significant, owing to variation between the two lines (Fig. 6e). This suggests that these arginines contribute only minorly to the multi-pronged interaction between Axin and GSK3 – perhaps unsurprisingly, given that this interaction also relies on other LIRα residues, including H352 and S359 (Fig. 5d, e). Nevertheless, despite its weak phenotype, the RR > D double-mutant corroborates our conclusion derived from P372A that the multi-pronged interaction between Axin and GSK3 attenuates the Wnt response, presumably by obstructing the binding of GSK3 to LRP6 (see Discussion).

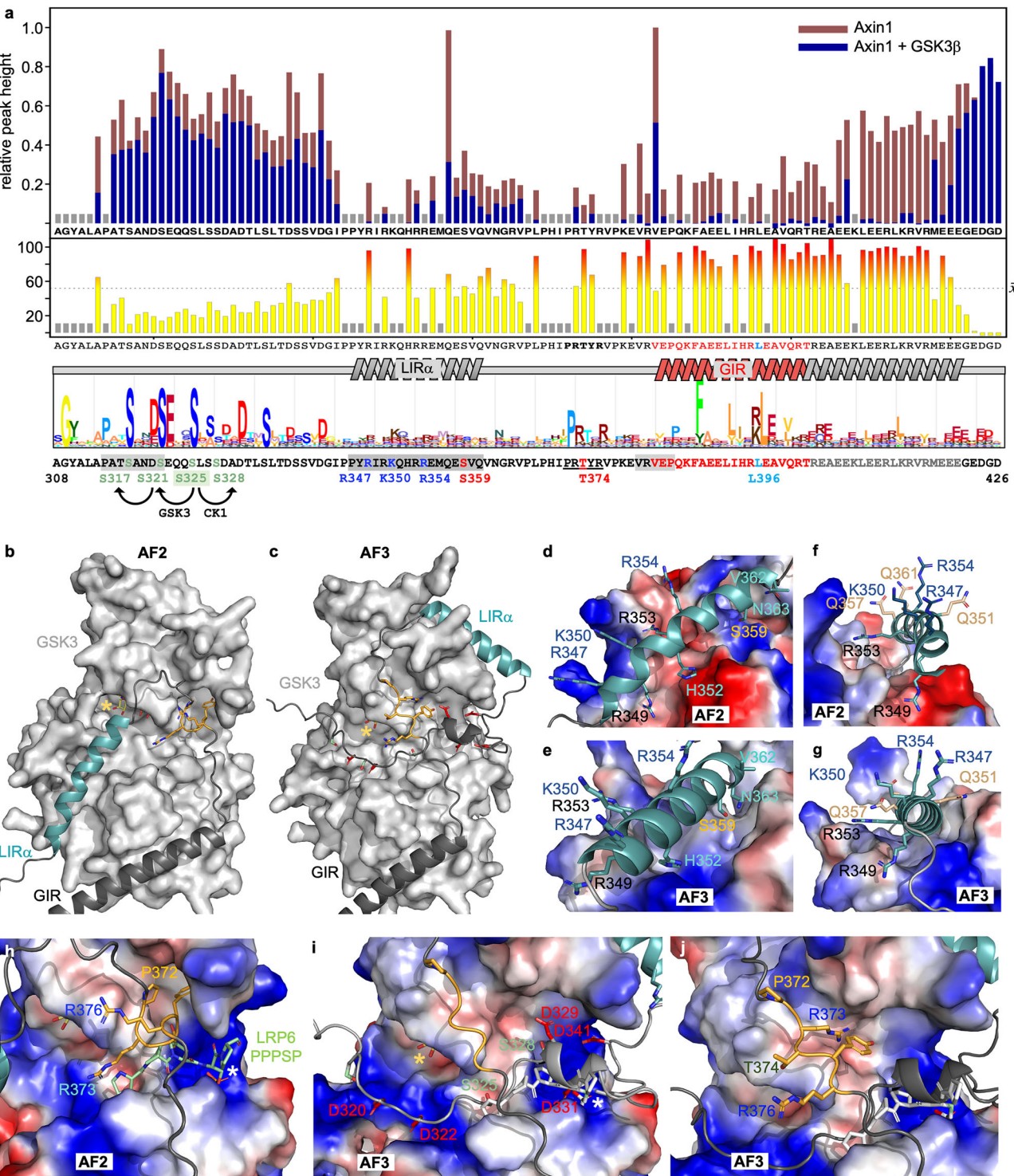

**Fig. 5 | A multi-pronged interaction between Axin and GSK3. a** Relative peak heights in a BEST-TROSY spectrum of 80 μM $^{15}$N-labeled Lip-A3$_{Axin}$ +/−80 μM GSK3β-K85R and corresponding bleach map (*underneath*), with coloring and labeling as in Fig. 2c; *below*, sequence of A3$_{Axin}$ (conservation indicated by JackHMMER plot) spanning Axin LIRα (*dark gray underlay*) and GIR α-helices as predicted by AF2 and AF3; *red*, minimal Axin GIR visible in crystal structures of the GSK3-Axin complex[27,51]; *light blue*, L396 mutated in L > Q (see Fig. 1a); *underlined*, conserved PRTxR motif; *light gray underlay*, binding sites for USP7 (PATS, ANDS) or SIAH (VRVEP); see also Fig. 4a (for residue labeling and colors). **b, c** Surface

representations of GSK3 (*light gray*) forming multi-pronged interactions with Axin as predicted by AF2 and AF3 through GIR (*dark gray*) as indicated in panels; Axin LIRα (*cyan*) and intervening PRTxR motif (*gold*), with P372, R373 and R376 in stick representation; *yellow*, ATP (visible in co-crystal structure; PDB code 4NM5); *yellow asterisks*, catalytic site. Zoomed views of (**d**–**g**) LIRα-binding grooves and (**h**–**j**) catalytic pocket of GSK3 in electrostatic surface representation, interacting with (**d**–**g**) LIRα (*cyan*) or (**h**–**j**) PRTxR motif (*gold*) in stick representation; *light green* (**h**) or *white* (**i, j**), LRP6 PPPpSP (PDB code 4NM7); *white asterisks*, priming site; *yellow asterisk*, catalytic site.

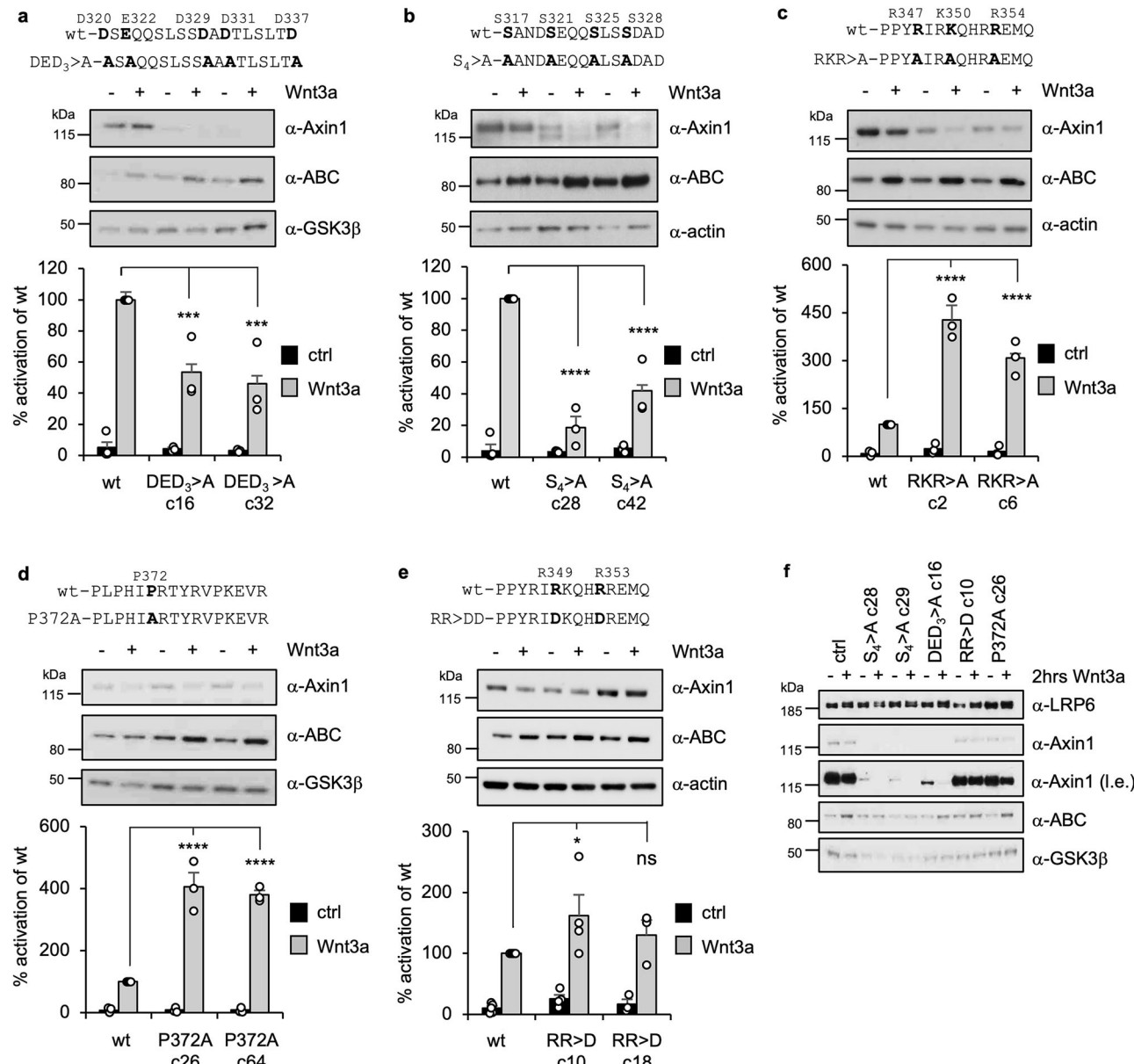

**Fig. 6 | Wnt responses of cells bearing Axin1 point mutants.** SuperTOP assays and corresponding Western blots of HEK293T lines bearing CRISPR-engineered **a** DED₃ > A, **b** S₄ > A, **c** RKR > A, **d** P372A or **e** RR > D mutations in endogenous Axin1 (in an Axin2 KO background) +/− Wnt3a, as indicated; representative Western blots from the same lysates were probed with antibodies as indicated on the right; *below*, names of selected mutant lines; *open circles*, individual values obtained from at least three independent experiments are set relative to the mean value of three samples from Wnt-stimulated parental HEK293T cells (wt, set to 100%); statistical significance was determined by one-way ANOVA with multiple comparisons, comparing each treatment to Wnt-stimulated wt cells (+) following normalization;

all *p*-values are too small to be determined exactly (****, $p < 0.0001$) except for (**a**) ***, $p = 0.0008$ (wt+ vs DED₃ c16 +) or ***, $p = 0.0002$ (wt+ vs DED₃ c32 +); (**e**) *, $p = 0.0448$ (wt+ vs RR > D c18 +) or *ns* (not statistically significant), $p = 0.6179$ (wt+ vs RR > D c18 +); *error bars*, ± SEM. **f** Western blot representative of three independent experiments, showing comparative analysis of selected mutant lines (+/− Wnt3a) shown in (**a**–**e**); l.e., longer exposure of the α-Axin1 blot. Note the marked destabilization of DED₃ > A and S₄ > A observed in each independently isolated mutant line, which may reflect reduced binding of the USP7 de-ubiquitylase to these LIRup mutants (see text).

## Functional AP2-binding sites within LRP6

We previously found that the sequences between PPPSPxS motifs A and B of the LRP6 ctail (called B element) are required for signaling activity[56]. This element contains two tandem Y-motifs (Fig. 7a) which may bind to the AP2 clathrin adaptor, based on previous evidence from coIP and endocytosis assays[40,57]. LRP6 and FZD are continuously subject to clathrin-dependent endocytosis and lysosomal degradation, apparently by Dishevelled-dependent coupling to the transmembrane ubiquitin ligase ZNRF3/RNF43[58,59]. Targeting of FZD and LRP6 to endocytic clathrin-coated pits appears to rely on an AP2-binding motif

of Dishevelled[60] and/or on the tandem Y-motifs of LRP6[57]. Indeed, the B element spans the sequences YRPY and YRHF, where the latter conforms to the classical YxxΦ motif recognized by the cargo-binding μ subunit of AP2 in the 'open' form of the AP2 complex following its activation by PIP2-mediated recruitment to the plasma membrane[61–63]. However, YxxΦ is poorly conserved beyond vertebrates and missing from LRP5 paralogs whereas YxxY is near-invariant amongst LRP5/6 orthologs throughout the animal kingdom (Supplementary Fig. 8). We note that YxxY functions as an endocytic signal in Drosophila cells[64] and may thus represent a variant YxxΦ motif in this invertebrate.

Nevertheless, it is unclear whether YxxY can bind to AP2µ since the hydrophobic Φ-binding pocket of AP2µ may not tolerate the polar hydroxyl group of the tyrosine at P + 3[65]. Indeed, co-crystal structures of the cargo-binding domain of AP2µ with various YxxΦ motifs have identified F, L, I, M and V as the only residues that can be accommodated by their Φ pocket[62,66,67].

To test whether YRPY can bind to AP2, we purified recombinant YxxΦ-binding domain of rat AP2µ[65] to monitor its binding to a minimal Lip-tagged SYRPYS peptide by ITC. This was not feasible though, because the hydrophobicity of this peptide renders it insoluble in the ITC buffer. However, a minimal SYRHFA peptide binds to AP2µ with low affinity ($K_D$ 16.2 ± 1.2 µM; Fig. 7b) as expected for a

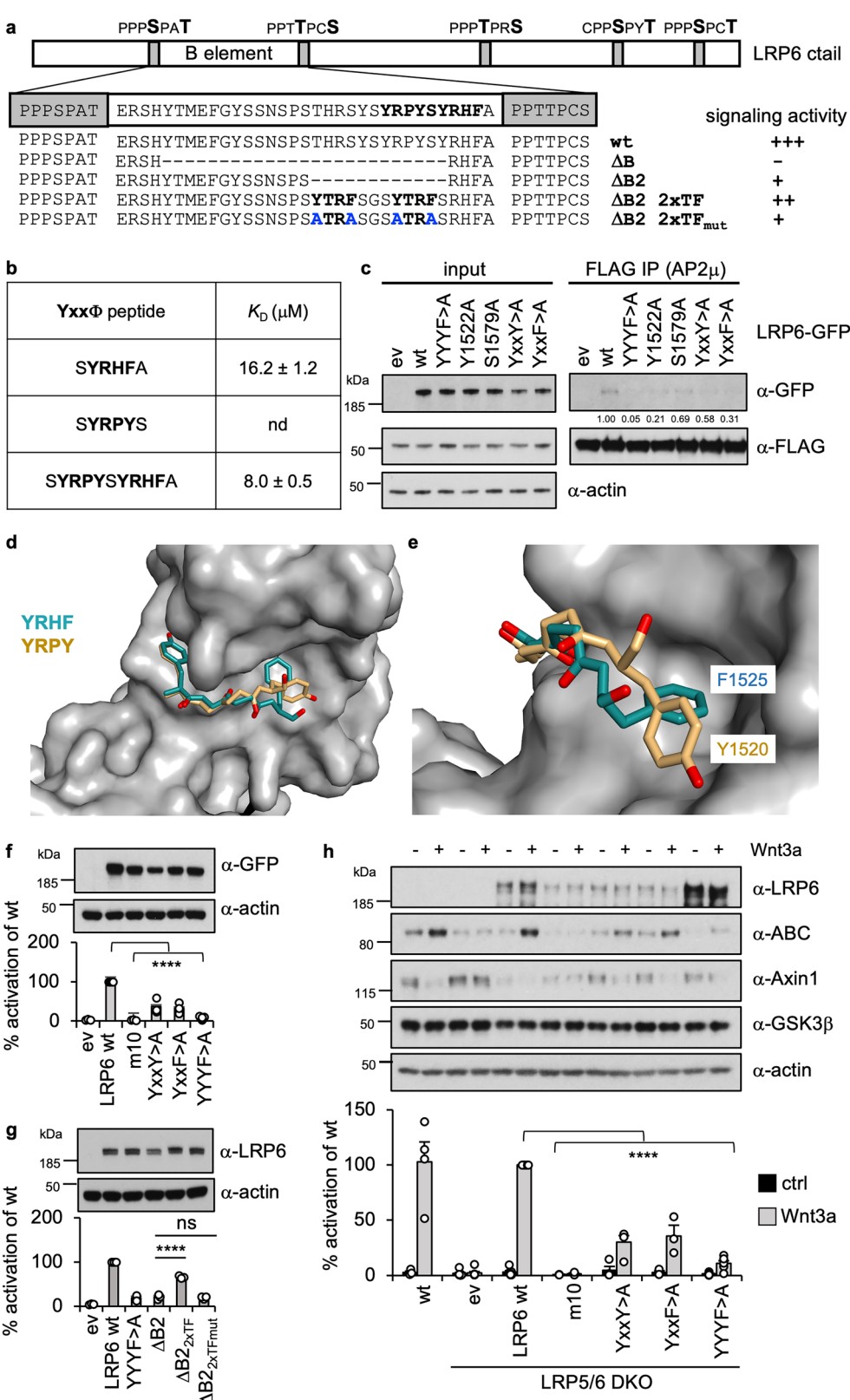

**Fig. 7 | Functional AP2 binding sites within LRP6. a** Cartoon of the LRP6 ctail with PPPSPxS motifs (*gray bars*), with sequence of wt and mutant B elements underneath (for sequence conservation, see Supplementary Fig. 8), and corresponding signaling activities (from SuperTOP assays shown in (**f**–**h**) and Supplementary Fig. 10a). **b** Binding affinities determined by ITC, following titrations of His-tagged AP2μ with His-tagged wt or mutant peptides containing YxxΦ motif (*bold*); $K_D$ values were calculated from three independent experiments and their standard deviations are given (±SEM); nd, binding not detectable. **c** CoIP assays as in Fig. 1, following co-expression of wt or mutant LRP6-GFP with FLAG-AP2μ, as indicated; numbers underneath blots indicate mean values relative to wt (= 1.00) as obtained by densitometry of IP blots normalized to corresponding input blots, each showing similar results across three independent experiments. **d, e** Surface representations

of AP2μ (*light gray*) bound to YxxΦ motifs (represented in stick), showing an unusual pose of Y + 3 in atypical AP2μ-binding motif (*gold*) compared to the pose of F + 3 in the classical YRHF motif (*cyan*); electron density maps are shown in Supplementary Fig. 9. SuperTOP assays (as in Fig. 6) of wt and mutant LRP6 in (**f, g**) transiently or (**h**) stably transfected LRP5/6 DKO cells +/− Wnt3a; *open circles*, individual values obtained from at least three independent experiments are relative to the mean value of three samples from Wnt-stimulated parental HEK293T cells (wt, set to 100%); statistical significance was determined by one-way ANOVA with multiple comparisons, comparing each condition to (**f**) the wt LRP6 control, (**g**) all other conditions or (**g**) to Wnt-stimulated LRP6 wt; ****, $p < 0.0001$; ns, not statistically significant ($p = 0.9982$); *error bars*, ±SEM.

classical YxxΦ motif[62]. The affinity is twice as high ($K_D$ 8.0 ± 0.5 μM) between AP2μ and an extended peptide that includes YRPY (SYR-PYSYRHFA; Fig. 7b). In support of this, FLAG-AP2μ coIPs weakly but consistently with wt LRP6-GFP upon co-expression in cells (Fig. 7c) whereas coIP was undetectable for LRP6 mutants bearing alanine substitutions in either of the two YxxΦ (YxxY > A, YxxF > A) or in both motifs (YYYF > A; Fig. 7a, c).

Next, we co-crystallized the SYRHFA and SYRPYS peptides with purified AP2μ YxxΦ-binding domain to examine their binding mode. We thus determined co-crystal structures of each complex (at 2.8–2.9 Å resolution; Supplementary Table 1). As expected, SYRHFA binds to AP2μ in the typical 'two-pins plug in a socket' fashion seen in previous co-crystal structures[65] (Fig. 7d). By contrast, SYRPYS adopts a distinct pose, with its downstream Y lying flat across the Φ pocket and its apical hydroxyl group pointing towards the backbone of AP2μ R402 (Fig. 7d, e; Supplementary Fig. 9). This is an unusual pose that has not been observed previously for any AP2μ-binding motif (including the non-canonical YxxGΦ motif[68]). We were unable to test whether this pose might change in the presence of additional residues upstream (which can affect binding[69]) since extended peptides containing a Y at P-2 are as insoluble as SYRPYS. We conclude that each of the tandem Y-motifs is a bona fide AP2μ ligand. The duplication of the YxxΦ motifs in mammalian LRP6 might serve to increase its binding avidity for AP2.

To test whether the tandem Y-motifs are required for signaling by LRP6, we expressed single-motif (YxxY > A, YxxF > A) and double-motif (YYYF > A) mutants in transiently transfected LRP5/6 DKO cells[70] (together with the LRP6 chaperone Mesd, to promote transport to the plasma membrane[9]) and conducted SuperTOP assays +/− Wnt, using the LRP6m10 mutant (see Fig. 1a) as a benchmark. As expected from previous work[25,32,56], LRP6m10 shows no activity whatsoever (Fig. 7f), similarly to ΔB whose Wnt-induced activity is barely detectable above background (Fig. 7f; Supplementary Fig. 10a). The double-mutant YYYF > A signals ~10x less than its wt control upon Wnt stimulation, while the two single-motif mutants retain moderate signaling activity (Fig. 7f). This is consistent with earlier results based on LRP6 overexpression in LRP6-depleted cells[40] although the disabling effect of YYYF > A is clearly stronger in the absence of endogenous LRP6 and LRP5 (Fig. 7f), likely because there is no contribution from residual endogenous LRP6. Therefore, the tandem Y-binding motifs in LRP6 are essential for efficient Wnt signaling activity.

We next asked whether a heterologous duplicated AP2-binding motif (from the transferrin receptor, termed 2xTF) could restore signaling if inserted into LRP6 ΔB, or into a smaller internal deletion termed ΔB2 (Fig. 7a). Indeed, wt 2xTF but not its YFYF > A mutant version restores considerable signaling activity in ΔB2 whose own activity is much reduced, like that of YYYF > A (Fig. 7g). This corroborates the functional importance of the tandem Y-motif in Wnt signal transduction.

Previous functional studies were typically based on monitoring signaling by overexpressed N-terminal deletions of LRP6 (that lack their Wnt-binding domains) in the presence of endogenous

LRP6[25,32,33,36,38,57,71] which contributes to the signaling output. Indeed, overexpression of full-length wt LRP6 on its own can produce some Wnt-independent signaling activity[7,57]. To test the signaling function of the YYYF > A mutant in a more physiological setting, we developed a complementation assay based on stable re-expression of LRP6-GFP in LRP5/6 DKO cells[70], as previously devised for DVL2[72]. Of note, cells lacking LRP6 alone (LRP6 KO) cannot respond to Wnt (except for the Wnt-induced destabilization of Axin), whereas LRP5 KO cells are as Wnt responsive as their parental controls (Supplementary Fig. 10b), consistent with the notion that LRP6 is the primary Wnt co-receptor[71]. If LRP6-GFP is expressed in LRP5/6 DKO cells, the Super-TOP activity of the exogenous protein is strictly Wnt-dependent, even though the levels of LRP6-GFP are higher than those of endogenous LRP6 (Fig. 7h). Likewise, activated β-catenin (ABC) and Axin destabilization are only observed after Wnt stimulation (Fig. 7h; Supplementary Fig. 10c). Therefore, LRP6-GFP restores multiple Wnt responses in LRP5/6 DKO cells.

In contrast to wt LRP6-GFP, YYYF > A shows only residual Wnt-inducible SuperTOP activity and weak accumulation of ABC, whereas the single-motif mutants remain moderately Wnt responsive (Fig. 7h). The loss of Wnt responses of YYYF > A is even more compelling in the light of its expression levels that are higher than those of wt LRP6-GFP and single-motif mutants (Fig. 7h), likely owing to reduced endocytic targeting and lysosomal degradation[57]. We conclude that the tandem Y-motifs are critical for Wnt-dependent signaling of LRP6 in a complementation assay based on LRP5/6 DKO cells.

## Proximity between Wnt signalosome and endocytosis components

As mentioned in the Introduction, LRP6 is targeted by AP2 to large clathrin-coated structures upon Wnt stimulation[40]. To examine whether components of these clathrin structures are associated with LRP6, we used BioID proximity-labeling to monitor the proteome associated with its ctail in Wnt-stimulated cells, by tagging wt and YYYF > A mutant LRP6 with BirA* (a promiscuous version of the biotin ligase BirA[73]). We used a tetracycline-controlled transcriptional activation system based on T-REx-293 cells[74] to control the expression levels of the baits, and we generated stable cell lines bearing wt or YYYF > A mutant baits integrated at the same genomic locus. Despite being expressed at significantly higher levels than endogenous LRP6, we found that bait expression is comparable to that of LRP6-GFP in stably transfected LRP5/6 DKO cells whose signaling activity is strictly Wnt-dependent (Supplementary Fig. 10c). Furthermore, the stabilization of β-catenin is Wnt-dependent in cells expressing tetracycline-induced wt LRP6-BirA* whereas β-catenin is not stabilized in Wnt-stimulated cells expressing YYYF > A-BirA* (Fig. 8a), indicating that the mutant bait is inactive in transducing the Wnt signal.

Next, we labeled cells with 50 mM biotin for 12 h in Wnt3a-conditioned medium, and prepared lysates for one-step biotin-avidin affinity purification and subsequent analysis by LC-MS/MS mass spectrometry. We consistently identified several Wnt signaling components with the wt bait, with GSK3β and GSK3α being amongst the

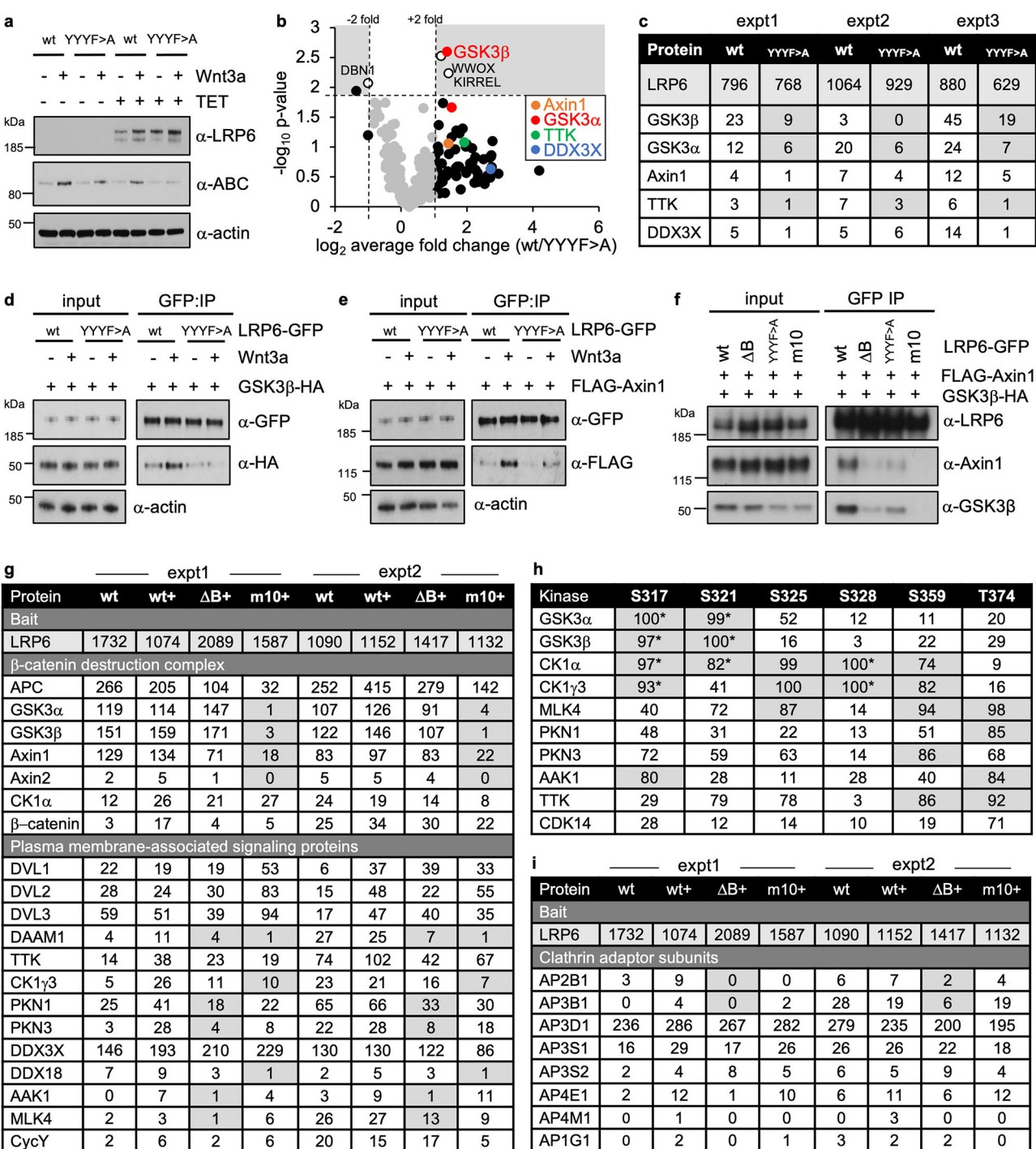

**Fig. 8 | Proximity between signalosome and endocytosis components.**
**a** Western blot of lysates from FlpIn T-Rex HEK293 cells +/− Wnt3a before or after 24 h of tetracycline (TET) induction of wt or YYYF > A mutant BioID baits (representative of three independent experiments), probed with antibodies (α) as indicated on the right; ABC, activated β-catenin. **b** Volcano plot showing YYYF-dependent hits identified by BioID (*black*, >2x compared to wt); *Y axis*, statistical significance of values obtained from three independent experiments was determined by Student's *t*-test comparing each identified protein to the LRP6-BirA* bait (in cases where no peptide counts were identified in a sample, the value was replaced with 1 for calculation of ratios); selected hits are labeled (*colored*, related to Wnt signaling; *black*, unrelated to Wnt signaling). **c** Total unweighted spectral counts (>95% probability) of LRP6-BirA* bait (*black*) and selected hits from 3 biological replicates (expt1-3); *gray underlay*, hits reduced >2x in each experiment. **d–f**

CoIP assays as in Fig. 1 (representative for three independent experiments), following co-expression of wt or mutant LRP6-GFP with FLAG-Axin and/or GSK3β-HA as indicated. **g** ΔB- or LRP6m10-sensitive hits identified in two independent TurboID experiments (expt1 and 2) with (+) or without 2 h of Wnt3a stimulation; shown are total unweighted spectral counts (>95% probability) of LRP6-BirA* bait (*top*) and selected Wnt signaling-related hits (*underneath*); *gray underlay*, hits that are reduced >2x compared to wt in both experiments by ΔB (deleting YxxY, YxxΦ) or LRP6m10 (see also Supplementary Fig. 11). **h** Key phospho-acceptor sites in Axin LIRup (S317-S328), LIRα (S359) or PRTxR (T374) and cognate kinases shown in (**g**) with Kinase Library (KL) scores >80 (*gray underlay*); *, primed phospho-acceptor sites. (**i**) Subunits of clathrin adaptor complexes identified in two independent TurboID experiments shown in (**g**); *gray underlay*, hits reduced >2x in both experiments.

top hits, in addition to proteins without known functions in Wnt signaling (Fig. 8b; Source Datafile 1). Importantly, when comparing these hits with those identified by the YYYF > A mutant bait, GSK3β was most affected, being >2x reduced in each of 3 independent experiments (Fig. 8b, c). Axin1 was also consistently reduced within the proteome identified by YYYF > A (Fig. 8b, c; Source Datafile 1). We validated these results by coIP after co-expression of LRP6-GFP with FLAG-Axin or GSK3-HA in LRP5/6 DKO cells, which confirmed that significantly less GSK3-HA (Fig. 8d) or FLAG-Axin (Fig. 8e) coIP with YYYF > A compared to wt LRP6-GFP, consistent with our BioID results (Fig. 8b, c) and Wnt signaling assays (Fig. 7f–h). Intriguingly, we also identified threonine tyrosine kinase (TTK, aka MPS1) amongst the YYYF-sensitive hits (Fig. 8b, c): TTK is a kinetochore-associated kinase whose autophosphorylation is required for Dishevelled-dependent spindle-checkpoint activation[75]. Of note, TTK has multiple conserved YxxΦ motifs in its unstructured N- and C-terminal tails and is therefore a plausible substrate for AP2.

Since these BioID experiments did not identify components of the endosomal machinery, we decided to repeat them with a more efficient labeling approach called TurboID[76] which had in the meantime been developed. This allowed us to monitor the LRP6-associated proteome after a short pulse of Wnt stimulation, which avoided the onset of feedback mechanisms (e.g. those based on Naked/NKD[77]) that could affect the composition of the Wnt signalosome. We also generated an unstimulated control sample although, like many other mammalian cells, HEK293 cells experience autocrine Wnt signaling[78], and so the effect of additional Wnt stimulation can be small. As mutant LRP6 baits, we used ΔB (removing both AP2 binding sites and other potentially functional motifs within the B element), and LRP6m10 (retaining AP2 binding but abolishing binding of GSK3 and Axin[32]) for comparison. As expected, ΔB acts similarly to YYYF > A in reducing coIP of FLAG-Axin1 and GSK3-HA with LRP6-GFP (Fig. 8f).

Next, we used these TurboID baits to label Wnt-stimulated T-REx-293 cells for 2 h. We found several Wnt degradasome and signalosome components including the Dishevelled-associated factors DAAM1[79] and DDX3[80] and its DDX18 paralog, with GSK3 and Axin paralogs showing consistently LRP6m10-sensitive labeling, as expected[32] (Fig. 8g; Supplementary Fig. 11). We also found TTK amongst the ΔB- and LRP6m10-sensitive hits, and membrane-associated kinases AAK1[70], protein kinase N1 (PKN1)[81], PKN3[82] and CK1γ3[83] all of which regulate Wnt signaling. Our LRP6-proximal hits further include CDK14 which initiates Wnt signaling following activation by membrane-associated cyclin Y[34] (see Introduction), and mitogen-activated kinase 4 (MLK4), a tumor suppressor in β-catenin-driven colon cancers[84] (Fig. 8g). Each of these kinases is predicted to phosphorylate Axin LIR at one or more of its functionally relevant serine or threonine residues with high probability (KL scores >80; Fig. 8h). Cyclin Y-activated CDK14 also phosphorylates S1490 within PPPSPAT of LRP6[34], consistent with its high KL score (Supplementary Fig. 12).

Finally, we identified multiple subunits of different clathrin adaptor complexes (Fig. 8i), including the large β subunit of AP2 (and its µ subunit in expt 2; Source Datafile 1), large and small subunits of AP3 (which recognizes YxxΦ similarly to AP2[85] and mediates transport of YxxΦ-bearing cargo from early endosomes to lysosomes[86,87]), and the large ε subunit of AP4 (which recognizes YxxΦE motifs and mediates transport of ATG9-containing autophagy vesicles from the trans-Golgi network to the cell periphery[87,88]). We also found Gadkin which appears to promote AP1-dependent recycling of trans-Golgi-derived vesicles to the cell periphery following their AP2-dependent endocytosis[89,90]. Of note, the high spectral counts of the AP3 subunits suggest that a high fraction of the overexpressed LRP6 bait is en route to lysosomal degradation. In summary, these TurboID experiments corroborate the notion that the Wnt signalosome is in close proximity to clathrin adaptor subunits and to AAK1 which associates with AP2 to phosphorylate its cargo-binding µ subunit.

## Discussion

In this study, we have defined the elements and motifs that mediate the mutual interactions between signalosome components, measured their mutual affinities and demonstrated the function of these interactions in physiological cell-based assays. To our surprise, we discovered that Axin and GSK3 bind to the same region in the LRP6 ctail – Axin with low and GSK3 with barely measurable affinity – yet their docking at LRP6 in vivo requires their cooperation as it depends on their mutual binding to each other. We also found that the AP2-mediated targeting of LRP6 to large clathrin-coated structures[9,40] is critical for efficient Wnt responses. Below, we propose two mechanistic explanations how the AP2-dependent targeting of LRP6 to these clathrin locales might drive signalosome assembly and impact on canonical Wnt signal transduction.

The first follows an idea initially proposed by He and colleagues[38] that the clustering of LRP6 may amplify the recruitment of Axin and GSK3 to LRP6. Indeed, Wnt-induced co-clustering of LRP6 with large clathrin-coated structures was observed subsequently, and components of these structures were co-purified with LRP6 biochemically[40]. Based on this, we previously proposed that the AP2-driven clustering of LRP6 could be an avidity-boosting device that facilitates mutual assembly of signalosome components[11]. This attractive concept was consistent with a large body of evidence from a variety of models that implicated endocytic components in Wnt signaling (for a systematic review, see ref. 91). Further corroboration came with the recent discovery of a negative feedback loop mediated by AAK1, which down-regulates signaling after prolonged Wnt stimulation by promoting clathrin-dependent internalization of LRP6[70]. Notwithstanding this, Rim et al. argued against a role of endocytosis in Wnt signaling since they observed normal Wnt responses after depletion of endocytosis components[92]. However, their evidence was inconclusive because of the intrinsic difficulties in generating null mutants: deletion of endocytic components is either early embryonic[93] or cell-lethal in the case of single-paralog genes such as that encoding AP2µ (judging by the fact that we were unable to isolate cell lines bearing CRISPR-generated deletions of the gene encoding AP2µ), or well tolerated in cases such as the large AP2 α subunit which is encoded by two closely related paralogous genes that are likely to be functionally redundant[86,87]. Furthermore, knockdown of endocytic components (used as an alternative method by Rim et al.[92]) is inadequate since residual levels of these proteins generally suffice to provide normal function[62]. Therefore, the concept of AP2-dependent clustering of LRP6[11,40,91] remains valid and is corroborated by our current data (Figs. 7, 8) as a plausible mechanism underpinning robust Wnt signalosome assembly and signal transduction, as depicted in our model (Fig. 9).

A key premise for this model is the local activation of PI4K and PI5K by Dishevelled upon its binding to Wnt-occupied FZD, which leads to a PIP2-enriched patch surrounding the Wnt receptor complex. This triggers the recruitment of AP2 (Fig. 9) and its consequent activation towards LRP6, resulting in the clustering of multiple receptor complexes in large clathrin-coated structures, as previously reported[40]. These may reflect clathrin-coated plaques or similar clathrin-coated structures that are larger and more stable than the canonical clathrin-coated pits (which are transient structures that initiate rapid endocytosis of membrane protein cargo)[94,95]. How these large clathrin locales form and how long they persist in the plasma membrane is largely unknown, but they appear to have various roles in cell adhesion or signaling[96]. Based on our results (Figs. 7, 8i) and previously published evidence[91], we propose that the AP2-driven clustering of receptor complexes might allow a single Axin-GSK3 complex to bind simultaneously to their cognate PPPSPAT elements in adjacent LRP6 ctails through 'trans' cooperativity. This would enable this complex to overcome the formidable affinity hurdle that opposes its recruitment to LRP6 in the absence of Wnt. Evidently, this mechanism would synergize with the Dishevelled-driven co-polymerization of

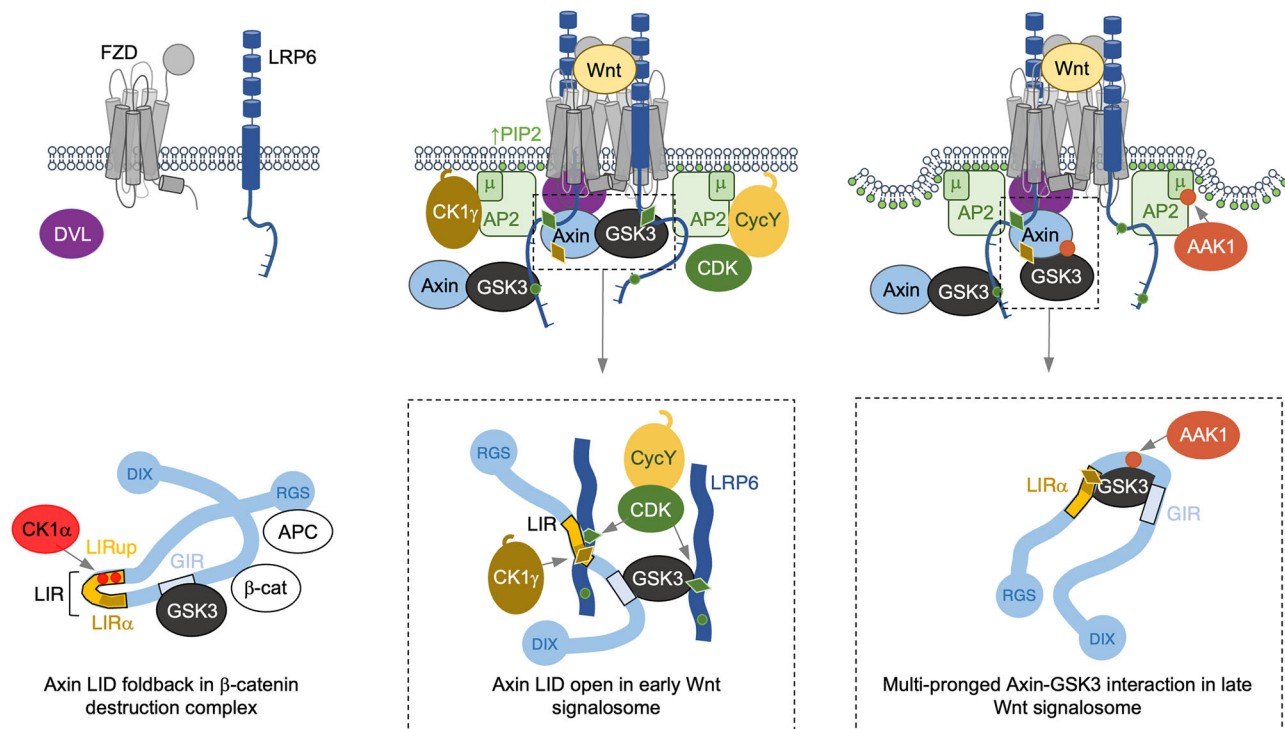

**Fig. 9 | Model of Wnt signalosome assembly.** Signalosome complexes during different phases of Wnt signaling and corresponding interactions of the Axin scaffold with itself or its ligands (*underneath*); also shown are key phosphorylations (*diamonds*, priming phosphorylations; *dots*, effector phosphorylations) imparted by LRP6-associated kinases (Fig. 8h; see also text), as indicated by matching colors and arrows (*top*, arrows for Axin-GSK3 complexes omitted for clarity). (*Left*) Without Wnt, the β-catenin destruction complex is assembled by APC and Axin, which binds to GSK3 (through GIR, *pale blue*) and, jointly with CK1α, earmarks β-catenin for proteasomal degradation. CK1α also phosphorylates Axin LIRup (*red dots*) to promote its binding to LIRα (*ochre*) and formation of an intramolecular foldback that opposes binding between LIR (*yellow*) and LRP6; unphosphorylated PPPSPxS motifs are indicated by short lines. (*Middle*) Upon Wnt-dependent coupling of FZD and LRP6, a patch of high PIP2 (*green dots*) is generated by Dishevelled in the plasma membrane near the receptor complex where PIP2 activates the AP2 clathrin adaptor towards LRP6, thus promoting clustering of LRP6 and clathrin

coating of the PIP2 locale. Hence, LRP6 encounters cyclin Y-activated CDK following its AP2-dependent co-targeting to clathrin locales and is thus phosphorylated at S1490 (*green diamonds*) which enables Axin and GSK3 to dock at LRP6. AP2-driven clustering of LRP6 allows simultaneous binding of Axin and GSK3 to adjacent ctails and exposes Axin to co-targeted CK1γ which phosphorylates LIRα at S359 (*brown diamonds*) to prime Axin for a subsequent multi-pronged interaction with GSK3; phosphorylation of S359 may also promote Axin binding to LRP6, which would explain the agonist role of CK1 during Wnt signaling (see text); *dark green dots*, phosphorylations of PPPSPxS that render these motifs competitive inhibitors of GSK3, which promotes transduction of the Wnt signal to β-catenin. (*Right*) AAK1 phosphorylates AP2 (*terracotta dot*) at peripheral sites of clathrin locales to initiate localized endocytosis while simultaneously phosphorylating Axin PRTxR (*terracotta dot*) to promote its multi-pronged interaction with GSK3. This causes detachment of GSK3 from the receptor complex and thus presages termination of Wnt signaling.

Axin, which increases the local concentration of Axin at the receptor complex (see Introduction). Together, the two mechanisms could ensure that the docking of Axin and GSK3 at LRP6 depends on its Wnt-induced coupling with FZD.

Recently, Yue et al. (2025) discovered that the extracellular part of the Wnt3a:FZD8:LRP6 receptor complex comprises two dimerizing molecules of Wnt3a, each of which binds to its cognate domains in one LRP6 and two FZD molecules, thus exhibiting a 2:4:2 stoichiometry (Wenqing Xu, personal communication). This Wnt-dependent coupling of two LRP6 molecules could juxtapose their ctails and thus enable simultaneous binding of Axin and GSK3. In this case, AP2-dependent clustering of LRP6 would not be strictly required for recruitment of the Axin-GSK3 complex. However, the Wnt dimerization interface is only partially conserved through evolution: for example, Wnt8 paralogs bear an arginine in a key position of their dimerization interface that may repel dimerization (Yue et al., 2025). Therefore, in the case of monomeric Wnts, the need for trans cooperativity remains a plausible explanation for the reliance of signalosome assembly on AP2-driven clustering of LRP6 in large clathrin locales.

As an alternative or additional function, we propose that the targeting of LRP6 to AP2-containing structures assists assembly of a

signalosome by exposing its components to kinases that are co-targeted to these locales by AP2, and that these kinases could remodel the structure or activity of the signalosome by phosphorylating its components (Fig. 9). Plausible candidates are S/T or tyrosine kinases whose association with LRP6 depends on YYYF or the B element (which contains YYYF). While these YYYF-dependent hits do not comprise any tyrosine kinases[97] (Supplementary Fig. 11), they include several S/T kinases: the top-scoring ones include TTK, AAK1, CK1γ3, PKN1 and PKN3 (Fig. 8g), which have membrane-proximal roles in various Wnt signal transduction branches[36,70,75,81,82].

One plausible target for these kinases is Axin LIR, given its regulatory roles in modulating Wnt signaling (Fig. 6). The sequences from LIRup to GIR contain only two S/T residues (S359 and T374). These achieve high KL scores for PKN3 or CK1γ3 (S359), PKN1 or AAK1 (T374), or TTK or MLK4 (both; Fig. 8h) and are therefore likely phospho-acceptors for these kinases. Furthermore, structural modeling of phosphorylated sequences by AF3[44] predicts that the phosphorylation of T374 within the PRTxR motif may enable Axin to form a multi-pronged complex with GSK3, with PRpTxR binding across its catalytic groove and pT374 tucking into its priming pocket (Supplementary Fig. 13, *left*) similarly to phosphorylated PPPSPxS motifs[27] (Fig. 5h). However, the tucking of pT374 into the priming site appears

contingent on prior phosphorylation of S359 (Supplementary Fig. 13, *middle*) which Axin may acquire on its exposure to CK1γ within the clathrin locales surrounding the Wnt receptor complex (Fig. 9). Therefore, the dual phosphorylation of S359 and T374 could convert Axin from an agonist co-factor synergizing with GSK3 in its docking at LRP6 into a competitive inhibitor of GSK3 that causes its dissociation from LRP6. Future work will be required to determine the timing of these two key phosphorylations within LIRα and PRTxR and the identity of the kinases that impart them on Axin to promote its multi-pronged interaction with GSK.

Functional validation for the PRTxR motif was provided by the P372A mutant cells whose Wnt-dependent hyperactivity is consistent with an antagonist role of the multi-pronged Axin-GSK3 complex (Fig. 6d). The normal levels of basal β-catenin activity in the P372A mutant cells imply that this complex attenuates signal transduction subsequent to, rather than before, Wnt stimulation (Fig. 9). Of note, the threonine of this motif (T374) could be phosphorylated by AAK1 or PKN1, given their high KL scores for T374 (Fig. 8h). These kinases are Wnt antagonists that function during advanced stages of signaling[70] when they promote endocytosis of LRP6[81]. Furthermore, while PKN1 has not been studied in the context of clathrin-mediated endocytosis, AAK1 is known to phosphorylate T156 within the μ cargo-binding subunit of AP2, which enhances its affinity to YxxΦ and ensures efficient targeting of YxxΦ cargo at specific sites earmarked for subsequent endocytosis[98,99]. At the same time, AAK1 may phosphorylate Axin at T374 which could strengthen its interaction with GSK3 (see above) and enable the two proteins to form a multi-pronged complex. A corollary is that the phosphorylation of T374 by AAK1 (or other cognate kinases such as TTK, PKN1 or MLK4) may presage termination of signaling by triggering internalization of the Wnt receptor-associated signalosome (Fig. 9). In support of this, the Wnt-driven phosphorylation of AP2μ by AAK1 depends not only on LRP6, Dishevelled and PIP2, but also occurs in a time-delayed fashion following the initiation of Wnt signalosome assembly at LRP6[70]. More work is required to assess whether, and when, Axin might encounter other LRP6-proximal kinases (Fig. 8h) that could affect its function in Wnt signalosome assembly or function. Furthermore, the factor(s) causing the time delay between the initiation of this process and the AP2-dependent internalization of the Wnt signalosome remain to be determined.

Another plausible target for the LRP6-proximal kinases identified by TurboID (Fig. 8h; Supplementary Fig. 11) is LIRup, given its ability to form a phospho-dependent foldback with LIRα (Fig. 4). The kinases that are likely to phosphorylate LIRup within degradasome-associated Axin are CK1α and GSK3: CK1α phosphorylates S325 to prime phosphorylation of S328 by itself, and of S317 and S321 by GSK3 (Fig. 4a). These phosphorylations protect Axin from proteasomal degradation, thus enabling it to function effectively within the degradasome[49,50]. This mechanism appears to operate in parallel to the phospho-dependent molecular foldback loop within Axin (Fig. 4g) which also antagonizes Wnt signaling (Fig. 6c), apparently by lowering its accessibility to LRP6 (Fig. 4b, c). This may guard against fortuitous binding of Axin to LRP6 in the absence of a Wnt signal. As an aside, a distinct foldback within the C-terminal region of Axin requires local dephosphorylation following Wnt stimulation[30], suggesting that the two foldbacks form during different phases of Wnt signaling and achieve distinct outcomes. Indeed, Axin may oscillate between different conformations pending on phosphorylation of distinct regulatory sites.

Do some CK1 isoforms also function during Wnt stimulation? We believe this to be the case, given the unique ability of CK1γ amongst CK1 family members to promote Wnt signaling[36,83]. Of note, the γ1−3 isoforms of this family albeit not α, β, δ and ε (some of which act redundantly with CK1γ3 under certain conditions[83]) can associate with the plasma membrane through palmitoylation of their C-termini[36] (possibly by ZDHHC18, an LRP6-proximal palmitoyl transferase;

Supplementary Fig. 11). Furthermore, the disordered ctail of CK1γ3 contains a YxxΦ motif (YDWI) whose Φ residue is mutated to alanine in γ1 and γ2 isoforms. This motif could target palmitoylated CK1γ3 to the high-AP2 patch surrounding the Wnt-occupied receptor complex, where it may encounter Axin upon its initial binding to LRP6 (Fig. 9). CK1γ3 may thus phosphorylate S359 in LIRα (Fig. 8h) to prime it for subsequent phosphorylation at T384 by AAK1 (Fig. 8h); together, these phosphorylations could promote the multi-pronged interaction between Axin and GSK3 and, ultimately, termination of Wnt signaling (Fig. 9, *right*). However, it is also conceivable, given the position of S359 within the α-helical element of LIR, that its CK1γ-mediated phosphorylation may promote binding of Axin to LRP6 (Fig. 9, *middle*), which would explain the agonist role of CK1γ in Wnt signaling[36,37,83]. If so, the CK1γ-mediated phosphorylation of Axin S359 alone would initiate or, in conjunction with the phosphorylation of Axin T384, downregulate Wnt signaling – providing a context-dependent phosphorylation switch that would be interesting to test in future studies.

An alternative explanation for the agonist role of CK1 in Wnt signaling may be provided by the CK1γ-mediated phosphorylation of S325 and S328 in Axin LIRup (Fig. 8h) whose alanine substitutions (in $S_4 > A$ mutant cells) reduce the Wnt response (Fig. 6b). Dual phosphorylation of S325 and S328 could allow LIRup to bind to GSK3 as a pseudo-substrate (Supplementary Fig. 13, *right*) similarly to PPPpSPxT (Fig. 5i, j), thus blocking access of other substrates to its catalytic pocket. Therefore, this dual phosphorylation of Axin may enable it to inhibit GSK3 directly, bypassing the need for LRP6. This would explain how polymerized Dishevelled promotes Wnt-independent β-catenin signaling in cell-based assays[100] and intestinal tumorigenesis[101], and how locally clustered Dishevelled drives physiological β-catenin responses in early Xenopus embryos[102]. Future studies are needed to assess whether, and under which circumstances, Axin may function as a direct competitive inhibitor of GSK3.

Finally, LRP6 itself could be a target for any of the kinases that associate with its ctail in a YYYF-dependent fashion, for example S1490 whose phosphorylation is pivotal for the docking of Axin and GSK3 at its ctail[25,32,33]. GSK3 is widely thought to be kinase that catalyzes this phosphorylation, based on evidence that GSK3 can phosphorylate S1490 in vitro and in vivo[26,32,38]. Consistent with this are the top KL scores for the two GSK3 isoforms (Supplementary Fig. 12). However, given that the affinity of GSK3 for unphosphorylated PPPSPAT element is barely measurable (Fig. 3b), we consider it more likely that the physiologically relevant kinase for S1490 is a CDK paralog[34]. In support of this, amongst the 59 kinases predicted to phosphorylate S1490 with high probability are 16 CDK paralogs, with CDK8 and CDK19 immediately below GSK3 (Supplementary Fig. 12). As mentioned, CDKs rely on activation by cyclins, e.g. by atypical cyclin Y which associates with the plasma membrane through myristoylation of its unstructured ctail[35]. Furthermore, its ctail bears a YxxYYxxΦ motif (YAKYYFDL), constituting two plausible AP2-binding sites that are conserved throughout the animal kingdom. Therefore, cyclin Y is likely targeted to the PIP2 patch surrounding Wnt-occupied receptor complexes, to prime cells for future canonical Wnt signaling by recruiting a CDK and activating it towards S1490 (Fig. 9).

However, cyclin Y-dependent CDKs may not be able to prime Wnt responses in post-mitotic cells such as neurons or early embryos that have not undergone a recent S2/M transition. In these cells, plausible candidates for phosphorylating S1490 include Nemo-like kinase (NLK), an ancient MAPK-related kinase bearing a $YxxΦx_{10}YxxY$ motif and priming Wnt responses in early embryos[103,104], or JNK1-3 with Wnt-priming functions in embryonic and neural stem cells[105–107]. Finally, as mentioned above, some cellular systems have evolved β-catenin signaling mechanisms that are primarily driven by clustered Dishevelled rather than FZD and Wnts[102].

Another possible kinase target within the LRP6 ctail are the serines preceding YxxY (S1516) or YxxΦ (S1521). Remarkably, these serines are

predicted to be phospho-acceptor sites for numerous kinases, including PKN1 (S1516), MTOR (S1516) and CK1ε (S1521) with KL scores in the top two percentiles. Phosphorylation of these serines is likely to hinder their binding to AP2μ and may thus attenuate Wnt signal transduction (Fig. 7). Of the top-scoring kinases for S1516 and S1521, PKN1 associates with LRP6 in a ΔB-sensitive fashion (Fig. 8g) and antagonizes Wnt signaling[81]. Therefore, LRP6-associated kinases such as PKN1 could attenuate Wnt signaling by antagonizing LRP6 binding to AP2μ, and hence its AP2-dependent clustering in the clathrin locales surrounding Wnt-occupied receptor complexes, which would be interesting to test in future.

In summary, our study has led us to propose a model of Wnt signalosome assembly at the LRP6 ctail that envisages a pivotal role of its AP2-dependent clustering in clathrin locales (Fig. 9). While further work is required to test individual aspects of this model, our results have highlighted that the conformational flexibility of Axin and its encounter with LRP6-proximal kinase in clathrin locales may have pivotal roles in the assembly and modulation of the Wnt signalosome.

## Methods

### Cell cultures and lines

HEK293T (ATCC, Cat#CRL-3216), HEK293T LRP5/6 DKO[70] and FlpIn T-REx (Thermo Fisher Scientific, Cat#R78007) cells were cultured in 6-well culture dishes in DMEM+GlutaMax (Gibco, Cat#11594446), supplemented with 10% fetal bovine serum (FBS; Gibco, Cat#11594446) plus 1% penicillin/streptomycin (Sigma) at 37 °C in a humidified atmosphere with 5% $CO_2$, and regularly screened for mycoplasma. To generate LRP6-expressing HEK293T cell lines for LRP6 complementation tests, LRP6-GFP was inserted into the pBABE vector and stably re-expressed in LRP5/6 DKO cells[70], as previously described for DVL2 complementation tests[72].

### Generation of plasmids

Sequences for in vitro and cell-based assays were amplified by polymerase chain reaction (PCR) from either plasmid templates or synthetic genes (gBlocks, IDT), cloned into mammalian or bacterial expression vectors by restriction-free cloning using Gibson Assembly Master Mix (NEB, Cat#E2611L). Point mutations and deletions were generated by Quikchange, using KOD Hot Start DNA polymerase (Merck Millipore, Cat#71086) or Q5 polymerase (NEB, Cat#M0491L). All plasmids were verified by DNA sequencing.

### coIP assays

HEK293T cells were seeded at ~70% confluency and transfected with a 1:3 ratio DNA:PEI (Polyethylenimine, Polysciences, Cat#23966), or with Lipofectamine3000 (Invitrogen, Cat#L3000008, according to manufacturer's instructions) mixture after cells had attached. For all coIPs, one well of a 6-well plate per coIP was used. For LRP6-GFP, FLAG-Axin1 and GSK3β-HA coIPs, optimal binding conditions were determined by titrating amounts and ratios of freshly prepared transfected plasmid DNA (Supplementary Fig. 1). This revealed that high levels of FLAG-Axin1 expression (seen in cells transfected with >75 ng per well) severely reduced coIP with LRP6-GFP. In cases of LRP6-GFP expression, its chaperone Mesd was co-expressed[9], to promote transport to the plasma membrane. Cells were lysed ~24 h post-transfection in 20 mM Tris pH 7.4, 200 mM NaCl, 10% glycerol, PhosSTOP (Sigma, Cat#04906837001), EDTA-free protease inhibitor cocktail (Roche, Cat#04693159001) and 0.1% Triton-X. Lysates were sonicated and cleared by centrifugation (16,100 g, 10 min), and supernatants were incubated with GFP-trap (Chromotek, RRID: AB_2631357) for at least 90 min at 4 °C on an over-head tumbler. Immunoprecipitates were washed in lysis buffer and eluted by boiling in 4x lithium dodecyl sulphate (LDS) sample buffer (Invitrogen, Cat#NP0007) for 10 minutes. Input lysates (1% of total) and coIP eluates (20% of total) were separated by SDS polyacrylamide gel electrophoresis (SDS-PAGE),

blotted onto polyvinylidene difluoride membranes, checked for equal loading by Ponceau staining and processed for Western blotting using the following primary antibodies: α-GFP and α-FLAG (RRIDs: AB_439690; AB_439687, Sigma), α-HA and α-actin (RRIDs: AB_307019; AB_2305186, Abcam), α-active β-catenin, α-GSK3β, α-pS1490 LRP6, α-LRP6 and α-Axin1 (RRIDs:AB_11127203; AB_490890; AB_2139327; AB_10831525; AB_2274550, Cell Signaling Technologies). Primary and secondary antibodies were diluted 1:1000–5000 in phosphate-buffered saline (PBS), 0.01% Triton-X and 5% milk powder. Blots were washed with PBS containing 0.01% Triton-X and developed with ECL Western Blotting Detection Reagent.

### Protein expression and purification

Recombinant His6x-Lip-Axin (Axin$_{308-426}$, A3$_{Axin}$; Axin$_{308-366}$, A5$_{Axin}$, wt and mutant versions), His6x-Lip-L1$_{LRP6}$ (LRP6$_{1463-1538}$ bearing YYYF > A mutations to render it soluble in aqueous buffer) and His6x-Lip-L2$_{LRP6}$ (LRP6$_{1539-1613}$), GST-GSK3β (full-length wt or K85R mutant protein), His6x-Lip-YxxΦ peptides and His6x-AP2μ were purified from BL21-CodonPlus(DE3)-RIL cells (Agilent) E. coli bacterial strains. Bacteria were grown at 37 °C in LB media supplemented with appropriate antibiotic to OD$_{600}$ 0.6, then moved to 16–21 °C, followed by induction with 0.4 mM IPTG at OD$_{600}$ 0.8. Bacteria were harvested by centrifugation (8000 g, 30 min); cell pellets were shock-frozen in liquid nitrogen and stored at −80 °C until use. Harvested cells expressing the above proteins were resuspended in lysis buffer (25 mM Tris pH 8.0, 200 mM NaCl, 20 mM imidazole pH 8 and EDTA-free protease inhibitor cocktail; Roche) and lysed by sonication (Branson). Lysates were cleared by ultracentrifugation (140,000 g, 30 min at 4 °C) and incubated with Ni-NTA agarose (Qiagen, Cat#30210) and washed with lysis buffer. After extensive washing, samples were eluted with lysis buffer supplemented with 500 mM imidazole and loaded onto a HiLoad 26/600 Superdex 75 pg column (GE Healthcare) equilibrated in 25 mM sodium phosphate pH 6.7 and 150 mM NaCl. His6x-AP2μ were loaded onto a HiLoad 26/600 Superdex 200 pg column (GE Healthcare).

For the purification of GST-GSK3β, harvested cells were resuspended in lysis buffer (20 mM Tris pH 7.4, 300 mM NaCl, 5% glycerol, 0.01% Triton X-100, 5 mg/mL DNase I, 1 mM DTT and EDTA-free protease inhibitor cocktail; Roche) and lysed by sonication (Branson). After clarification by ultracentrifugation (140,000 g, 30 min at 4 °C), cleared lysate was incubated for 2 h with glutathione agarose (Pierce™) and washed extensively with wash buffer containing 20 mM Tris pH7.4, 300 mM NaCl, 5% glycerol and 1 mM DTT. GST was cleaved from GSK3β using 3 C protease (produced in-house) overnight at 4 °C. Eluted GSK3β was further purified by HiLoad 26/600 Superdex 200 pg column (GE Healthcare) equilibrated with 25 mM sodium phosphate pH 6.7 and 150 mM NaCl. Each step of the purification was done at 4 °C, and protein purity was assessed by SDS-PAGE.

### NMR

For NMR spectroscopy, His6x- Lip-L1$_{LRP6}$, His6x- Lip-L2$_{LRP6}$ and His6-Lip-Axin were expressed in M9 minimal medium supplemented with antibiotics, trace elements, 25 ml overnight culture and 2 g of $^{15}N$-$H_4Cl$ per litre of expression culture. Additionally, 0.4% glucose was added for $^{13}C$-$^{15}N$ double-labeling of samples. Cultures were grown and processed essentially as described above[13].C-$^{15}N$ double-labeled proteins were analyzed in 25 mM sodium phosphate pH 6.7, 150 mM NaCl buffer, 5% v/v $D_2O$. Spectra were recorded using Bruker Avance III spectrometers operating at 600, 700 or 800 MHz $^1H$ frequency, with 5 mm inverse-detect cryogenic probes and a sample temperature of 293 K (unless otherwise stated), using unmodified Bruker pulse programs. Backbone resonance assignments were obtained for separate $^{13}C$-$^{15}N$ double-labeled samples of 500 μM A3$_{Axin}$, 300–500 μM L1$_{LRP6}$ or 300 μM L2$_{LRP6}$. Assignments of non-proline resonances were obtained from semi-constant time 3D spectra; HNCO and HN(CA)CO for carbonyl carbons ($^{13}C'$), HNCA and CBCA(CO)NH for Cα- and Cβ-optimized HNCACB for

Cβ. Sequence-specific connectivities were confirmed using (H)N(CA)NH. The random coil indices (RCI) from TALOS-N[108] were used to determine that L1$_{LRP6}$ and A3$_{Axin}$ are intrinsically disordered. The flexible chains give strong signals in $^{13}$C'-detect NMR experiments, enabling assignment of proline resonances from (HCA)CO(NCA)NCO, (HCACO)N(CA)NCO, H(CA)NCO and H(CACO)NCO, each using an in-phase anti-phase scheme for virtual decoupling of $^{13}$Cα from $^{13}$C' in $t_3$. Frequencies were referenced according to the unified scale, with the $^1$H signal of internal dimethylsilapentane sulfonate (DSS) at 0.0 ppm. All spectra were processed with TopSpin version 3 (Bruker) and analyzed using NMRFAM-Sparky version 1.3[109]. Sequence specific connectivity was aided with the program MARS version 1.2[110].

Protein-protein interactions were inferred from peak height attenuation in 2D BEST-TROSY[111], for H$_N$-N correlation, or IPAP-(HACACO)NCO[112] spectra for CO-N correlation, using unmodified Bruker pulse programs. BEST-TROSY were acquired with 192 and 1024 points in $t_1$ and $t_2$, respectively, 16 scans per $t_1$ increment and a recycle delay of 0.7 s. CON were acquired for 150 μM $^{13}$C/$^{15}$N-labeled LRP6-L1 at 151 MHz $^{13}$C, with 128 and 1024 points in $t_1$ and $t_2$, respectively, 320 scans per $t_1$ increment and a recycle delay of 1.4 s. Peak heights ($I$) were fit in Sparky. Relative peak height attenuation was calculated from 100x($I_{ref}-I_{complex}$)/$I_{ref}$.

## ITC

Affinities between Lip-L1$_{LRP6}$ and GSK3, Lip-A3$_{Axin}$ or Lip-A5$_{Axin}$, between Axin Lip-LIRα and phosphorylated Axin 43PPP peptides (Axin$_{308-351}$ including pS317, pS321, pS325) or unphosphorylated 43mer peptides, or between His6x-AP2μ and His6x-Lip-YxxΦ peptides were determined by ITC at 25 °C with a Malvern Panalytical ITC200 instrument in 25 mM sodium phosphate pH 6.7 and 150 mM NaCl buffer or, in the case of His6x-Lip-YxxΦ peptides and His6x-AP2μ, in 50 mM Tris-HCl pH 7.4 and 300 mM NaCl buffer. All peptides were obtained from Biomatic. Synthetic 43mer or 43PPP peptide solution in the ITC cell was prepared by weight of lyophilized peptide material to 50 μM, and Axin Lip-LIRα was concentrated to 500 μM. His6x-Lip-YxxΦ peptides were concentrated to 2 mM (SYRHFA) or 1.74 mM (SYRPYSYRHFA), and the concentration of His6x-AP2μ was 114 and 100 μM, respectively, for the two His6x-Lip-YxxΦ peptides. Lip-DED$_4$ > A, Lip-RKR > A (based on Lip-A5$_{Axin}$) and purified GSK3 protein were concentrated to 50 μM, and Lip-L1$_{LRP6}$ to 500 μM. Titrations consisted of 19 injections of 2 μL preceded by a small 0.5 μL pre-injection that was not used during curve fitting. Experiments were performed at a reference power of 6 μcal/s and with injections at 180 s intervals with constant stirring at 750 rpm. All ITC binding data were corrected with the appropriate control heats of dilution and fitted using the 'one set of binding sites' model in MicroCal PEAQ-ITC analysis software (v1.41). Experiments were performed at least 3 times with different batches of purified proteins.

## Protein crystallization and data collection

Pure AP2μ was concentrated with a 20 kD MWCO Vivaspin 20 concentrator (Sartorius) to 10 mg/ml. Prior to crystallization, 3x excess of synthetic peptides (dissolved in DMSO at 5 mM) was added to the protein. Crystallization trials with multiple commercial crystallization kits were performed in 96-well sitting-drop vapor diffusion plates (Molecular Dimensions) at 18 °C and set up with a mosquito HTS robot (TTP Labtech). Drop ratios of 0.2 μL + 0.2 μL (protein solution + reservoir solution) were used for coarse and fine screening. Initial hits were obtained under multiple conditions and optimized subsequently. Data were collected from crystals harvested from 2.0 to 2.2 M NaCl, 0.4 M Na/K phosphate, 0.1 M MES pH 7.1, 15–20% glycerol. Crystals were directly flash frozen in liquid nitrogen. Diffraction data were collected at the Diamond Light Source (DLS, UK) on beamline I04-1. Data processing was performed with XIA2 DIALS and scaled using Aimless from CCP4 (Collaborative Computational Project, Number 4,

1994)[113]. Structures were solved by molecular replacement using a previously published AP2μ structure (PDB code 1BW8). Structure refinement was performed with REFMAC followed by manual examination and rebuilding of the refined coordinates in the program COOT[114]. Color figures were prepared with PyMOL (Schrödinger).

## AlphaFold predictions

Prediction of the structure of the Axin-GSK3 complex was performed via Google CoLab using AF2 with MMSeqs2 (https://colab.research.google.com/github/sokrypton/ColabFold/blob/main/AlphaFold2.ipynb) or AF3 (https://alphafoldserver.com/).

## Mass spectrometry

To generate BioID plasmids, coding sequences of LRP6 (and mutants thereof) were amplified from pCS2 LRP6-GFP by PCR and inserted directly upstream of BirA* in pcDNA5/FRT/TO using Gibson assembly[115]. To replace the BirA* tag with TurboID, the TurboID coding sequence was amplified from pUAS-V5-TurboID-NES (Addgene #116904) by PCR and inserted downstream of LRP6. To generate FlpIn T-REx stable cell lines, LRP6-BirA* and LRP6-TurboID pcDNA5/FRT/TO plasmids (wt and mutant) were co-transfected with pOG44 (Flp recombinase vector) and selected with 250 μg/ml hygromycin B (ThermoFisher, Cat#10687010). For each stably transfected cell line, LRP6 expression was induced with tetracycline (Sigma, Cat#T8032) for 24 h and biotin (Sigma, Cat#04906837001) labeling performed – 12 h for BirA* and 2 h for TurboID experiments, in each case in Wnt3a-conditioned medium (WCM), typically diluted 1:1 with fresh medium to avoid starvation of cells incubated in WCM for extended periods (of note, FlpIn T-Rex are HEK293 cells which experience substantial levels of autocrine Wnt signaling[78], hence the low levels of additional stimulation in the second TurboID experiment which was performed with undiluted WCM). TurboID samples were confirmed by Western blotting. A total of 1.4-2.1 ×10$^8$ adherent cells were grown to full confluence, washed once with phosphate-buffered saline, flash-frozen in liquid nitrogen, and stored at −80 °C for 1–20 days. BioID pull-downs were done using Streptavidin Dynabeads (Invitrogen, Cat#65001) as described[73], and protein was eluted from the beads by boiling for 15 min in LDS sample buffer (Invitrogen, Cat#NP0007). All samples were resolved on 4–12% Bis-Tris polyacrylamide gels, and gels were stained with Imperial Protein Stain (ThermoScientific, San Jose CA, USA). Gel slices (2–3 mm) were prepared for mass spectrometric analysis by manual in situ digestion with trypsin, and digests were analyzed by nano-scale capillary LC-MS/MS using an Ultimate U3000 HPLC (ThermoScientific Dionex, San Jose, USA). The analytical column outlet was directly interfaced via a nano-flow electrospray ionization source, with a hybrid dual pressure linear ion trap mass spectrometer (Orbitrap Velos, ThermoScientific, San Jose, USA). LC-MS/MS data were searched against a protein database (UniProt KB) using the Mascot search engine program (Matrix Science, London, UK) or, in the case of TurboID, with MaxQuant software (https://www.maxquant.org/). MS/MS data were validated using the Scaffold program (Proteome Software Inc., Portland OR, USA) and processed with R package. The raw mass spectrometry proteomics data have been deposited to the ProteomeXchange Consortium[116] via the PRIDE partner repository[117].

## CRISPR engineering

The CRISPR design tool CRISPOR (crispor.tefor.net) was used to design the following gRNAs targeting Axin1 (TCTCCCTGCGGTGCTGCTTA, GCTGCTTACGGATCCTGTAT, CTGCTCGCTGTCGTTGGCAC, GCAGCGTGTAAGTCCCGCCC) or Axin2 (CAACCCATCTTCGTTCCGCC). These were inserted into PX458 (Addgene) by Bbs1 (NEB, Cat#R3539S) cloning and guides targeting Axin1 were used with single stranded repair templates (Ultramer DNA Oligos, IDT) for co-transfection of HEK293T cells to generate Axin1 P372A (GGATGG-GATCCCCCCATACAGGATCCGTAAGCAGCACCGCAGGGAGATGCAG

GAGAGCGTGCAGGTCAATGGGCGGGTGCCCCTACCTCACATTGCCG
TAAGTACCGGCTTTGCGGTCCTCAGCCCATCGTCCTCCCCGCTCAGC
GTGGGCCTGGTG), Axin1 RR > D (ATTGGCACGGTGCTGGCCCTCCT
GCTCCTCTCTGAGTTAACGGCTGCGCCTCTTTCTCCTGGACAGGGATG
GGATCCCCCATATCGAATAGACAAGCAGCATGACCGGGAGATGCAG
GAGAGCGTGCAGGTCAATGGGCGGGTGCCCCTACCTCACATTCCC
GTAAGTACCGGCTTTGCGGTCCTCAGCCCATCGTCCT), Axin1 RKR > A
(ATTGGCACGGTGCTGGCCCTCCTGCTCCTCTCTGAGTTAACGGCTG
CCTCTTTCTCCTGGACAGGGATGGGATCCCCCCATATGCGATCAGAG
CGCAGCATCGCGCGGAGATGCAGGAGAGCGTGCAGGTCAATGGGC
GGGTGCCCCTACCTCACATTCCCGTAAGTACCGGCTTTGCGGTCCTC
AGCCCATCGTCCT), Axin1 DED$_3$ > A (TGGCGGGAGCCAGTCAACCCC-
TATTATGTCAATGCCGGCTATGCCCTGGCCCCAGCCACCAGTGC-
CAACGCCAGCGCGCAACAGTCCCTGAGCTCCGCTGCAGCCACCCT
AAGCTTGACGGCCAGTTCCGTGTAAGTCCCGCCCCGGATACCCAGTC
CCTCACGGACAGCAGTGTGTAAGTCCGGTCCCAGACACCTTGT),
Axin1 S > A (CATGCCCTGCTTGCCTGTTTGCAGGTATGGATCCTGGC
GGGAGCCAGTCAACCCCTATTATGTCAATGCCGGCTATGCCCTGGC
CCCAGCCACCGCTGCTAATGACGCCGAACAACAGGCCCTCTCAGCA-
GATGCAGACACCCTGTCCCTCACGGACAGCAGCGTGTAAGTCCCG
CCCCGGATACCCAGTCCCTCACGGA). Single cell clones were grown
in culture media supplemented with Plasmocin (InvivoGen, Cat#ant-
mpp) to protect against mycoplasma infection and analyzed by DNA
sequencing. Briefly, cells were lysed in Squash buffer (10 mM Tris
pH8.0, 1 mM EDTA, 25 mM NaCl, supplemented with 200 µg/ml Pro-
teinase K) and heated in a Thermal Cycler (65 °C for 15 min, 96 °C for
2 min, 65 °C for 4 min, 96 °C for 1 min, 65 °C for 1 min, 96 °C for 30 s).
DNA was amplified using the following primers (Axin1 exon3_ampF:
CTCTGTGTGTGCACCAAAGC; Axin1 exon3_ampR: CAACCGACA-
CAAAGCTGAGC; Axin1 exon3_seqR: TCTCACTGAGACCTGGGGAG;
Axin2 exon1_ampF: AGCCCCTGCTGACTTGAG; Axin2 exon1_ampR:
TAGGTCTTGGTGGCAGGCTT; Axin2 exon1_seqF: GCTATGTTGGT-
GACTTGC; Axin1 exon4_ampF: CCTGGGCTTTACCATCACGT; Axin1
exon4_ampR: CCTGGAGACGACCACTGTTC; Axin1 exon4_seqR:
GGATCACACGCTTACGCCTA) and sequence chromatograms were
imported into TIDE (tide.nki.ni).

The CRISPR guides for knocking out AP2µ were designed by
Horizon Discovery, as follows: GATGTCATCTCGGTAGACTC (exon 1),
TACCCACAGAATTCCGAGAC (exon 3), GCATGCCTGAATGCAAGTTT
(exon 5), GTGGAAGGTGCAGTCATCAA (exon 6). Guides were tested
in combination (as pools) or individually, however it was not possible
to isolate any viable knock-out lines, even after transient transfec-
tions of guides, indicating that complete knock-out of AP2µ is cell-
lethal.

## SuperTOP assays

HEK293T cells were plated in 6-well plates at a concentration of
$1 \times 10^6$ cells per well and co-transfected with 500 ng of wt or mutant
LRP6-GFP or GFP empty vector plasmids, 250 ng of M50 Super 8x
TOPFlash plasmid (#12456, Addgene) and 50 ng of CMV-Renilla
plasmid using PEI (Polysciences). After 16–18 h, Wnt pathway sti-
mulation was achieved by incubating cells in WCM or control
medium collected from L cells (ATCC, Cat#CRL-2648, Cat#3216)
according to the manufacturer's instructions, for 6 h unless other-
wise stated. Luciferase reporter assays were performed according
to the manufacturer's protocol (Dual Luciferase Assay kit, Promega,
Cat#E1910).

## Statistical analysis

All error bars are represented as mean ± SEM for 3–6 independent
experiments. Statistical significance was calculated in Prism V10.0
(GraphPad) by two-tailed Student's $t$-test or one-way ANOVA followed
by Dunnett's multiple comparisons; $p$-values between indicated data
points are given in each figure legend (and are denoted in individual
panels as *, $p < 0.05$; **, $p < 0.01$; ***, $p < 0.001$; ****, $p < 0.0001$).

## Reporting summary

Further information on research design is available in the Nature
Portfolio Reporting Summary linked to this article.

## Data availability

Atomic structures and structure factors generated in this study have
been deposited in the Protein Data Bank (PDB) under accession code
9FIX and 9FIY. Previously published structures are available at PDB
under accession codes 1BW8, 4NM5 and 4NM7. NMR assignments have
been deposited in the Biological Magnetic Resonance Bank (https://
bmrb.io) under accession numbers BMRB 52482, BMRB 52483 and
BMRB 52493. The mass spectrometry proteomics data are available via
ProteomeXchange under identifiers PXD054245 and
PXD061649. Source data are provided with this paper.

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

## Acknowledgements

We thank Dom Bellini, Stephen McLaughlin, Chris Johnson, the LMB Mass Spectrometry and Cell Sorting Facilities for technical support, David Owen and Gail Johnson for plasmids, the Diamond Light Source for beamline access (beamline I04), Bill Weis, Michael Enos, Xi He, Feng Cong, Scottie Robinson, Eva Wenzel, Phil Beachy and Hugh Pelham for helpful discussions, and Emma Thurston for help with the manuscript preparation. This work was supported by grants from the Medical Research Council (U105192713 to M.B.), Cancer Research UK (C7379/A24639 to M.B.) and the Wellcome Trust (226525/Z/22/Z to M.V.G.). To Bill Weis who inspired much of this work.

## Author contributions

M.B. and M. V. G. designed the project and acquired funding; M.V.G., E.F.-E., T-M.L., T.J.R., M.R., and C.B. performed the research and analyzed the data; M.B. supervised the project and wrote the original draft; all authors contributed to the writing of the paper.

## Competing interests

The authors declare no competing interests.
