## [Peer Review file · Nature Communications]

Wnt signalosome assembly is governed by conformational flexibility of Axin and by the AP2 clathrin adaptor

Corresponding Author: Dr Mariann Bienz

Version 0:

Reviewer comments:

Reviewer #1

(Remarks to the Author)

The Wnt pathway is an evolutionarily conserved cell-cell communication pathway that plays a critical role in the development of metazoans and, when improperly regulated, has a major impact on a legion of human diseases, particularly cancer. Despite several decades of investigation, the detailed mechanism of how chemically encoded information is transmitted from the ligand to the cytoplasm remains unclear.

A key feature of our current model of Wnt signaling is the interaction between the co-receptor, LRP6, and the scaffold protein, Axin. In the absence of the Wnt ligand, it assembles a large complex (β -catenin destruction complex) to promote the degradation of β -catenin. A key feature of this model is that Wnt binding promotes the interaction between LRP6 and Axin, resulting in the inhibition of β -catenin degradation. The prevailing model focuses on the role of LRP6-mediated inhibition of GSK3 activity and subsequent inhibition of β -catenin phosphorylation/ubiquitin-mediated degradation. Paradoxically, activation of LRP6 and its interaction with Axin requires GSK3 activity. The complex interplay between the role of GSK3/Axin and LRP6 remains a major question in the Wnt field.

In the current manuscript by Gammon et al., the authors provide biophysical, biochemical, and gene editing to start to unravel the interaction between GSK3, Axin, and LRP6 during Wnt signaling. They show that the membrane-proximal PPPSPxS motif of LRP6 is the predominant region for GSK3 and Axin binding. They provide evidence for an intrinsically disordered segment of Axin, the LRP6 interacting domain (LID), that can be in two conformations: "foldback" and "horseshoe," both of which inhibit Axin/GSK3 binding to LRP6. Finally, they show that signaling by LRP6 requires its two AP2 binding motifs, suggesting a requirement for a clathrin-mediate process for Wnt signaling.

Thus, the current paper provides a structural model of how Axin, through the action of GSK3, could transition from promoting the degradation of β -catenin to interacting with and activating the Wnt co-receptor, LRP6. The experiments are generally of high quality, and the topic is important to the Wnt field. A few concerns (mostly controls), however, need to be addressed before this manuscript is suitable for publication in Nature Communications:

- 1) Overall, the NMR data studies are well done, particularly the use of the Alanine mutations of Axin (DED4>A) as negative controls. The authors state that all three of their Axin constructs (Lip-A1, Lip-A3, and Lip-A5) bind to LRP6 (Lip-L1), but the majority of the NMR data shown focuses on Lip-A5. I imagine that this is because this construct is the smallest and gives the cleanest data. It would be useful to include the data for the other constructs in the supplement.
- 2) In Fig 1b, the samples are not loaded evenly across the gel, and the lane with the most significant amount of LRP6-GFP, Flag-Axin, and GSK3-HA pulled down appears to show the greatest level of expression. An IP control showing background binding with WT proteins would be useful. Also, the samples are overloaded for the immunoblotting of IP LRP6-GFP.
- 3) Phospho-LRP6 appears to be shifted up in the input lane containing all three wild-type proteins in Fig 1b. Is this a real shift or a gel artifact? Please comment on this.
- 4) Supplementary Fig1c- the expression of GSK3 is not even across the conditions. The authors state that mutant GSK3 fail to bind LRP6. To this reviewer, GSK3 appears to be co-precipitated with LRP6-GFP across all the conditions. The gel appears to be overloaded, so it's difficult to observe changes.

- 5) Fig2a- Please indicate A4 relative to A3 and A5. Also, the labeling of the LID-GID fragment is confusing.
- 6) Fig 3d- Would changing the three serine to aspartate or glutamate mimic the Axin foldback conformation?
- 7) Fig 4- Are there any differences in the phospho-LRP6 or GSK3/Axin levels when co-IP'ed with LRP6 for the Axin P372A mutants?
- 8) If the P372A blocked the termination and prolonged Wnt signaling (as was implied), one would predict that a brief activation by Wnt3a (followed by wash-out), would result in prolonged elevated β -catenin and phospho-LRP6 levels compared to controls.
- 9) Fig 4b- the immunoblot of LRP6 pulled down with AP2 is barely visible
- 10) Fig 4- Loading controls are lacking for 4d and 4e
- 11) The model, as presented in Figure 7, is confusing. Could the authors describe their model more clearly in terms of the transitions between the three states of Axin-GSK3 during Wnt signaling?
- 12) In the discussion, the authors should go into more detail about their model and how it fits with the model from the He lab (Kim et al., 2013), proposing a "closed" and "open" conformation for Axin occurring via an intra-molecular interaction.
- 13) Early work from the Wu lab showed that Axin interacts with LRP5 in a yeast two-hybrid (Mol Cell, 2001). This was followed by studies from the Wieschaus lab (Tolwinski et al., Dev Cell 2003) showing that full-length Axin did not interact with Arrow in a yeast-two hybrid and that "full-length Axin does not interact with Arrow because the N-terminal half of Axin prevents interaction with Arrow." Both studies should be cited. Similarly, the paper by Cseleyi et al. (PNAS 2008) demonstrating that the intracellular domain of LRP6 could directly inhibit GSK3 activity using purified, recombinant proteins should be mentioned.

Reviewer #2

(Remarks to the Author)

This is an exciting study that provides important new insights into the assembly mechanism of the Wnt signalosome, which forms upon Wnt stimulation. The authors demonstrate extensive cooperativity between AXIN, GSK3 and LRP6 in the assembly of the signalosome, from where they go to dissect the mechanisms underlying this cooperativity. The most exciting finding of the study is the direct, phospho-regulated interaction between AXIN and LRP5/6. The study goes a long way towards answering the long-standing question of how AXIN is recruited to LRP6. Binding between two non-domain portions of two different proteins, alongside a phosphorylation-controlled foldback mechanism, provides a precedent in the field. The work is technically excellent, covering a wide range of biophysical techniques and CRISPR editing for functional studies. Deciphering the binding events at the molecular level is technically challenging territory as the involved interactions are weak and dependent on avidity. NMR spectroscopy studies have proven extremely valuable in this context, and also in providing new insights into GSK3 binding to AXIN. By combining mechanistic and structural studies with proteomics, the authors further tie in the relevance of the endocytic machinery into signalosome function, confirming and expanding on previous findings. The work covers a lot of ground and opens up new ways of thinking about the intricacies of Wnt signalosome assembly and the complexities of regulatory events in Wnt/ β -catenin signalling. I am enthusiastic to see this work published. I also enjoyed the comprehensive Discussion section that ties together different aspects of signalosome regulation.

Specific comments

Introduction, line 8: Jesse Zalatan's group have done some interesting work showing how GSK3 activity is directed towards β -catenin in the pathway, insulated from other substrates. Rather than just citing the review, could the authors consider referencing these studies?

Can the authors indicate Lip-A4, A308-E384 in the schematic shown in Figure 2A? Can the LIDup be indicated as well?

p. 5, paragraph 3: Contrary to the text, which refers to all three Axin fragments in Figure 2b, only fragment A5 is shown in the figure. Can the authors show the equivalent data (BEST-TROSY spectra) for the different Axin fragments, if needed in the supplementary material? This will make it a lot easier for the reader to follow the narrative.

Figure 2 and Supplementary Figure 2: Can the authors expand the panel labelling, please, to include the isotopic labelling of Lip-L1? (I.e., it is not immediately clear from Figure 2 that panel e is from ^{13}C labelled protein.)

p. 5, last paragraph: The authors refer to an AlphaFold 2 prediction, but do not include a reference to a figure (Figure 3c). Can this be added?

p. 6, paragraph 2: ITC is not in Figure 2e, and I cannot find it elsewhere either. Can the authors also include the ITC data for GSK3 and Lip-L1? This is discussed, but without a figure. Similarly, ITC data referred to in the last paragraph on page 6 are

not shown.

Could the foldback model in Figure 2e be tested by DLS or SEC-MALS? Foldback would be predicted to reduce the hydrodynamic radius. Testing the phospho-regulated interaction with LRP6 biophysically would also be interesting.

p. 7, paragraph 2: Can the authors add that the data are from ITC? This is of course clear from the figure but not the text.

Figure 4a: Can the helices be labelled for clarity? Does the sequence region in red correspond to the extended alpha-helical region within GID referred to on p. 7, paragraph 3? This could be pointed out.

For the AlphaFold predictions, can the authors please include the pLDDT and PAE plots in the supplementary material? How similar were the top-5 models for both AF2 and AF3? Does the predicted AF3 model have worse confidence metrics?

The reporter assays in Figures 4e and e suggest an antagonism between GSK3 and LRP6 for AXIN binding. Can this be tested directly by interaction assays?

Figure 5a: Can the authors expand the figure legend to indicate which data the signalling activities referred to in the figure correspond to? I cannot see data for deltaB, for example.

Figure 5c: Does mutation of each of the single tandem motifs in the GFP fusion affect interaction? If yes, this would demonstrate functionality of the atypical motif at the level of binding. This would nicely complement Figure 5f.

Panels d and e of Figure 5 are very small. The structural findings could be presented more prominently. Some details (such as the map corresponding to the peptide ligand bound to the protein) could be included in the supplementary material.

p. 10, paragraph 2: Can you explain the LRP6m10 mutation? It is not part of panel 5a.

p. 11, paragraph 1: The authors may want to change the term "near-physiological" to something more specific as it may mean different things to different readers.

p. 14, penultimate paragraph: Given the high degree of intrinsic disorder, I am not so sure a spatially confined self-interaction between adjacent elements in AXIN would suffice to place the RGS and GID closer together. This point could be de-emphasised.

Minor points

p. 6, paragraph 3: The term "LIDup" is introduced earlier, but not used to refer to "the sequences upstream of LIDalpha". This made me wonder whether the description refers to LIDup or a different region. Can names be used more consistently?

The nomenclature of mutant variants for Figure 5f appears inconsistent between the text and the figure. (YYF>A in the text has no equivalent in the figure.)

Figures 4 and 5, and supplementary material: Where the authors show western blots accompanying reporter assays, it may be worth labelling the Western blots separately for clarity as alignment with the bar graphs is not always possible.

Can the authors indicate molecular weight markers for all Western blots?

Could the authors consider finding different names for LID, GID and BID as none of these are actually "domains"?

Omitting the "Lip" pre-fix from construct names may help with simplifying nomenclature as all constructs share the same tag. This would also help with making the different construct names more recognisable. At the moment, due to the shared tag reference, the names are almost identical.

In general, the authors could expand their explanations of the NMR experiments to make them more accessible to readers not familiar with looking at this type of data.

The order of the figures means there is a lot of jumping back and forth when reading. Could this be improved?

p. 8, paragraph 1: "pulse-chase experiments that the turnover"  "pulse-chase experiments showing that the turnover"

p. 9, paragraph 2: "doable"  "feasible"

Note to the Editor: Can figure legends be placed directly underneath figures? This would help reviewers.

Reviewer #3

(Remarks to the Author)

A critical step for Wnt/ -catenin pathway activation is the recruitment of AXIN1 and its associated kinases GSK3 and CK1 to the Wnt receptor complex at the plasma membrane, after which a sequence of molecular events mediate inhibition of the GSK3 kinase.

In this study, Gammons et al aim to shed light on the poorly understood question of how the AXIN-GSK3 complex is recruited to the cytosolic tail of LRP6. They use NMR, biochemical methods, structural modeling and CRISPR-mediated gene editing to uncover that an LRP6 region incorporating the membrane-proximal PPPSPxS motif interacts with both AXIN1 and GSK3. The LRP6-binding site in AXIN1 is mapped to the intrinsically disordered (IDR) region, to a sequence N-terminal to the GSK3-interaction domain. Interestingly, access of this AXIN1 IDR region (AXIN1-LID) for LRP6 binding appears regulated by phosphorylation. Convincing data are shown that phosphorylation of AXIN1-LID induces a back-folded conformation which makes this site inaccessible for LRP6 binding and thus prevents inappropriate Wnt pathway activation in unstimulated cells. In the last part of the manuscript, the authors anticipate that clustering of LRP6 tails by interaction with the clathrin adaptor AP2 is essential for AXIN and GSK3 binding cooperativity and WNT pathway activation.

The work describes a number of interesting new findings that shed light on a poorly resolved issue in the field of WNT/ -catenin signaling. The combination of technologies is a strong point. The overall model that is presented however needs more convincing experimental support.

Also, the manuscript is not an easy read, partly due to a lack of explanation of the rationale and technologies and the complex names of motifs and regions. It needs editing to make this work more accessible to a broad readership including newcomers to the field.

Major points:

A major concern is that, due to protein expression problems, focus has been placed on a narrow region of the AXIN1 IDR. Since IDRs often use discontinuous motifs for interactions with partners, a risk is that the work may not have identified all possible LRP6-interaction sites. Can the authors exclude a role of other parts of AXIN1 in facilitating LRP6 binding?

Page 4: '...efficient co-IP between the three proteins is only seen with their wild-type (wt) versions, but not with catalytically-dead GSK3 (K85R) nor with the non-phosphorylatable mutant LRP6m10'

Why did the authors choose to use K85R for detecting NMR binding in later experiments? The rationale for this is not clear (since this GSK3 version does not bind in in co-IP) and not commented on.

Page 5: 'Phosphorylation at S1490 can only be detected if the wt proteins are co-expressed' (Figure 1b). How do the authors reach this conclusion? The phospho-blot of the IP shown on the right shows strong signal of S1490 in all lanes.

Also Figures 2 and Suppl Fig 2 shown NMR-based LRP6 interaction data for Lip-A5 mutants, but bleach maps for wild-type Lip-A5 are missing. These data should be included for comparison.

The identified region in LRP6 that accommodates both Axin and GSK3 binding is interesting and supported by convincing data. The LRP6 interface is indicated as the 'proximal PPPSPxS motif' (e.g. in the abstract) but this does not seem to be entirely correct. NMR data indicate that AXIN1-based fragments indeed bind this motif as well as flanking regions on both sides. For GSK3, however, the PPPSPxS motif seems to participate less in the interaction as compared to flanking regions. This observation needs better specification in the abstract and discussion of the results. In particular, since the phosphorylated version of the motif must bind GSK3. How these observations correlate is not entirely clear.

AXIN-GSK3 interaction models generated by AF2 and AF3 are not in agreement, which is a weak point of the presented work. The authors take the AF2 model as best fitting, since it is in agreement with most of their data. Alternative explanations are however not considered.

Furthermore, any information on estimated confidence of the predicted models is missing. For all AF modeling, ipTM values must be indicated. If ipTM values are below 0.5-0.6, the model is considered unreliable. Any AF data that do not meet this threshold for prediction quality should be removed.

Figure 4: The hyperinducible phenotype of AXIN1-P372 lines is interesting and supports the model that this mutant does not interfere with destruction complex activity but displays improved interaction with LRP6 (the latter event being induced by Wnt binding to the receptors). It is essential however to better substantiate these findings in cells at the endogenous level: do the indicated mutants (P372, RR>D, DED3) indeed display increased/decreased binding to LRP6 upon Wnt pathway activation?

Previous findings indicated that GSK3 and Axin associate preferentially with LRP6 upon phosphorylation of its PPPSPxP motifs. In fact, in the summarizing model, GSK3 binds to a phospho-site in LRP6, but the interaction mapping in the current manuscript do not include phospho-forms of LRP6. How do the authors factor in phosphorylation in their interaction model, would this influence relative binding affinity of GSK3 and affect interactions with AXIN?

Western blots for IP experiments need to be quantified.

Minor points:

Page 3: Make clear that motifs within LRP6 are indicated as 'motif A', 'motif B' etc

“This supports a model that envisages clustering of LRP6 in clathrin-coated (CC) structures upon Wnt stimulation by AP2, ...”

Sentence is multi-interpretable and therefore confusing

The naming of motifs and regions makes the manuscript a very difficult read (Lip, LID, GID, A3, LIDa, LiDup, L1). Please keep the protein name always included in these names (e.g. AXIN1-LID, LRP6-...) preferably including aa boundaries, for clarity throughout the manuscript.

Page 5: ‘To be able to observe Axin-dependent attenuation of the proline residues within PPPSPAT...’ Reads odd, this should be ‘peak attenuation’ or ‘signaling attenuation’

Page 5: Lip-A4 is mentioned but not shown in any of the figures

Page 6: ‘¹³C-detected 2D-CON spectra’ – provide rationale for using this method, to accommodate a broad readership

Page 6: ‘We therefore incubated an ¹⁵N-labeled distal fragment (Lip-L2) with Lip-A5, which...’ The figure (Suppl Fig 3) shows Lip-A3.

Page 8: describe the predicted role of Proline in the structure first and then discuss the Pro-mutant.

Page 8: ‘the AXIN2-ko line’ – needs better introduction

Page 8: ‘Assessing Wnt responses of two independent lines...’ Please indicate every time which line you talk about, to avoid confusion. In this case the AXIN1 P372-mutant lines are compared with WT.

Page 8: indicate which subfigure of Suppl fig 4 you refer to

Page 9: more clearly explain that the YYYYF carries two mutated motifs

Page 10: ‘Indeed, wt 2xTF but not its YYYY>A mutant version...’ Should this not be YFYF in the case of the transferrin receptor?

Page 11: ‘The thus identified consistently...’ sentence is flawed

Page 14: ‘CC structures’ needs better explanation

Suppl fig 1b: make clear whether GSK3 is co-expressed in figure and legend.

Page 9: ‘LRP6 and Frizzled are subject to clathrin-dependent endocytosis and lysosomal degradation in the absence of Wnt, following earmarking by Dishevelled and the transmembrane ubiquitin ligase ZNRF3/RNF43, which appears to rely on an AP2-binding motif in Dishevelled and on the tandem Y-motifs in LRP6’
This statement suggests that ZNRF3/RNF43 are involved in AP-2 interactions. This has not been shown, please adapt.

Reviewer #4

(Remarks to the Author)

The study by Gammons et al. addresses a highly relevant and timely topic in the field of Wnt signaling, focusing on the conformational flexibility of Axin and its interaction with both Lrp6 and the AP2 clathrin adaptor. The Wnt pathway is crucial for cellular signaling processes related to development and disease, and understanding the assembly of the Wnt signalosome is a significant goal for the field. While this paper provides valuable insights into the dynamics of Axin and its interactions, the manuscript suffers from conceptual and methodological weaknesses that need to be addressed. The research is ambitious and presents important observations, particularly in mapping the Axin-Lrp6 interaction and highlighting the competition between Axin and GSK3 on Lrp6. However, there are several critical issues, especially regarding the overall data interpretation, and the clarity of the presented model.

Strengths:

1. Clarification of Axin and Lrp6 Interactions - The authors make substantial progress in understanding the interplay between Axin and Lrp6, which is fundamental to Wnt signaling. The detailed mapping of these interactions is a key strength of the study and adds new information to the field. This is a timely contribution, given the current focus on the structural basis of Wnt signalosome assembly.
2. Competition of Axin and GSK3 - The study highlights how Axin competes with GSK3 for interaction sites on Lrp6, offering insights into how the Wnt signalosome dynamically regulates its components.

Major Concerns/Weaknesses

1. Disconnection Between Axin-Lrp6 and AP2 parts - A major concern is that the paper feels like two distinct stories: the detailed mapping of Axin-Lrp6 interactions, and the linkage to AP2 and clathrin. The transition between these sections is not well articulated, and the mechanistic connection between Axin-Lrp6 interactions and AP2 is unclear. Without a stronger

integration of these two aspects, the paper seems fragmented, as if two incomplete studies have been combined into one. The authors need to more convincingly demonstrate how the findings regarding Axin-Lrp6 feed directly into the role of AP2 in the assembly of the Wnt signalosome. At present, the paper gives the impression of being two separate research efforts, leaving readers to infer the link between the two.

2. Overinterpretation of the Data - The proposed model, while interesting, appears overinterpreted in its key parts. Although the data align with the proposed model, the authors stretch the conclusions beyond what the experimental evidence strongly supports.

- In particular, Fig. 4b/c presents a proposed binding mode that lacks support from AlphaFold's newer versions. Additionally, the use of motif C in the model, when motif A is discussed in other sections, creates confusion about which motif is functionally relevant. This is the key component of the model and the axin foldback interaction with GSK3 needs to be proven experimentally. Further, in my opinion the data in Fig. 3 c/d shall be improved by analysis of the functionally deficient Axin mutants and by the evidence of the electrostatic nature of the interaction.

- Another issue arises in Fig. 2h, where the sequence shown corresponds to the mutant YYYYF, which is only introduced later in Fig. 5. This adds to the confusion in interpreting the experimental results.

- Moreover, the proteomics data are presented in a somewhat disjointed manner. Different variants of the BioID assay are used for different mutants, but the rationale for these choices is not clearly explained. This undermines the consistency of the data and makes it difficult to evaluate the impact of the mutations and link.

3. Challenges in Following the Schematics and Figures - Many of the figures, particularly Fig. 2, are difficult to interpret without extensive cross-referencing with the text. This detracts from the overall clarity of the manuscript. For instance, the term "Lip" in Fig. 2 is confusing and should be replaced with more descriptive labels like "Lrp6 (L1)" and "Axin1 (A3)" to facilitate understanding. Readers unfamiliar with the proteins will struggle to follow the experiments without digging into the manuscript for explanations.

- Adding sequential information (e.g., amino acid positions) at boundaries or motifs in the figures would significantly enhance their clarity. The study works with four different proteins – Axin, Lrp6, GSK3 and AP2 – and it was a real challenge for the reviewer to follow the text/figures. In its current form, the schematics demand a deep familiarity with the protein constructs used in the study, which limits accessibility to a broader readership.

Minor Concerns:

- The figure legends require more detail to clearly explain what is being shown. As it stands, the figures are not self-explanatory, and clearer labels should be used for the different constructs and proteins.

- Throughout the paper, the naming of the protein constructs could be more consistent. Given the complexity of the interactions and the number of different proteins and mutants tested, clearer and more standardized labeling would help readers keep track of the experimental components

Conclusion:

This manuscript addresses an important and timely topic in Wnt signaling, providing new insights into the conformational flexibility of Axin and its interaction with Lrp6 and link to AP2. However, there are significant conceptual and methodological issues that need to be resolved. The current version of the manuscript presents two premature stories that feel artificially merged into one model. In its current form, the work does not sufficiently connect the findings from the Axin-Lrp6 mapping with the role of AP2 in the assembly of the Wnt signalosome. To address these concerns, I recommend either filling the gaps in the data to strengthen the proposed model or treating each of the two stories separately to ensure that each one is complete on its own. The paper would also benefit from a clearer and more conservative interpretation of the data, as well as improvements in the clarity and accessibility of the figures.

Reviewer #5

(Remarks to the Author)

Gammons et al report important new insights into the interactions governing Wnt signalosome assembly. Their data, drawing on a range of methods, provide compelling evidence for (i) a novel site in Axin they term the LID which can mediate direct interaction with LRP6 (ii) a mechanism for conformational occlusion of this LID site (antagonism of LRP6 binding) triggered on phosphorylation by GSK3 and (iii) a model for the role of AP2-mediated LRP6 clustering in enabling an Axin-GSK3 complex to bind simultaneously to two LRP6 receptors. Each of these three findings provides hereto missing pieces that knit together to fill a substantial gap in our understanding of Wnt signalling. The data for the horseshoe model of the Axin-GSK3 interaction are less compelling given the different interface resulting from alphafold3 but are presented with well balanced arguments and certainly should be included.

There are a few points where revisions to the text would help the reader, particularly non Axin specialists.

P4 The sentence starting "We also provide evidence" is rather confusing, possibly because of the repeat use of "antagonize".

P7 BID suddenly appears in the text but isn't defined, maybe the schematic of Axin could be referenced?

P7 Ditto the sudden appearance of GiD

Version 1:

Reviewer comments:

Reviewer #1

(Remarks to the Author)

The authors have satisfactorily addressed my concerns, resulting in a significantly improved revised manuscript. This is an excellent paper that will make a valuable contribution to the Wnt field by providing some molecular clarity to the mechanism of Wnt pathway activation.

Reviewer #2

(Remarks to the Author)

The revisions have further improved an already excellent manuscript. All my points have been addressed satisfactorily. The revised version contains exciting new details of the multi-pronged interaction between AXIN and GSK3, predictions of the kinases responsible for AXIN phosphorylation, and a plausible potential explanation for a phosphorylation-regulated AXIN stability control system by USP7. As I pointed out in the initial round of review, this study provides important new insights into Wnt signalosome formation and covers a vast ground. The work will substantially advance the Wnt field and, given the proteomics components, serve as a valuable resource. There are many interesting observations whose exhaustive investigation go beyond the scope of a single manuscript, so the findings will open up many interesting avenues for future study. Congratulations to the authors!

There remain a few relatively small and easy-to-address points for the final revision prior to publication.

Specific points

- line 157: The authors report very similar bleach maps for WT or kinase-dead GSK3 and Lip-L1 (LRP6). Readers may wonder how this fits with the phosphorylation dependency in the IP experiment (Figure 1b). It may help to highlight the different contexts of these experiments to address this question up-front.
- line 183: The authors state that the residues upstream of LIRalpha (LIRup) do not interact with 15N-labelled Lip-L1 (LRP6), cross-referencing Figure 2a. However, I don't think this is clear from that figure.
- Figure 3: Can the stoichiometries referred to in the text (line 190) be indicated in the figure?
- lines 238/239: The text states that intramolecular interactions are shown in Figure 4d, e. For clarity, the authors may choose a different wording here as the interactions themselves are not modelled.
- p. 8, paragraph 3: Are the described features similar in the top-ranking AlphaFold outputs? It will be good to include this information in the main text.
- Generating the proteins for biochemical, biophysical and structural studies is technically challenging. Can the authors add SDS-PAGE gels showing the final proteins? I suppose the final buffer is sodium phosphate? (The counter-ion is not indicated.)
- lines 277-286: Well-accepted terms of kinases (N-lobe, C-lobe) could be used, rather than "apex", etc.
- On the note of AlphaFold predictions, the authors describe pLDDT as a confidence measure for interactions, whereas it is a confidence metric for local structure (per residue). This of course is in turn indirectly linked to interaction confidence; however, the latter is better illustrated by the PAE score.
- p. 11, last paragraph: Can the ITC data be added as supplementary material? On the note of ITC, can the authors indicate how many experiments were performed?
- Supplementary Figure 8: Can the contour level be indicated?
- Supplementary Figure 9c appears to suffer from some graphics anomaly. In general, the supplementary file looks like a low-quality scan. Can this be rectified?
- Supplementary Figure 9c: No matter whether the recombinant gene is introduced transiently or stably, there is overexpression. I therefore suggest the nomenclature of "OE" to be changed to "transient".
- Suggestion for title: "... flexibility of Axin and [the] AP2 clathrin adaptor"
- Wnt>b-catenin  Wnt/b-catenin
- p. 3: Myristoylation is not sufficient for membrane anchoring, and it is likely it needs to collaborate with palmitoylation for this purpose.
- line 200: Figure 2h

- Figure 6b: The subscript 4 is missing in the mutant variant name.
- Line 319: AF3 (instead of AF2)
- Line 343: contributes  contribute

Reviewer #3

(Remarks to the Author)

The authors have addressed my concerns, mostly satisfactorily. They added valuable new data that support and solidify their model and the manuscript is better readable. However, there are still some issues that require attention:

- The newly expanded list of proteins identified using TurboID is interesting, especially in combination with the Kinase Library analysis. However, this also led to an addition of more speculative interpretations. The proposed roles of the kinases and the phosphorylated residues in the model presented in Figure 9 are primarily inferred from proximity labeling data, predictions from AlphaFold models and the Kinase Library. These conclusions thus currently lack support from direct interaction or functional validation, and this limitation should be acknowledged more explicitly in the text.
- At many places in the manuscript, the authors mention 'unpublished results' or 'personal communication'. For clarity and transparency, these results should be added to the supplemental information or properly referenced.
- The experiments shown in Figure 4g are confusing regarding the boundaries of LIRup vs LIRa, as the 43-mer LIRup peptide contains a part of LIRa RKR motif as well (2 of the 3 RKR residues). This complicates the readout and interpretation. Please better explain the setup of this experiment.
- Figure 5b and c: the disordered Axin region that binds to GSK3beta should be made better visible by using a more contrasting color.
- Figure 9: Please indicate the meaning of symbols within the figure itself and add arrows from kinase to target site, to make the figure more self-explanatory. Also, please indicate the consequences of the different phosphorylation events in the figure.
- Line 600 – Fig 6c should be Fig 6d

We are very pleased about the positive feedback from our reviewers and thank them for their constructive comments, which we have addressed as detailed below (highlighting our **revisions in bold** below, and in **red** in the manuscript file).

Reviewer #1 (Remarks to the Author)

The Wnt pathway is an evolutionarily conserved cell-cell communication pathway that plays a critical role in the development of metazoans and, when improperly regulated, has a major impact on a legion of human diseases, particularly cancer. Despite several decades of investigation, the detailed mechanism of how chemically encoded information is transmitted from the ligand to the cytoplasm remains unclear.

A key feature of our current model of Wnt signaling is the interaction between the co-receptor, LRP6, and the scaffold protein, Axin. In the absence of the Wnt ligand, it assembles a large complex (β -catenin destruction complex) to promote the degradation of β -catenin. A key feature of this model is that Wnt binding promotes the interaction between LRP6 and Axin, resulting in the inhibition of β -catenin degradation. The prevailing model focuses on the role of LRP6-mediated inhibition of GSK3 activity and subsequent inhibition of β -catenin phosphorylation/ubiquitin-mediated degradation. Paradoxically, activation of LRP6 and its interaction with Axin requires GSK3 activity. The complex interplay between the role of GSK3/Axin and LRP6 remains a major question in the Wnt field.

In the current manuscript by Gammon et al., the authors provide biophysical, biochemical, and gene editing to start to unravel the interaction between GSK3, Axin, and LRP6 during Wnt signaling. They show that the membrane-proximal PPPSPxS motif of LRP6 is the predominant region for GSK3 and Axin binding. They provide evidence for an intrinsically disordered segment of Axin, the LRP6 interacting domain (LID), that can be in two conformations: “foldback” and “horseshoe,” both of which inhibit Axin/GSK3 binding to LRP6. Finally, they show that signaling by LRP6 requires its two AP2 binding motifs, suggesting a requirement for a clathrin-mediate process for Wnt signaling.

Thus, the current paper provides a structural model of how Axin, through the action of GSK3, could transition from promoting the degradation of β -catenin to interacting with and activating the Wnt co-receptor, LRP6. The experiments are generally of high quality, and the topic is important to the Wnt field. A few concerns (mostly controls), however, need to be addressed before this manuscript is suitable for publication in Nature Communications:

1) Overall, the NMR data studies are well done, particularly the use of the Alanine mutations of Axin (DED4>A) as negative controls. The authors state that all three of their Axin constructs (Lip-A1, Lip-A3, and Lip-A5) bind to LRP6 (Lip-L1), but the majority of the NMR data shown focuses on Lip-A5. I imagine that this is because this construct is the smallest and gives the cleanest data. It would be useful to include the data for the other constructs in the supplement.

We focussed on Lip-A5 because it contains the minimal Axin region that retains full binding to LRP6. Indeed, the bleach-maps obtained from the BEST-TROSY spectra of all three constructs are essentially indistinguishable. We now **include the bleach map from the BEST-TROSY spectrum for Lip-A5 as well** (in revised Supplementary Fig. 2d). But we decided to remove all mention of Lip-A4 from our manuscript, given that this fragment is

only marginally shorter than Lip-A5 and is therefore redundant with the latter. Presumably, this reviewer meant Lip-A4 rather than Lip-A1 (which does not exist).

2) *In Fig 1b, the samples are not loaded evenly across the gel, and the lane with the most significant amount of LRP6-GFP, Flag-Axin, and GSK3-HA pulled down appears to show the greatest level of expression. An IP control showing background binding with WT proteins would be useful. Also, the samples are overloaded for the immunoblotting of IP LRP6-GFP.*

These experiments involve co-expression of three proteins, and it is therefore rather challenging to get the loading completely even across a gel. However, we have conducted extensive optimisation of these colPs, testing additional ratios of Axin, GSK3 and LRP6 to those already shown in Supplementary Fig. 1. We have therefore decided to **show additional titration experiments** (in **revised Supplementary Fig. 1**), to document more compellingly that we observe robust colP only if all three proteins are wild-type. This also **obviates the need for an IP control** (minus bait) since binding is undetectable for most of the mutants (e.g. in the last 18 lanes of Supplementary Fig. 1).

Regarding the overloading of LRP6-GFP bait, we used consistently 1% of the total input lysate and 20% of the total eluate for all our IP blots, as now **specified in the Methods (p21, bottom)**, for optimal visualisation of the colP signals (from Flag-Axin and GSK3-HA), which meant that the GFP bait signals ended up being overexposed even after the shortest possible exposure (for 1 second) because of the high sensitivity of our GFP antibody. Rather than running a separate gel with less GFP bait for each colP experiment, we thought it more important to run multiple independent experiments (e.g. $n = 8$ in the case of the S>A mutants shown in revised Fig. 4c, as now **stated on p7, 3rd paragraph, and in the legend of revised Fig. 4c**). The same applies to the basic colP experiment shown in Fig. 1, which was performed at >3 times, after testing multiple expression conditions for optimisation of this experiment (see above).

3) *Phospho-LRP6 appears to be shifted up in the input lane containing all three wild-type proteins in Fig 1b. Is this a real shift or a gel artifact? Please comment on this.*

This shift is real, reflecting multiple phosphorylations of the various PPPSPxS motifs upon overexpression. However, since the shift patterns revealed by the anti-pS1490 antibody are extraneous and irrelevant for our main conclusion from these experiments, we decided to **remove the anti-pS1490 lanes** from this figure. As a result, the main message is conveyed more crisply from the simplified figure, namely that the robust association of Axin and GSK3 with LRP6 is only seen if wild-type versions of each protein are expressed.

4) *Supplementary Fig1c- the expression of GSK3 is not even across the conditions. The authors state that mutant GSK3 fail to bind LRP6. To this reviewer, GSK3 appears to be co-precipitated with LRP6-GFP across all the conditions. The gel appears to be overloaded, so it's difficult to observe changes.*

This reviewer is correct that the blots in our original Supplementary Fig. 1c made it appear as if both wild-type and mutant GSK3 associated with LRP6. However, we believe that this reflects background binding since the affinity (measured by ITC) is >100 microM, which is not detectable by colP. However, we decided to **remove these panels** as they simply served to confirm previously published data (in reference 40).

5) Fig2a- Please indicate A4 relative to A3 and A5. Also, the labeling of the LID-GID fragment is confusing.

We decided to delete Lip-A4 (for reasons stated above, see point 1), but **clarified the labelling of Lip-GID in Fig. 1a** (please note that we have also changed the nomenclature of LID, GID and BID to LIR, GIR and BIR, respectively, following a recommendation by reviewer 2).

6) Fig 3d- Would changing the three serine to aspartate or glutamate mimic the Axin foldback conformation?

We have tested this by colP, by changing these serines to aspartates, but the results were the same as those from the corresponding alanine substitutions (not unexpectedly since aspartate often does not mimic phosphorylation). Therefore, it did not seem worth testing this also by ITC, given the considerable cost of the peptides that would be required for this experiment. However, **we now include our colP results from the S>D mutants (in revised Fig. 4c).**

7) Fig 4- Are there any differences in the phospho-LRP6 or GSK3/Axin levels when co-IP'ed with LRP6 for the Axin P372A mutants?

We did not detect any significant differences in the levels of Axin or GSK3, nor in the patterns detected by the anti-pS1490 antibody, between wild-type and P372A lines (now **stated on p10, 2nd paragraph**). However, we would not expect to see any differences in these patterns between wild-type and mutant lines since the increase in the levels of Wnt signalling in the mutant lines is relatively small (~4x higher than in the wild-type line, as shown in revised Fig. 6d, compared to the >1000x increase seen in Axin DKO or GSK3 DKO mutants, as we now **point out a few lines further down in the same paragraph**. Indeed, the mild but consistent Wnt hyperactivity of the P372A lines is only measurable by SuperTOP because these assays are highly sensitive and quantitative (as we now **state on p9, 2nd paragraph**).

8) If the P372A blocked the termination and prolonged Wnt signaling (as was implied), one would predict that a brief activation by Wnt3a (followed by wash-out), would result in prolonged elevated β -catenin and phospho-LRP6 levels compared to controls.

To test this, we have performed wash-out experiments with wild-type and P372A cells and monitored recovery after varying times of Wnt stimulation. However, we found no differences in the recovery times between wild-type and mutant cells, even after short (1 hour) pulses of Wnt stimulation – almost certainly because the hyperactivity of the P372A lines is rather mild (see above, point 7). Clearly, it would take a lot more experimentation to get the precise conditions right under which a delayed shut-down of Wnt signalling could be observed in these mutants. Note though that we also observed a similar tendency to hyperactivity in another mutant (RR>D) now **shown in revised Fig. 6e**.

We have **rewritten our discussion of this point**, to avoid overstating our conclusions regarding the role of the multi-pronged interaction between Axin and GSK3 in presaging the termination of Wnt responses.

9) Fig 4b- the immunoblot of LRP6 pulled down with AP2 is barely visible

True: the relatively low affinity between AP2 and LRP6 (8 microM; see revised Fig. 7b) means that this interaction is challenging to detect by coIP. Nevertheless, the small difference between wild-type and mutant LRP6 is highly reproducible, and we now also **include the results from the single-motif mutants** (see revised Fig. 7c), to document this result more thoroughly.

10) *Fig 4- Loading controls are lacking for 4d and 4e*

We always use GSK3 as a loading control in these experiments (since the signal from actin is too strong), which we consider valid since we never observed any differences in the total levels of GSK3 in our experiments (please note that the Axin-associated pool of GSK3 dedicated to Wnt signalling is only a small fraction of the total cellular GSK3).

11) *The model, as presented in Figure 7, is confusing. Could the authors describe their model more clearly in terms of the transitions between the three states of Axin-GSK3 during Wnt signaling?*

We have **thoroughly revised our model cartoons** (as shown in revised Fig. 9) and **rewritten the legend of this figure**, hopefully to convey our hypotheses more clearly.

12) *In the discussion, the authors should go into more detail about their model and how it fits with the model from the He lab (Kim et al., 2013), proposing a “closed” and “open” conformation for Axin occurring via an intra-molecular interaction.*

The 'closed' and 'open' conformations of Axin proposed by He and colleagues in their 2013 paper refer to a separate intra-molecular interaction (in its C-terminus, between its beta-catenin-binding region and its DIX domain) although the transition between these two conformations is also regulated by phosphorylation. The two sets of intramolecular interactions are therefore likely to occur independently, during different phases of Wnt signalling, and to achieve different outcomes, as we now **state on p19 (top)**.

13) *Early work from the Wu lab showed that Axin interacts with LRP5 in a yeast two-hybrid (Mol Cell, 2001). This was followed by studies from the Wieschaus lab (Tolwinski et al., Dev Cell 2003) showing that full-length Axin did not interact with Arrow in a yeast-two hybrid and that “full-length Axin does not interact with Arrow because the N-terminal half of Axin prevents interaction with Arrow.” Both studies should be cited. Similarly, the paper by Cseleyi et al. (PNAS 2008) demonstrating that the intracellular domain of LRP6 could directly inhibit GSK3 activity using purified, recombinant proteins should be mentioned.*

The reference to the work by the Wu lab (Mao et al, Mol Cell 2001) was already cited in our original manuscript, but we have now also **added** Cselenyi et al (2008) as **reference 52 (p9)**. However, the Tolwinski et al (2003) study was entirely based on fly proteins which differ substantially from their vertebrate counterparts. For example, Hamada et al (1999) who discovered Drosophila Axin state in their paper that “D-Axin does not bind to Zw3/Shaggy/Drosophila GSK3beta” – in other words, it seems that D-Axin does not possess a GIR and its role in signalosome assembly may therefore differ from that of its vertebrate counterparts. Also, based on our biophysical data, we would expect full-length D-Axin to bind to Arrow in yeast cells, contrary to the findings of Tolwinski et al, which further supports the notion that the mechanism of signalosome assembly may be different in flies. The Tolwinsky study is therefore of limited relevance to our paper, and so we would prefer not to discuss it, given that our work is based entirely on human proteins – and rather challenging to absorb as it stands, because of the intrinsic complexity of the system.

Reviewer #2 (Remarks to the Author)

This is an exciting study that provides important new insights into the assembly mechanism of the Wnt signalosome, which forms upon Wnt stimulation. The authors demonstrate extensive cooperativity between AXIN, GSK3 and LRP6 in the assembly of the signalosome, from where they go to dissect the mechanisms underlying this cooperativity. The most exciting finding of the study is the direct, phospho-regulated interaction between AXIN and LRP5/6. The study goes a long way towards answering the long-standing question of how AXIN is recruited to LRP6. Binding between two non-domain portions of two different proteins, alongside a phosphorylation-controlled foldback mechanism, provides a precedent in the field. The work is technically excellent, covering a wide range of biophysical techniques and CRISPR editing for functional studies. Deciphering the binding events at the molecular level is technically challenging territory as the involved interactions are weak and dependent on avidity. NMR spectroscopy studies have proven extremely valuable in this context, and also in providing new insights into GSK3 binding to AXIN. By combining mechanistic and structural studies with proteomics, the authors further tie in the relevance of the endocytic machinery into signalosome function, confirming and expanding on previous findings. The work covers a lot of ground and opens up new ways of thinking about the intricacies of Wnt signalosome assembly and the complexities of regulatory events in Wnt/beta-catenin signalling. I am enthusiastic to see this work published. I also enjoyed the comprehensive Discussion section that ties together different aspects of signalosome regulation.

Specific comments

Introduction, line 8: Jesse Zalatan's group have done some interesting work showing how GSK3 activity is directed towards beta-catenin in the pathway, insulated from other substrates. Rather than just citing the review, could the authors consider referencing these studies?

We have **added** the Zalatan reference (Gavagan et al, 2023) **as reference 4 (p2)** as this study is indeed highly relevant in this context, but would also like to retain the reference to the Stamos & Weis review (reference 5), to acknowledge the immense body of work by the Weis lab in this area.

Can the authors indicate Lip-A4, A308-E384 in the schematic shown in Figure 2A? Can the LIDup be indicated as well?

We have **labelled both LIDup and LID α in Fig. 2a** (renamed LIRup and LIRalpha, see below), but we have deleted Lip-A4 altogether from our manuscript as this construct is redundant with Lip-A5 and Lip-A3 (given that these behave essentially the same, see our response to comment 1 by Reviewer 2).

p. 5, paragraph 3: Contrary to the text, which refers to all three Axin fragments in Figure 2b, only fragment A5 is shown in the figure. Can the authors show the equivalent data (BEST-TROSY spectra) for the different Axin fragments, if needed in the supplementary material? This will make it a lot easier for the reader to follow the narrative.

We now **include the bleach map from the BEST-TROSY spectrum for Lip-A5** (in **revised Supplementary Fig. 2d**).

Figure 2 and Supplementary Figure 2: Can the authors expand the panel labelling, please, to include the isotopic labelling of Lip-L1? (I.e., it is not immediately clear from Figure 2 that panel e is from ^{13}C labelled protein.)

We now **mention the isotopic labelling of Lip-L1 in the legends** of these figures (to avoid crowding in the individual bleach-map panels). Also, we **have inserted 'from 2D-CON'** in the **panel labelling** of Fig. 2e, to alert the reader to the fact that this bleach map is based on a recording capable of detecting proline-dependent interactions and therefore not directly comparable to the other bleach maps.

p. 5, last paragraph: The authors refer to an AlphaFold 2 prediction, but do not include a reference to a figure (Figure 3c). Can this be added?

Done (see **p6, 1st paragraph**).

p. 6, paragraph 2: ITC is not in Figure 2e, and I cannot find it elsewhere either. Can the authors also include the ITC data for GSK3 and Lip-L1? This is discussed, but without a figure. Similarly, ITC data referred to in the last paragraph on page 6 are not shown.

We originally stated the K_D values in the individual panels of Fig. 2e and 2g, but we have now deleted these and instead show the **corresponding ITC profiles** from which these values were derived in an **new main figure (Fig. 3)**.

Could the foldback model in Figure 2e be tested by DLS or SEC-MALS? Foldback would be predicted to reduce the hydrodynamic radius. Testing the phospho-regulated interaction with LRP6 biophysically would also be interesting.

We cannot do these tests since we would need the full-length peptide (with and without phosphorylations on all three serines), but our peptide company was unable to supply the fully phosphorylated version of the full-length peptide (Axin308-366). Quite apart from this, we doubt that it would be possible to detect a phosphorylation-dependent difference in the hydrodynamic radius of the LIR peptide by these methods because of its rather small size.

p. 7, paragraph 2: Can the authors add that the data are from ITC? This is of course clear from the figure but not the text.

We now **state that these data are derived from ITC** (see **p7, bottom**).

Figure 4a: Can the helices be labelled for clarity? Does the sequence region in red correspond to the extended alpha-helical region within GID referred to on p. 7, paragraph 3? This could be pointed out.

We have now **labelled these helices**. This should make it clearer that the red sequences in revised Fig. 5a correspond to the minimal GSK-interacting region of Axin as seen in the co-crystal structures (as mentioned in the legend for this panel).

For the AlphaFold predictions, can the authors please include the pLDDT and PAE plots in the supplementary material? How similar were the top-5 models for both AF2 and AF3? Does the predicted AF3 model have worse confidence metrics?

We now **show the pLDDT and PAE plots** for both models (in the **new Supplementary Fig. 4**). In each case, the top three models are essentially the same, and although the

pose of LIRup is predicted with higher confidence by AF3 than by AF2, our functional validation is more consistent with the AF2 rather than the AF3 model. However, we now **emphasise** (on **p9, bottom** and **p10, top**) that both models predict a multi-pronged interaction between GSK3 and Axin (mediated by GIR, PRTxR and LIRalpha) which is incompatible with simultaneous binding of GSK3 to LRP6.

The reporter assays in Figures 4e and e suggest an antagonism between GSK3 and LRP6 for AXIN binding. Can this be tested directly by interaction assays?

This would be challenging since the foldback of Axin (which opposes its binding to LRP6 while remaining compatible with GSK3 binding) depends on phosphorylation of LIRup.

Figure 5a: Can the authors expand the figure legend to indicate which data the signalling activities referred to in the figure correspond to? I cannot see data for deltaB, for example.

We now **show the corresponding signalling assays** (including the data for ΔB) in the **revised Supplementary Fig. 9a**.

Figure 5c: Does mutation of each of the single tandem motifs in the GFP fusion affect interaction? If yes, this would demonstrate functionality of the atypical motif at the level of binding. This would nicely complement Figure 5f.

This is indeed the case, as we now **show in revised Fig. 7c**.

Panels d and e of Figure 5 are very small. The structural findings could be presented more prominently. Some details (such as the map corresponding to the peptide ligand bound to the protein) could be included in the supplementary material.

We now show electron density maps for each structure of AP2 bound to its cognate YxxPhi peptide from LRP6 in the **new Supplementary Fig. 8**.

p. 10, paragraph 2: Can you explain the LRP6m10 mutation? It is not part of panel 5a.

We now **refer to Fig. 1a** in which this mutant is shown and also describe it explicitly when we first use it as a control for our colP experiments (on **p5, top**).

p. 11, paragraph 1: The authors may want to change the term "near-physiological" to something more specific as it may mean different things to different readers.

We have **revised** this sentence to 'We conclude that the tandem Y-motifs are critical for Wnt-dependent signaling of LRP6 in a complementation assay based on LRP5/6 DKO cells' (**p13, 3rd paragraph**).

p. 14, penultimate paragraph: Given the high degree of intrinsic disorder, I am not so sure a spatially confined self-interaction between adjacent elements in AXIN would suffice to place the RGS and GID closer together. This point could be de-emphasised.

Fair point: we have now **deleted** this sentence.

Minor points

p. 6, paragraph 3: The term "LIDup" is introduced earlier, but not used to refer to "the

sequences upstream of LIDalpha". This made me wonder whether the description refers to LIDup or a different region. Can names be used more consistently?

We do indeed refer to LIRup, as now **stated in the revised text (p6, 3rd paragraph)**.

The nomenclature of mutant variants for Figure 5f appears inconsistent between the text and the figure. (YYYY>A in the text has no equivalent in the figure.)

Thanks for pointing this out: we are now using **consistent nomenclature** in the figures and text (YxxY>A and YxxF>A for single-motif mutants, and YYYY>A for the double-motif mutant).

Figures 4 and 5, and supplementary material: Where the authors show western blots accompanying reporter assays, it may be worth labelling the Western blots separately for clarity as alignment with the bar graphs is not always possible.

We have now **labelled** the lanes of the blots in the supplementary figures. To avoid crowding in the main figures, we have **added -/+ Wnt** above the lanes in the blots, which helps their alignment with the graphs considerably.

Can the authors indicate molecular weight markers for all Western blots?

Done

Could the authors consider finding different names for LID, GID and BID as none of these are actually "domains"?

This is a valid point, especially in the case of LID. We have now **changed the nomenclature** of these elements to **LIR** and **GIR** (for 'interacting region'). We have also deleted the abbreviation BID in the text but retain this abbreviation in the figures (and explain it in the corresponding legends).

Omitting the "Lip" pre-fix from construct names may help with simplifying nomenclature as all constructs share the same tag. This would also help with making the different construct names more recognisable. At the moment, due to the shared tag reference, the names are almost identical.

We would prefer to retain the 'Lip' prefix since GSK3 is not Lip-tagged. However, we have **added** 'LRP6' to 'Lip-L1' and 'Lip-L2', and 'Axin' to 'Lip-A3' and 'Lip-A5' as subscripts, **to clarify which protein** these abbreviations refer to.

In general, the authors could expand their explanations of the NMR experiments to make them more accessible to readers not familiar with looking at this type of data.

We have **inserted an additional sentence** to clarify that 'bleaching' of an individual resonance provides evidence that the corresponding residue partakes in an interaction. We also use the term 'bleaching' consistently throughout, explaining when we first use the term (on **p5, 3rd paragraph**) that it refers to line broadening of an individual peak (as 'bleaching' is more intuitive).

The order of the figures means there is a lot of jumping back and forth when reading. Could this be improved?

We have thought long and hard about the order in which to present our data and believe that the current order is optimal. Cross-referring between figures should be much easier in a journal layout where the figures are interspersed with the text (and the legends are underneath the figures).

p. 8, paragraph 1: "pulse-chase experiments that the turnover"  "pulse-chase experiments showing that the turnover"

Done

p. 9, paragraph 2: "doable"  "feasible"

Done

Reviewer #3 (Remarks to the Author)

A critical step for Wnt/ β -catenin pathway activation is the recruitment of AXIN1 and its associated kinases GSK3 and CK1 to the Wnt receptor complex at the plasma membrane, after which a sequence of molecular events mediate inhibition of the GSK3 kinase. In this study, Gammons et al aim to shed light on the poorly understood question of how the AXIN-GSK3 complex is recruited to the cytosolic tail of LRP6. They use NMR, biochemical methods, structural modeling and CRISPR-mediated gene editing to uncover that an LRP6 region incorporating the membrane-proximal PPPSPxS motif interacts with both AXIN1 and GSK3. The LRP6-binding site in AXIN1 is mapped to the intrinsically disordered (IDR) region, to a sequence N-terminal to the GSK3-interaction domain. Interestingly, access of this AXIN1 IDR region (AXIN1-LID) for LRP6 binding appears regulated by phosphorylation. Convincing data are shown that phosphorylation of AXIN1-LID induces a back-folded conformation which makes this site inaccessible for LRP6 binding and thus prevents inappropriate Wnt pathway activation in unstimulated cells. In the last part of the manuscript, the authors anticipate that clustering of LRP6 tails by interaction with the clathrin adaptor AP2 is essential for AXIN and GSK3 binding cooperativity and WNT pathway activation.

The work describes a number of interesting new findings that shed light on a poorly resolved issue in the field of WNT/ β -catenin signaling. The combination of technologies is a strong point. The overall model that is presented however needs more convincing experimental support.

Also, the manuscript is not an easy read, partly due to a lack of explanation of the rationale and technologies and the complex names of motifs and regions. It needs editing to make this work more accessible to a broad readership including newcomers to the field.

Major points:

A major concern is that, due to protein expression problems, focus has been placed on a narrow region of the AXIN1 IDR. Since IDRs often use discontinuous motifs for interactions with partners, a risk is that the work may not have identified all possible LRP6-interaction sites. Can the authors exclude a role of other parts of AXIN1 in facilitating LRP6 binding?

We cannot rule out that additional Axin sequences help Axin to bind to LRP6. However, we consider this unlikely, given the moderately high (<1 microM) affinity of the minimal LID (now called LIR for 'LRP6-interacting region', in response to a comment made by reviewer 2) for LRP6: if this affinity were just a few fold higher, this would be in the same order of magnitude as the typical cellular concentration of Axin. It would therefore enable Axin to bind to LRP6 in the absence of Wnt, which is clearly undesirable. In support of this argument, the affinity between Axin and LRP6 cannot be much higher than low micromolar, given how challenging it is to detect the association between overexpressed Axin and LRP6 by coIP assays. Consistent with this, we determined by ITC that the affinity between the minimal LIR and LRP6_L1 proteins is <1 microM. Finally, we would like to emphasise that, although the LIR is intrinsically unstructured, it does not consist of discontinuous motifs: rather, it spans a labile alpha-helix (LIRalpha) and its immediate upstream flanking sequences (LIRup) that can fold back on LIRalpha – in other words, Axin LIR is a discrete module.

Page 4: '...efficient co-IP between the three proteins is only seen with their wild-type (wt) versions, but not with catalytically-dead GSK3 (K85R) nor with the non-phosphorylatable mutant LRP6m10'. Why did the authors choose to use K85R for detecting NMR binding in later experiments? The rationale for this is not clear (since this GSK3 version does not bind in in co-IP) and not commented on.

We have now **clarified** (on **p5, 3rd paragraph**) that wt and catalytically-dead GSK3 produce essentially the same bleach map (compare **Fig. 2d** and **Supplementary Fig. 2c**), although the perturbations are slightly stronger with the mutant (possibly due to substrate trapping) which is why we have used it for some NMR experiments. Note also that these NMR assays monitor the *direct* binding of GSK3 to LRP6 in vitro (without any phosphorylation) whose affinity is ~100x below the detection limit of coIP assays in cells. By contrast, the coIP assays monitor robust association of the Axin-GSK3 complex with LRP6 in cells, which requires a far higher affinity. This enhanced affinity is provided partly by the phosphorylation of LRP6 at S1490, and partly by GSK3's co-operation with Axin (as shown in Fig 1).

Page 5: 'Phosphorylation at S1490 can only be detected if the wt proteins are co-expressed' (Figure 1b). How do the authors reach this conclusion? The phospho-blot of the IP shown on the right shows strong signal of S1490 in all lanes.

This sentence was indeed incorrect: thanks for pointing this out! It should have read 'hyperphosphorylation of the LRP6 ctail, as manifest by an upshift of the anti-pS1490 signal on a Western blot, can only be detected...'. In other words, this antibody reveals not only phosphorylation of S1490, but also (indirectly) phosphorylation of other sites in the LRP6 ctail, which results in a 'hyper-shift' of the signal. However, since this represents extraneous information that is somewhat complicated to explain, we decided to **remove the anti-pS1490 lanes** from this figure. The simplified Fig. 1 is now much easier to understand and conveys more crisply that the robust association of Axin and GSK3 with LRP6 is only seen if wild-type versions of each protein are expressed. We have also added **additional titration experiments** (in **revised Supplementary Fig. 1**), to document our conclusion from these experiments even more compellingly.

Also Figures 2 and Suppl Fig 2 shown NMR-based LRP6 interaction data for Lip-A5 mutants, but bleach maps for wild-type Lip-A5 are missing. These data should be included for comparison.

We now show the **bleach map** from the BEST-TROSY spectrum for **Lip-A5** (in **revised Supplementary Fig. 2d**).

The identified region in LRP6 that accommodates both Axin and GSK3 binding is interesting and supported by convincing data. The LRP6 interface is indicated as the ‘proximal PPPSPxS motif’ (e.g. in the abstract) but this does not seem to be entirely correct. NMR data indicate that AXIN1-based fragments indeed bind this motif as well as flanking regions on both sides. For GSK3, however, the PPPSPxS motif seems to participate less in the interaction as compared to flanking regions. This observation needs better specification in the abstract and discussion of the results. In particular, since the phosphorylated version of the motif must bind GSK3. How these observations correlate is not entirely clear.

Monitoring binding of the proline residues in the PPPSPAT motif requires recording 2D-CON spectra (as **explained at the top of p6**; we have also inserted **‘from 2D-CON’ into the panel labelling** of Fig. 2e, to alert the reader to the fact that this bleach map is not directly comparable to the other bleach maps based on BEST-TROSY spectra). However, we were only able to record these spectra for Axin but not for GSK3, likely because its binding to LRP6 is too weak, as we now **explain on p6 (top)**. Direct comparison between the Axin and GSK3 binding sites therefore must be based on their bleach maps derived from BEST-TROSY spectra, which look very similar (Fig. 2c and d). This supports our conclusion that Axin and GSK3 bind essentially to the same segment of LRP6 (with similar limits), as we **conclude on p5 (3rd paragraph)**.

However, the relevant sentence in the Abstract was indeed inaccurate, and so we have inserted **‘...and its flanking sequences’** into this sentence. We have also ensured that, throughout our manuscript, we only use the term ‘motif’ when we refer to the PPPSPAT sequence itself, and that we use the term ‘PPPSPAT element’ when we refer to the Axin- or GSK3-binding site of LRP6.

AXIN-GSK3 interaction models generated by AF2 and AF3 are not in agreement, which is a weak point of the presented work. The authors take the AF2 model as best fitting, since it is in agreement with most of their data. Alternative explanations are however not considered. Furthermore, any information on estimated confidence of the predicted models is missing. For all AF modeling, ipTM values must be indicated. If ipTM values are below 0.5-0.6, the model is considered unreliable. Any AF data that do not meet this threshold for prediction quality should be removed.

We are grateful to this reviewer for emphasising this point! This has prompted us to look more carefully into the differences between the two models, and although this has not changed our conclusions (as they were based on our in vivo validation of P372 in genetically manipulated cells), we have now **describe the differences between the AF2 and AF models more clearly (bottom of p8 and top of p9)**. We also document these differences by **showing the pLDDT and PAE plots** for both models (see **new Supplementary Fig. 4**). Importantly, both models agree with our experimental NMR data, showing that the binding between Axin and GSK3 in solution is more extensive than previously thought (extending from GIR across PRTxR and LIRalpha) and that both models place the PRTxR motif across the catalytic pocket of GSK3 (as now **emphasised at the bottom of p8**) which is incompatible with simultaneous binding of this pocket to LRP6. We have also **removed the term ‘horseshoe complex’** (from the text and the model figure) and **replaced it with ‘multi-pronged interaction’** (or **‘multi-pronged complex’**), to encapsulate the common key feature of the two predicted models. Finally,

we also consider the possibility that Axin may oscillate between different poses (**p9, 1st paragraph**).

Indeed, we have also made extensive use of the capability of AF3 to model phosphorylation-dependent structures (see **new Supplementary Fig. 12**). This has generated interesting insights and hypotheses that we believe to be relevant for the in vivo situation, as discussed on **p18 (1st paragraph)** and **p19 (2nd paragraph)** and illustrated in our **revised model Fig. 9**.

Figure 4: The hyperinducible phenotype of AXIN1-P372 lines is interesting and supports the model that this mutant does not interfere with destruction complex activity but displays improved interaction with LRP6 (the latter event being induced by Wnt binding to the receptors). It is essential however to better substantiate these findings in cells at the endogenous level: do the indicated mutants (P372, RR>D, DED3) indeed display increased/decreased binding to LRP6 upon Wnt pathway activation?

Unfortunately, the relatively low affinity between Axin and LRP6 (1 microM; revised Fig. 3a) makes it challenging to detect this interaction by coIP even if the proteins are co-overexpressed, and impossible to detect it between endogenous proteins by this method. While we would have loved to do this, it is unfortunately not feasible.

Previous findings indicated that GSK3 and Axin associate preferentially with LRP6 upon phosphorylation of its PPPSPxP motifs. In fact, in the summarizing model, GSK3 binds to a phospho-site in LRP6, but the interaction mapping in the current manuscript do not include phospho-forms of LRP6. How do the authors factor in phosphorylation in their interaction model, would this influence relative binding affinity of GSK3 and affect interactions with AXIN?

We had initially planned to measure the affinities of Axin for the phosphorylated PPPSPAT element and compare this to the unphosphorylated one. However, to generate a stoichiometrically (fully) phosphorylated element, we needed a synthetic peptide, but the company that provided our peptides was unable to generate a soluble PPPSPAT peptide (with or without phosphorylations) because its flanking sequences are highly hydrophobic. We therefore could not address the question how phosphorylation of the PPPSPAT element affects its affinity to Axin. However, based on extensive evidence in the literature, it is likely that the phosphorylation of S1490 would increase the affinity of GSK3 and Axin to LRP6 (this phosphorylation is certainly essential for the competitive inhibition of GSK3 by PPPSPxS motifs, as shown by the Weis group). This may also explain why phosphorylation strengthens the association of Axin and GSK3 with LRP6 in cells, as shown compellingly by the He group in several of their studies. Indeed, it is supported by our proximity-labelling data which show a consistent dependence of Axin and GSK3 on the unphosphorylatable LRP6m10 mutant – particularly pronounced in the case of GSK3 (see **revised Fig. 8f**).

However, while GSK3 is thought to be the kinase phosphorylating PPPSPAT at S1490, it seems implausible that this kinase would catalyse the initial phosphorylation at this site (to prime subsequent Wnt responses), given its exceedingly low affinity for the PPPSPAT element. We consider it more likely that this ‘priming’ phosphorylation is catalysed by cyclin Y-activated CDKs, as shown by the Niehrs group in their compelling study (Davidson et al, 2009). **This alternative possibility is now mentioned in the 2nd paragraph of the Introduction (citing reference 32) and fully addressed in the Discussion (p19, bottom and p20, first two paragraphs).** Furthermore, our finding of CDK14 by TurboID (**revised**

Fig. 8f) and the high Kinase Library score for S1490 as a target site for CDKs (**new Supplementary Fig. 11**) provide further support for this notion, as mentioned on **p15 (1st paragraph)**.

Western blots for IP experiments need to be quantified.

We have now **quantified the colP** signals in our **main figures** (except for one, see below) and have added the corresponding values underneath the blots. We also state the **number of independent experiments** in the text where appropriate (e.g. on **p5, top**) and in the relevant **figure legends**.

However, we have not been able to use densitometry to quantify the blots documenting the cooperation between Axin and GSK3 (Fig. 1b; Supplementary Fig. 1) since the colP signals for wt versus mutant proteins are outside the linear range (i.e. even overexposed blots do not reveal a colP signal between mutant proteins). In other words, these signals are pretty much 'black or white', providing 'yes/no' answers. Rather than assigning somewhat artificial numbers to these signals, we thought it preferable to corroborate this conclusion under different conditions, and we therefore decided to show **additional titration experiments** (in revised **Supplementary Fig. 1**).

Minor points:

Page 3: Make clear that motifs within LRP6 are indicated as 'motif A', 'motif B' etc

Done (see revised text on **p3, 2nd paragraph**)

"This supports a model that envisages clustering of LRP6 in clathrin-coated (CC) structures upon Wnt stimulation by AP2," Sentence is multi-interpretable and therefore confusing

We have now **fixed this sentence** (on **p4, 2nd paragraph**).

The naming of motifs and regions makes the manuscript a very difficult read (Lip, LID, GID, A3, LIDa, LiDup, L1). Please keep the protein name always included in these names (e.g. AXIN1-LID, LRP6-...) preferably including aa boundaries, for clarity throughout the manuscript.

We have now **replaced** Lip-L1 and Lip-L2 with **Lip-L1_{LRP6}** and **Lip-L2_{LRP6}**, and Lip-A3 and Lip-A5 with **Lip-A3_{Axin}** and **Lip-A5_{Axin}**, respectively, and we have also **inserted 'Axin'** before each mention of GIR, LIRalpha and LIRup in a new sentence. The amino acid boundaries are always specified when these terms are first defined (e.g. **p5, 2nd paragraph**).

Page 5: 'To be able to observe Axin-dependent attenuation of the proline residues within PPPSPAT...' Reads odd, this should be 'peak attenuation' or 'signaling attenuation'

We have now **modified this sentence** (see **p6, top**). We have also substituted 'line broadening' or 'peak attenuation' with 'bleaching' throughout our text (after specifying this when we first use the term on **p5, 3rd paragraph**), as this term is less technical and more intuitive to readers unfamiliar with NMR.

Page 5: Lip-A4 is mentioned but not shown in any of the figures

We decided to **delete Lip-A4** from our manuscript, given that this fragment is only marginally shorter than Lip-A5 and produces essentially the same bleach map as the latter. The results from this fragment are therefore redundant with those from the minimal Lip-A5 fragment.

Page 6: '13C-detected 2D-CON spectra' – provide rationale for using this method, to accommodate a broad readership

We have now inserted a sentence (on **p6, top**) explaining that we used this approach because proline residues are invisible in BEST-TROSY recordings.

Page 6: 'We therefore incubated an 15N-labeled distal fragment (Lip-L2) with Lip-A5, which....' The figure (Suppl Fig 3) shows Lip-A3.

Apologies: now **corrected**.

Page 8: describe the predicted role of Proline in the structure first and then discuss the Pro-mutant.

We have rewritten this passage, introducing the evolutionary conservation of this proline and its predicted pose in the AF2 model on **p10 (2nd paragraph)** before describing the mutant phenotype of P372A (see also the **last paragraph on p8** which compares the positions of the PRTxR motif in the two AF models).

Page 8: 'the AXIN2-ko line' – needs better introduction

We have **inserted a sentence** (on **p9, 2nd paragraph**) explaining that the Wnt response is unaffected in the Axin2 KO line as wt Axin1 fully compensates for the loss of Axin2 in this line.

Page 8: 'Assessing Wnt responses of two independent lines...' Please indicate every time which line you talk about, to avoid confusion. In this case the AXIN1 P372-mutant lines are compared with WT.

Done (i.e. we now state the name of the mutant line each time we describe the results obtained with these mutant cell lines; e.g. on **p9, 2nd paragraph**).

Page 8: indicate which subfigure of Suppl fig 4 you refer to

Done (indeed, we now specify the sub-panel of supplementary figures throughout the text whenever appropriate).

Page 9: more clearly explain that the YYYYF carries two mutated motifs

We have now **modified the nomenclature** of these mutants (see **p12, top**) and made it clear whether we refer to **single- or double-motif mutants**.

Page 10: 'Indeed, wt 2xTF but not its YYYYF>A mutant version...' Should this not be YFYF in the case of the transferrin receptor?

Indeed! We have **corrected** this (**p12, bottom**).

Page 11: 'The thus identified consistently...' sentence is flawed

We have fixed this sentence.

Page 14: 'CC structures' needs better explanation

We have revised this text to emphasise that these clathrin-coated structures are larger and more stable than the short-lived clathrin-coated pits that earmark membrane proteins for rapid endocytosis (**p16, bottom paragraph**). We also use the term '**clathrin locales**' throughout our manuscript, to emphasise the localisation rather than the structural aspect of these clathrin features.

Suppl fig 1b: make clear whether GSK3 is co-expressed in figure and legend.

This figure has been thoroughly **revised**, in response to comments raised by reviewer 1.

Page 9: 'LRP6 and Frizzled are subject to clathrin-dependent endocytosis and lysosomal degradation in the absence of Wnt, following earmarking by Dishevelled and the transmembrane ubiquitin ligase ZNRF3/RNF43, which appears to rely on an AP2-binding motif in Dishevelled and on the tandem Y-motifs in LRP6'

This statement suggests that ZNRF3/RNF43 are involved in AP-2 interactions. This has not been shown, please adapt.

We have now rephrased this sentence and split it into two (**p11, 2nd paragraph**).

Reviewer #4 (Remarks to the Author):

The study by Gammons et al. addresses a highly relevant and timely topic in the field of Wnt signaling, focusing on the conformational flexibility of Axin and its interaction with both Lrp6 and the AP2 clathrin adaptor. The Wnt pathway is crucial for cellular signaling processes related to development and disease, and understanding the assembly of the Wnt signalosome is a significant goal for the field. While this paper provides valuable insights into the dynamics of Axin and its interactions, the manuscript suffers from conceptual and methodological weaknesses that need to be addressed.

The research is ambitious and presents important observations, particularly in mapping the Axin-Lrp6 interaction and highlighting the competition between Axin and GSK3 on Lrp6. However, there are several critical issues, especially regarding the overall data interpretation, and the clarity of the presented model.

Strengths:

1. Clarification of Axin and Lrp6 Interactions - The authors make substantial progress in understanding the interplay between Axin and Lrp6, which is fundamental to Wnt signaling. The detailed mapping of these interactions is a key strength of the study and adds new information to the field. This is a timely contribution, given the current focus on the structural basis of Wnt signalosome assembly.

2. Competition of Axin and GSK3 - The study highlights how Axin competes with GSK3 for interaction sites on Lrp6, offering insights into how the Wnt signalosome dynamically regulates its components.

Major Concerns/Weaknesses

1. Disconnection Between Axin-Lrp6 and AP2 parts - A major concern is that the paper feels like two distinct stories: the detailed mapping of Axin-Lrp6 interactions, and the linkage to AP2 and clathrin. The transition between these sections is not well articulated, and the mechanistic connection between Axin-Lrp6 interactions and AP2 is unclear. Without a stronger integration of these two aspects, the paper seems fragmented, as if two incomplete studies have been combined into one. The authors need to more convincingly demonstrate how the findings regarding Axin-Lrp6 feed directly into the role of AP2 in the assembly of the Wnt signalosome. At present, the paper gives the impression of being two separate research efforts, leaving readers to infer the link between the two.

We did initially consider publishing the two parts separately. However, we decided against this when we realised that the AP2-dependent clustering of LRP6 could provide a mechanistic explanation for the observed cooperativity between GSK3 and Axin in their binding to LRP6 even though they bind to the same site: as we propose in our model (**revised Fig. 9**) and corresponding legend (now **modified to make this clearer**), the clustering of LRP6 allows them to bind simultaneously to their cognate sites in nearby tails within the LRP6 clusters. In other words, without the AP2 part, we would not be able to provide an explanation for the observed ‘trans’ cooperativity between Axin and GSK3 in their binding to LRP6 – as appreciated by the other reviewers.

2. Overinterpretation of the Data - The proposed model, while interesting, appears overinterpreted in its key parts. Although the data align with the proposed model, the authors stretch the conclusions beyond what the experimental evidence strongly supports.

- In particular, Fig. 4b/c presents a proposed binding mode that lacks support from AlphaFold’s newer versions.

We now **compare the two AlphaFold models** (AF2 and AF3) **side by side** (at the **bottom of p8 and top of p9**; see also **revised Fig. 5b-j and new Supplementary Fig. 4**) and explain why the AF2 model appears more compatible than the AF3 model with our in vivo validation data based on the P372A mutant (now shown in **revised Fig. 6**). We also emphasise that both models agree with our experimental NMR data, showing that the binding between Axin and GSK3 in solution is more extensive than previously thought (extending from GIR across PRTxR and LIRalpha) and that both models place the PRTxR motif across the catalytic pocket of GSK3 (**p8, bottom**) which is incompatible with simultaneous binding of this pocket to LRP6. But we have now replaced the term ‘horseshoe complex’ with ‘**multi-pronged interaction**’ (or ‘**multi-pronged complex**’), to encapsulate the common key feature of the two predicted models. Finally, we also consider the possibility (on **p9, 1st paragraph**) that Axin may oscillate between different poses in its multi-pronged interaction with GSK3.

Of note, we have also made extensive use of the capability of AF3 to model phosphorylation-dependent structures (see **new Supplementary Fig. 12**). This has generated interesting insights and hypotheses that we believe to be relevant for the in vivo situation, as discussed on **p18 (1st paragraph)** and **p19 (2nd paragraph)** and illustrated in our **revised model Fig. 9**.

Additionally, the use of motif C in the model, when motif A is discussed in other sections, creates confusion about which motif is functionally relevant.

In their 2014 paper, the Weis lab presented the structures of GSK3 bound to motifs C or E, which show almost identical poses of these motifs. It is therefore plausible that motif A

(within the PPPSPAT element) would also bind to GSK3 in the same way. However, we have **changed this panel (now Fig. 5h) and its labelling** which now shows the first 5 residues of motif E (PPPSP - identical in motif A), to avoid confusion.

This is the key component of the model and the axin foldback interaction with GSK3 needs to be proven experimentally.

Perhaps we misunderstood this point, but this interaction between Axin and GSK3 is primarily based on our NMR data (Fig. 5a) showing that the interaction is far more extensive than previously thought based on crystal structures of Axin-GSK3 complexes. Furthermore, we also provide in vivo validation for the predicted AF2 pose (shown in **revised Fig. 5b**), by showing that mutating P372 (which is crucial for placing the PRTxR motif across the catalytic pocket of GSK3 in the AF2 but not in the AF3 model; see **revised Fig. 5b, c**) in endogenous Axin1 increases the Wnt signalling levels of the P372A mutant cells (see **revised Fig. 6d**). We also obtained a similar result from a double-mutation of the two arginines (RR>D) that contribute to the multi-pronged interaction between GSK3 and Axin, although in this case, the result does not distinguish between the two models, given that Axin is predicted to interact with GSK3 through the same LIRalpha surface in both models (**revised Figs. 5d-g**).

Indeed, the key results underpinning our model (now shown in **revised Fig. 9**) are (i) the discovery that Axin and GSK3 bind to the same site within LRP6 yet need to cooperate in their binding to this co-receptor, and (ii) that this cooperativity might be provided in trans by the AP2-dependent clustering of LRP6 in clathrin locales (as inferred from the dependence of signalling by LRP6 on its AP2-binding sites) since this enables Axin and GSK3 to bind to their cognate sites on nearby LRP6 molecules in these locales, and (iii) that the interaction between Axin and LRP6 is regulated by phosphorylation of LIRup which promotes a foldback interaction with LIRalpha which is incompatible with simultaneous binding of LIR to LRP6 (and which may guard against fortuitous binding of Axin to LRP6 in the absence of Wnt). We have now **revised our model in Fig. 9 and rewritten its legend**, hopefully to convey these points more clearly.

Further, in my opinion the data in Fig. 3 c/d shall be improved by analysis of the functionally deficient Axin mutants and by the evidence of the electrostatic nature of the interaction.

We now demonstrate that alanine substitutions of the **three basic residues in LIR α (RKR>A) essentially obliterate the binding to phosphorylated LIRup peptide** (now shown in **revised Fig. 4f**), as expected. This identifies the 'concave' surface of LIRalpha (blue in both AF2 and AF3 models, shown as electrostatic surface representations in **revised Fig. 4d, e**) as the surface mediating the intramolecular interaction with phosphorylated LIRup, consistent with the phospho-dependent foldback model shown in revised Fig. 4g. It also supports the notion of an electrostatic basis for this foldback within LIR (which appears incompatible with simultaneous binding of Axin to LRP6, as indicated by our in vivo validation studies).

- Another issue arises in Fig. 2h, where the sequence shown corresponds to the mutant YYYYF, which is only introduced later in Fig. 5. This adds to the confusion in interpreting the experimental results.

We were compelled to mutate these four aromatic residues (preceding motif B) to alanine, to be able to express and purify sufficient quantities of soluble well-behaved Lip-L1LRP6 for

use in NMR binding experiments. The fact that these aromatics serve as AP2-binding sites in cells does not affect the results from in vitro binding assays, especially since neither Axin nor GSK3 binds to any of the sequences in or upstream of motif B (as shown in Fig. 2 & Supplementary Fig. 2).

- Moreover, the proteomics data are presented in a somewhat disjointed manner. Different variants of the BioID assay are used for different mutants, but the rationale for these choices is not clearly explained. This undermines the consistency of the data and makes it difficult to evaluate the impact of the mutations and link.

We are grateful for this comment and agree with this reviewer that this was the weakest part in our initial submission of our study. It prompted us to reanalyse our first TurboID experiment (conducted in 2020, during the height of the pandemic and therefore never properly analysed and discussed) whose results we now include (in **new Fig. 8f, h**). This has increased our confidence in our TurboID hits, and we have therefore been able to extract more information from these TurboID experiments. For example, they have allowed us to identify high-confidence kinases whose association with LRP6 is sensitive to deltaB (**new Fig. 8g**) or, in the case of TTK, also to YYYYF>A (revised Fig. 8a, b). This has shaped our discussion (rewritten in parts, e.g. **p17, 3rd paragraph; p18 top**) and model (redrawn to incorporate LRP6-proximal kinases that had already been linked to Wnt signalling and LRP6 in previous work; **revised Fig. 9**). Of note, the kinase aspects of our model was also **bolstered** by our use of the Kinase Library (a fantastically useful resource that recently published by the Cantley group; see **new reference 46**) which allowed us to predict kinases for phospho-acceptor sites in Axin and LRP6 with key roles in Wnt signalling, as shown by previous work (e.g. by He and colleagues for S1490) or by our current study. For example, this has prompted us to question whether GSK3 is the physiologically relevant kinase phosphorylating S1490 as commonly thought, as we now discuss on **p19 (bottom)** and **p20 (top)**.

In response to this **criticism**, we have **revised our text describing the proximity labelling experiments**, spelling out more clearly the rationale for these experiments and for our choices of mutant baits in the BioID and TurboID experiments (**p13, bottom; p14, bottom; p15, top**). In a nutshell, these experiments were primarily designed to obtain evidence for an interaction between the Wnt signalosome and the clathrin adaptor machinery (which we achieved by using the efficient TurboID baits; see **new Fig. 8h**), but also to confirm other predictions from our in vitro work (e.g. we found that the association of GSK3 and Axin with the LRP6 TurboID bait is m10-sensitive, indicating that their binding to LRP6 is enhanced by, or depends on, phosphorylation of S1490 in the PPPSPAT element; see **new Fig. 8f**).

3. Challenges in Following the Schematics and Figures - Many of the figures, particularly Fig. 2, are difficult to interpret without extensive cross-referencing with the text. This detracts from the overall clarity of the manuscript. For instance, the term "Lip" in Fig. 2 is confusing and should be replaced with more descriptive labels like "Lrp6 (L1)" and "Axin1 (A3)" to facilitate understanding. Readers unfamiliar with the proteins will struggle to follow the experiments without digging into the manuscript for explanations.

We now consistently use the terms **Lip-L1_{LRP6}** and **Lip-A3_{Axin}** throughout our manuscript (first on **p5, 2nd paragraph**), to make it clearer which fragment is derived from which protein. Lip is simply a tag, which we decided to keep in the names, to make it clear which of these proteins were tagged.

- *Adding sequential information (e.g., amino acid positions) at boundaries or motifs in the figures would significantly enhance their clarity. The study works with four different proteins – Axin, Lrp6, GSK3 and AP2 – and it was a real challenge for the reviewer to follow the text/figures. In its current form, the schematics demand a deep familiarity with the protein constructs used in the study, which limits accessibility to a broader readership.*

We systematically specify the boundaries of each fragment when it is first introduced and defined (e.g. **p5, 2nd paragraph**). We are aware that our study is demanding to absorb because of the intrinsic complexity of the system, but we hope that, when the figures are shown with their legends and embedded in the relevant text, it will be easier for the reader to follow the narrative. We have also **explained more clearly in our revised text and figure legends** what was done, and why.

Minor Concerns:

- *The figure legends require more detail to clearly explain what is being shown. As it stands, the figures are not self-explanatory, and clearer labels should be used for the different constructs and proteins.*

We have **revised the figure legends** to provide more clarity and explanations where needed, and generally to make them self-explanatory (incl the legend of the **revised Fig. 9**, which now explains the proposed model).

- *Throughout the paper, the naming of the protein constructs could be more consistent. Given the complexity of the interactions and the number of different proteins and mutants tested, clearer and more standardized labeling would help readers keep track of the experimental components*

When revising our manuscript, we have taken care to ensure **consistent use of nomenclature**, and to **clearly name** (and label) individual proteins in our main text, figures and their legends, to make the work more accessible to the reader.

Conclusion:

This manuscript addresses an important and timely topic in Wnt signaling, providing new insights into the conformational flexibility of Axin and its interaction with Lrp6 and link to AP2. However, there are significant conceptual and methodological issues that need to be resolved. The current version of the manuscript presents two premature stories that feel artificially merged into one model. In its current form, the work does not sufficiently connect the findings from the Axin-Lrp6 mapping with the role of AP2 in the assembly of the Wnt signalosome. To address these concerns, I recommend either filling the gaps in the data to strengthen the proposed model or treating each of the two stories separately to ensure that each one is complete on its own. The paper would also benefit from a clearer and more conservative interpretation of the data, as well as improvements in the clarity and accessibility of the figures.

Reviewer #5 (Remarks to the Author):

Gammons et al report important new insights into the interactions governing Wnt signalosome assembly. Their data, drawing on a range of methods, provide compelling evidence for (i) a novel site in Axin they term the LID which can mediate direct interaction with LRP6 (ii) a mechanism for conformational occlusion of this LID site (antagonism of

LRP6 binding) triggered on phosphorylation by GSK3 and (iii) a model for the role of AP2-mediated LRP6 clustering in enabling an Axin-GSK3 complex to bind simultaneously to two LRP6 receptors. Each of these three findings provides hereto missing pieces that knit together to fill a substantial gap in our understanding of Wnt signalling. The data for the horseshoe model of the Axin-GSK3 interaction are less compelling given the different interface resulting from AlphaFold3 but are presented with well balanced arguments and certainly should be included.

There are a few points where revisions to the text would help the reader, particularly non Axin specialists.

P4 The sentence starting “We also provide evidence” is rather confusing, possibly because of the repeat use of “antagonize”.

We have **rephrased** most of this introductory paragraph on **p4**.

P7 BID suddenly appears in the text but isn't defined, maybe the schematic of Axin could be referenced?

We have replaced this abbreviation in the main text with 'beta-catenin-interacting region' (e.g. **p7, 2nd paragraph**) and now use 'BIR' only in the figure panels (but spell out the abbreviation in the corresponding legends).

P7 Ditto the sudden appearance of GID

We now define GIR when we first use this term (on **p8, 2nd paragraph**) and explained it in the legends where necessary.

SECOND REBUTTAL

Reviewer #1 (Remarks to the Author):

The authors have satisfactorily addressed my concerns, resulting in a significantly improved revised manuscript. This is an excellent paper that will make a valuable contribution to the Wnt field by providing some molecular clarity to the mechanism of Wnt pathway activation.

Reviewer #2 (Remarks to the Author):

The revisions have further improved an already excellent manuscript. All my points have been addressed satisfactorily. The revised version contains exciting new details of the multi-pronged interaction between AXIN and GSK3, predictions of the kinases responsible for AXIN phosphorylation, and a plausible potential explanation for a phosphorylation-regulated AXIN stability control system by USP7. As I pointed out in the initial round of review, this study provides important new insights into Wnt signalosome formation and covers a vast ground. The work will substantially advance the Wnt field and, given the proteomics components, serve as a valuable resource. There are many interesting observations whose exhaustive investigation go beyond the scope of a single manuscript, so the findings will open up many interesting avenues for future study. Congratulations to the authors!

There remain a few relatively small and easy-to-address points for the final revision prior to publication.

Specific points

- line 157: The authors report very similar bleach maps for WT or kinase-dead GSK3 and Lip-L1(LRP6). Readers may wonder how this fits with the phosphorylation dependency in the IP experiment (Figure 1b). It may help to highlight the different contexts of these experiments to address this question up-front.

We have **added** a sentence at the end of the relevant paragraph (at the bottom of **p5**) saying "Of note, phosphorylation of this element is not essential for the binding by either protein in vitro."

- line 183: The authors state that the residues upstream of LIRalpha (LIRup) do not interact with ¹⁵N-labelled Lip-L1(LRP6), cross-referencing Figure 2a. However, I don't think this is clear from that figure.

We have **tidied up** this sentence, which now reads: "However, we do not observe any interactions of the minimal LIR α nor of the Axin sequences upstream of LIR α (A308-P344, called LIRup; **Fig. 2a**) with ¹⁵N-labeled Lip-L1_{LRP6} (**Fig. 2f**).

- Figure 3: Can the stoichiometries referred to in the text (line 190) be indicated in the figure?

We would prefer not to add this figure into the ITC graphs since the 1:1 stoichiometry mentioned in the text is an assumption, albeit by far the most common assumption for any

two interacting proteins.

- lines 238/239: *The text states that intramolecular interactions are shown in Figure 4d, e. For clarity, the authors may choose a different wording here as the interactions themselves are not modelled.*

We have improved this sentence, to encapsulate the key feature of LIR, as follows: “Of note, the charge distribution is strikingly asymmetric even within unmodified LIR, with negative charges concentrated in LIR_{up} and positive charges in LIR α (**Fig. 4d, e**). This asymmetry would become even more pronounced upon phosphorylation of the serine residues in LIR_{up}.”

- p. 8, paragraph 3: *Are the described features similar in the top-ranking AlphaFold outputs? It will be good to include this information in the main text.*

We now **specify** this as follows: “Both versions of AlphaFold predict confidently (in each of the top-5 models; **Supplementary Fig. 5**) that the GSK3-interacting α -helix of Axin (GIR) extends further downstream than the minimal α -helix observed in Axin-GSK3 co-crystals”.

- *Generating the proteins for biochemical, biophysical and structural studies is technically challenging. Can the authors add SDS-PAGE gels showing the final proteins? I suppose the final buffer is sodium phosphate? (The counter-ion is not indicated.)*

We have **added** a new supplementary figure (**Supplementary Fig. 3**) that shows SDS-PAGE analysis of representative examples of purified Lip-A3_{Axin} and Lip-A5_{Axin} protein preparations used for NMR and ITC, and included the counter ion (sodium) of the final sample buffers in the Methods sections for NMR and ITC.

- lines 277-286: *Well-accepted terms of kinases (N-lobe, C-lobe) could be used, rather than "apex", etc.*

We have **added 'N-lobe'** when referring to the ‘front surface’ or the ‘apex’ of GSK3 (as both features belong to this lobe).

- *On the note of AlphaFold predictions, the authors describe pLDDT as a confidence measure for interactions, whereas it is a confidence metric for local structure (per residue). This of course is in turn indirectly linked to interaction confidence; however, the latter is better illustrated by the PAE score.*

Rather than going into the details of these scores and their impact on the confidence of the predicted models, we have **deleted** the pLDDT scores from these sentences and instead simply refer to Supplementary Fig. 5 where this is all laid out clearly.

- p. 11, last paragraph: *Can the ITC data be added as supplementary material? On the note of ITC, can the authors indicate how many experiments were performed?*

We already show four ITC graphs in Fig. 3, to document the affinities of GSK3 and wt and mutant Axin for their cognate LRP6-binding site, and three further ITC graphs in Fig. 4f), to document the intramolecular interaction mediating the foldback loop within Axin LIR, because all these measurements are central to our paper. By contrast, the low micromolar affinities between AP2 μ and its cognate motifs in LRP6 are as expected from previously published measurements of other proteins and their cognate AP2-binding motifs (i.e. these affinities are well established in the literature), so there is nothing new about these results.

We would therefore prefer not to include the corresponding graphs, especially since these would require yet another supplementary figure, but our paper already contains 13 supplementary figures plus a supplementary table, in addition to the 9 main figures.

- *Supplementary Figure 8: Can the contour level be indicated?*

Done

- *Supplementary Figure 9c appears to suffer from some graphics anomaly. In general, the supplementary file looks like a low-quality scan. Can this be rectified?*

Apologies: this was a glitch that occurred during the pdf conversion which we have now **corrected**.

- *Supplementary Figure 9c: No matter whether the recombinant gene is introduced transiently or stably, there is overexpression. I therefore suggest the nomenclature of "OE" to be changed to "transient".*

Done

- *Suggestion for title: "... flexibility of Axin and [the] AP2 clathrin adaptor"*

We have **inserted** [by the] in the title and have also **added** [the] in the main text whenever we refer to the AP2 clathrin adaptor.

- *Wnt>b-catenin  Wnt/b-catenin*

Done

- *p. 3: Myristoylation is not sufficient for membrane anchoring, and it is likely it needs to collaborate with palmitoylation for this purpose.*

It is correct that myristoylation is not sufficient for membrane anchoring, and so we have **changed** this sentence on **p3** to "an atypical member of the cyclin family that is myristoylated and recruits its cognate CDKs to the plasma membrane".

- *line 200: Figure 2h*

Done

- *Figure 6b: The subscript 4 is missing in the mutant variant name.*

Have **added** this subscript

- *Line 319: AF3 (instead of AF2)*

Done

- *Line 343: contributes  contribute*

We have **improved** this summary sentence on **p10**, as follow: "However, the reduced Wnt response consistently observed in both mutants supports the notion that the conserved

acidic residues and interspersed serines within LIRup contribute to the binding between Axin and LRP6.”

Reviewer #3 (Remarks to the Author):

The authors have addressed my concerns, mostly satisfactorily. They added valuable new data that support and solidify their model and the manuscript is better readable. However, there are still some issues that require attention:

- The newly expanded list of proteins identified using TurboID is interesting, especially in combination with the Kinase Library analysis. However, this also led to an addition of more speculative interpretations. The proposed roles of the kinases and the phosphorylated residues in the model presented in Figure 9 are primarily inferred from proximity labeling data, predictions from AlphaFold models and the Kinase Library. These conclusions thus currently lack support from direct interaction or functional validation, and this limitation should be acknowledged more explicitly in the text.

We have **added** sentences acknowledging this limitation at the end of the appropriate paragraphs in our Discussion on **p18** (“Future work will be required to determine the timing of these two key phosphorylations within LIR α and PRTxR and the identity of the kinases that impart them on Axin to promote its multipronged interaction with GSK”), **p19** (“...providing a context-dependent phosphorylation switch that would be interesting to test in future studies), **p20** (“Future studies are needed to assess whether, and under which circumstances, Axin may function as a direct competitive inhibitor of GSK3”) and **p21** (“Therefore, LRP6-associated kinases such as PKN1 could attenuate Wnt signaling by antagonizing LRP6 binding to AP2 μ , and hence its AP2-dependent clustering in the clathrin locales surrounding Wnt-occupied receptor complexes, which would be interesting to test in future”).

We would also like to emphasize that the predictions by AlphaFold and by the Kinase Library are largely (AF) or entirely (KL) based on experimental data, as are the kinase identifications by our proximity labelling. Furthermore, we have limited our proposals for individual kinases or their phosphorylated target sites (and for the model shown in Fig. 9) to those kinases that had already been linked compellingly to individual steps during Wnt signalling in previously published studies. We therefore believe that our hypotheses regarding these kinases and their targets are based on sufficiently firm grounds to warrant discussion.

- At many places in the manuscript, the authors mention ‘unpublished results’ or ‘personal communication’. For clarity and transparency, these results should be added to the supplemental information or properly referenced.

We have now **replaced** the references to unpublished results either with an explicit sentence describing the results obtained, or by **showing** the results in the **new panel of Supplementary Fig. 2a** or **new Supplementary Fig. 3**.

The personal communication to Yue et al (2025) — cited with permission — refers to a manuscript currently under revision: Yue D, Sun G, Cao Y, Li H, Yang Y, Pan Z, Xue L, Zhang L, Wang Z & Xu W (2025). Structural basis of Wnt signalosome assembly.

- The experiments shown in Figure 4g are confusing regarding the boundaries of LIRup vs

LIR α , as the 43-mer LIRup peptide contains a part of LIR α RKR motif as well (2 of the 3 RKR residues). This complicates the readout and interpretation. Please better explain the setup of this experiment.

We had to include the first 7 residues of LIR α (containing three positively charged amino acids and a hydrophilic glutamine) at the 3' ends of our 43mer LIRup peptides in order to render these soluble in the aqueous ITC buffer. It is true that these 7 residues of LIR α (including RK of the RKR triplet) may allow residual intramolecular feedback in the case of 43PPP, but this should only minorly affect the corresponding K_D value since Lip-LIR α is in 10x molar excess over its peptide ligands (as specified in the Methods). However, we now add a caveat in the legend of this figure, saying that the K_D value determined for 43PPP may represent an underestimate.

- Figure 5b and c: the disordered Axin region that binds to GSK3beta should be made better visible by using a more contrasting color.

We have thought long and hard about the colour scheme of this figure and experimented with different colour combinations, and so we very much doubt that we can improve on the current version. In our opinion, the colours we have chosen for the key Axin elements that interact with GSK3 (dark grey for GIR, turquoise for LIR α and ochre for the PRTxR motif) contrast very well with the colour of GSK3 itself (light grey).

- Figure 9: Please indicate the meaning of symbols within the figure itself and add arrows from kinase to target site, to make the figure more self-explanatory. Also, please indicate the consequences of the different phosphorylation events in the figure.

We have **revised** this figure, by including **arrows between the kinases and their targets** wherever possible (and state in the legend where they were omitted for clarity). We have also changed the **colour of CK1 γ to brown**, to match this kinase to its target sites on LRP6 and Axin, and we have **framed** the Axin-GSK3 complexes in the top row (with dotted lines) to relate these to the cartoons underneath which show them together with the kinases that phosphorylate them (now **indicated by arrows**, and the complexes are also **labelled** now). Importantly, we have **corrected a mistake** in the previous legend that referred to the target sites of CK1 γ as green diamonds and to those of CycY-activated CDK as brown diamonds (which will have caused confusion – apologies!).

However, we believe that it is preferable to describe the consequences of the phosphorylations in the legend (which will be underneath the model cartoon in the published version) rather than attempting to visualize these with further drawings, especially since our model is already quite complicated, with six individual cartoons.

- Line 600 – Fig 6c should be Fig 6d

Done